# GENERALIZATION AND DISTRIBUTED LEARNING OF GFLOWNETS

**Tiago da Silva**
Getulio Vargas Foundation
tiago.henrique@fgv.br

**Amauri Souza**
Federal Institute of Ceará
amauriholanda@ifce.edu.br

**Omar Rivasplata**
University of Manchester
omar.rivasplata@manchester.ac.uk

**Vikas Garg**
YaiYai Ltd and Aalto University
vgarg@csail.mit.edu

**Samuel Kaski**
Aalto University, University of Manchester
samuel.kaski@aalto.fi

**Diego Mesquita**
Getulio Vargas Foundation
diego.mesquita@fgv.br

## ABSTRACT

Conventional wisdom attributes the success of Generative Flow Networks (GFlowNets) to their ability to exploit the compositional structure of the sample space for learning generalizable flow functions (Bengio et al., 2021). Despite the abundance of empirical evidence, formalizing this belief with verifiable non-vacuous statistical guarantees has remained elusive. We address this issue with the first data-dependent generalization bounds for GFlowNets. We also elucidate the negative impact of the state space size on the generalization performance of these models via Azuma-Hoeffding-type oracle PAC-Bayesian inequalities. We leverage our theoretical insights to design a novel distributed learning algorithm for GFlowNets, which we call *Subgraph Asynchronous Learning* (SAL). In a nutshell, SAL utilizes a divide-and-conquer strategy: multiple GFlowNets are trained in parallel on smaller subnetworks of the flow network, and then aggregated with an additional GFlowNet that allocates appropriate flow to each subnetwork. Our experiments with synthetic and real-world problems demonstrate the benefits of SAL over centralized training in terms of mode coverage and distribution matching.

## 1 INTRODUCTION

Generalization is a long-standing problem in the machine learning literature, asking whether a learning algorithm can reliably make predictions beyond the data it was trained on (Valiant, 1984; Vapnik, 2000; Catoni, 2007; Alquier & Guedj, 2017; Dziugaite et al., 2021; Lotfi et al., 2024a). In an age of rapid deployment of AI models to end-users, there has been an emerging interest in the design of theoretically robust algorithms, with remarkable results for GANs (Mbacke et al., 2023), diffusion models (Li et al., 2024), transformers (Lotfi et al., 2024a;b), and graph neural networks (Ju et al., 2023; Tang & Liu, 2023). In this pursuit for developing models with proven generalizability, a rich set of tools has been created (Vapnik & Chervonenkis, 2015; Shalev-Shwartz & Ben-David, 2014), with McAllester (1998; 1999)'s PAC-Bayesian theorems often providing the tightest statistical guarantees (Pérez-Ortiz et al., 2021; Dziugaite & Roy, 2017; 2018; Lotfi et al., 2024b;a). Notably, however, there is no one-size-fits-all solution for understanding generalization: the diverse nature of data and learning algorithms demands a distinctly unique approach to each problem.

In the realm of probabilistic methods, for example, it has been widely hypothesized that the outstanding performance of Generative Flow Networks (GFlowNets, Bengio et al., 2021; 2023; Lahlou at al., 2023), which have demonstrated exceptional results in problems such as design of biological sequences (Jain et al., 2022; Malkin et al., 2022) and combinatorial optimization (Zhang et al., 2023a;b), to name a few, emerges from their potential to exploit the compositional structure of the underlying state space to learn a generalizable flow assignment function in a flow network by only

observing a fraction of the network's nodes (Bengio et al., 2021; Nica et al., 2022; Shen et al., 2023; Atanackovic & Bengio, 2024; Krichel et al., 2024). Nonetheless, in spite of the wealth of empirical evidence indicating that generalization occurs in GFlowNet learning (Nica et al., 2022; Atanackovic & Bengio, 2024), no work so far has provided non-vacuous high-probability empirical bounds on the population risk of GFlowNets, which might serve as statistical certificates for generalization.

Given the described scenario, in this paper we develop the first non-vacuous generalization bounds for GFlowNets in the literature in Section 5. In doing so, the core questions we want to address in this work are when GFlowNets (provably) generalize and which factors potentially contribute to diminish their generalization performance. To kickstart our analysis, we present in Section 4 an example in which a GFlowNet catastrophically fails to generalize even after learning a compatible flow assignment for over 90% of the flow network. This example demonstrates that, to properly understand the generalization of GFlowNets, we must consider not only the *extension* of the observed flow network but also the *specific parts* that have been encountered during training, a fact that is also implicit in popular techniques such as the replay buffer (Vemgal et al., 2023) and local search (Kim et al., 2024b). From a technical perspective, this implies that the development of meaningful statistical guarantees for GFlowNets must be based not on data-agnostic theoretical results, but on data-dependent priors, in the fashion of Dziugaite & Roy (2018); Dziugaite et al. (2021); Pérez-Ortiz et al. (2021). This observation guides the establishment of the non-vacuous high-probability bounds for the population risk of GFlowNets in Section 5.1 and distinguishes our approach from other investigations of GFlowNets generalization (Krichel et al., 2024).

The empirical results in Section 5.1, however, do not provide a fine understanding of which characteristics of the flow network tend to hinder the generalization of GFlowNets. What effect do larger trajectory lengths, for instance, have on the provable learnability of generalizable flow assignments? Intuitively, generalization is harder in larger state spaces as, borrowing the terminology from the reinforcement learning (RL) literature (Bengio et al., 2021), the visited portion of an environment by an agent constrained by a fixed time budget decreases with increasing environment's size and, therefore, the agent would have to rely on increasingly sparse information for learning from larger trajectories. In Section 5.2, we formalize this intuition through the lens of PAC-Bayesian bounds, revealing the increasing difficulty in obtaining tight statistical certificates for larger state spaces. To achieve this, our technical contributions are two-fold: First, an oracle concentration inequality for the forward Kullback-Leibler (KL) divergence between the learned and targeted flow assignments (Theorem 5.2) inspired by Malkin et al. (2023)'s interpretation of GFlowNets as variational inference. Second, an Azuma-Hoeffding-type inequality (Seldin et al., 2012b) for independently sampled martingales representing the extent to which the learned flow assignment violates the detailed balance condition (Bengio et al., 2023) in the observed trajectories (Theorem 5.4).

Motivated by these results, we pose the following question: what can we do to mitigate the issues raised by an inadequate coverage of the state space, as illustrated in Section 4, and larger trajectory sizes, as analyzed in Section 5? As we show in Section 6, the answer lies in breaking up the flow network into *multi-source subnetworks* grouped together by a small *root network* (Figure 3). In this new paradigm, finding a compatible flow assignment becomes a two-stage process. Firstly, a different GFlowNet is trained on each subnetwork in a distributed fashion. Secondly, an additional GFlowNet trained on the root network learns to assign the correct amount of flow to each subnetwork, as estimated in the previous step. The resulting algorithm, which we call *Subgraph Asynchronous Learning* (SAL), has several advantages over the standard approach (Section 6.2). In particular, each GFlowNet within this framework needs to solve a problem that is relatively simpler than that of a unique, centralized model. Similarly, the asynchronous nature of the algorithm implies that we are able to visit a considerably larger fraction of the original flow network within a fixed time window and, therefore, that we get a significantly better coverage of the state space. As a consequence of this, we are also able to drastically improve the discovery of high-value states within the flow network, which is a metric of great interest in the GFlowNet literature (Bengio et al., 2021; Shen et al., 2023; Zhang et al., 2023a; Pan et al., 2023a; Jang et al., 2024; Kim et al., 2024b).

In summary, our contributions are:

1. We construct a family of examples in which a GFlowNet does not generalize even after learning a compatible flow assignment on arbitrarily large fractions of the flow network (Section 4);

2. We provide the first non-vacuous generalization bounds for GFlowNets (Section 5.1);

3. We derive oracle PAC-Bayesian inequalities for the population risk of GFlowNets, emphasizing the impact of the flow network's topology on generalization performance (Section 5.2);

4. We design the first distributed algorithm for learning GFlowNets with network-level parallelization, and evaluate its performance in common benchmarks in the literature (Section 6);

The first part of the paper establishes the notation and terminology to be used throughout the paper and reviews relevant results in the GFlowNet and PAC-Bayes literature, alongside an overview of our main contributions (Sections 2 and 3). The second part provides a formal treatment of GFlowNets from the viewpoint of the PAC-Bayesian theory (Section 5). The third and final part outlines the foundations of SAL and conducts an empirical evaluation of the algorithm in common benchmark problems (Section 6). We defer the proofs and details of the experiments to the supplement.

## 2 PRELIMINARIES

**Notations and terminology.** Let $G = (V, E)$ by a directed acyclic graph (DAG). A *forward policy* over $G$ is a Markov transition kernel $p_F \colon V \times V \to [0, 1]$ supported on $G$'s edges, i.e., such that $p_F(v, \cdot)$ is a distribution over $\{u \colon (v, u) \in E\}$, for each vertex $v$. We interchangeably use $p_F(\cdot|v)$ and $p_F(v, \cdot)$ for representing $p_F$. A *backward policy* $p_B$ over $G$ is a forward policy on the transpose graph $G^\top = (V, E^\top)$ with $E^\top = \{(u, v) \colon (v, u) \in E\}$. The *uniform* policy assigns the same probability mass to a state's children, i.e., $p_U(s'|s) = \mathbf{1}_{\{s' \in \mathrm{Ch}(s)\}}/|\mathrm{Ch}(s)|$ with $\mathrm{Ch}(s) = \{s' \colon (s, s') \in E\}$. We say that $G$ is *pointed* if there are nodes $s_o$ and $s_f$, respectively called *initial* (source) and *final* (sink) nodes, s.t. $s_o$ (resp. $s_f$) is the only node without incoming (resp. outgoing) edges and, for each $s \in V$, there is a trajectory (directed path) between $s_o$ and $s_f$ containing $s$. In this case, a trajectory $\tau$ in $G$ is *complete* if it starts at $s_o$ and finishes at $s_f$, which we denote by $\tau \colon s_o \rightsquigarrow s_f$. Clearly, a forward policy induces a distribution over trajectories starting at $s$ via $p_F(\tau|s) = \prod_{(s', s'') \in \tau} p_F(s''|s')$; when $\tau$ is unambiguously complete, we will often omit $s_o$ from this notation. Lastly, for probability measures $P$ and $Q$ on the same space, we let $\mathrm{KL}(P||Q)$, $\chi^2(P||Q)$ and $\mathrm{TV}(P, Q)$ respectively denote their Kullback-Leibler (KL) divergence, $\chi^2$ divergence, and total variation distance.

**GFlowNets.** We represent a GFlowNet (Bengio et al., 2021; 2023; Lahlou at al., 2023) $\mathcal{G}$ as a tuple $(\mathcal{S}, \mathcal{X}, G, A, \mathcal{T}, p_F, p_B, R, F)$ consisting of a set of *states* $\mathcal{S}$, a set of *terminal states* $\mathcal{X} \subseteq \mathcal{S}$, a pointed DAG $G = (\mathcal{S}, E)$, which is called a *state graph*, an *action mapping* $A \colon \mathcal{S} \to 2^{\mathcal{A}}$ associating each state $s$ with an abstract action space $A(s) \subseteq \mathcal{A}$ that is isomorphic to the children of $s$ in $G$, a *transition function* $\mathcal{T} \colon \cup_{s \in \mathcal{S}} (\{s\} \times A(s)) \to \mathcal{S}$ defining how a state $s$ is affected by an action $a \in A(s)$, *forward* $p_F$ and *backward* $p_B$ policies on $G$, a *reward function* $R \colon \mathcal{X} \to \mathbb{R}_+$ attributing a positive value to each terminal state, and a *flow function* $F \colon \mathcal{S} \to \mathbb{R}_+$ such that $F|_{\mathcal{X}} = R$. Importantly, only the elements of $\mathcal{X}$ are connected to the sink node $s_f$ of $G$. When there is no risk of ambiguity, we will simply write $\mathcal{G} = (p_F, p_B, F)$. The objective of a GFlowNet is to find a $p_F$ s.t. the marginal distribution $p_T(x) \coloneqq p_F(x|s_o) = \sum_{\tau \colon s_q \rightsquigarrow x} p_F(\tau)$ over $\mathcal{X}$ matches $R$ up to a normalizing factor (Bengio et al., 2021). In Appendix A, we illustrate how this abstract representation can be instantiated to accommodate three frequently considered use-cases (Malkin et al., 2022; 2023).

**Learning GFlowNets.** In this context, $\mathcal{S}, \mathcal{X}, G, A, \mathcal{T}$, and $R$ are problem-dependent, while $p_F, p_B$, and $F$ are unknowns that should be estimated. Remarkably, however, $p_B$ is often fixed as uniform (Shen et al., 2023; Liu et al., 2023; Zhang et al., 2023a), an assumption that we make throghout the paper, albeit most of our theoretical results and all our methods can be extended to the case of learnable $p_B$. Under these circumstances, many learning objectives have been proposed for learning $p_F$ and $F$. Two popular choices, which we adopt here, are the *trajectory balance* (TB, $\mathcal{L}_{\mathrm{TB}}(p_F, F)$, Malkin et al., 2022) and *detailed balance* (DB, $\mathcal{L}_{\mathrm{DB}}(p_F, F)$, Bengio et al., 2023) losses,

$$\mathbb{E}_{\tau \sim p_E}\left[\left(\log \frac{F(s_o)p_F(\tau)}{R(x)p_B(\tau|x)}\right)^2\right] \quad \text{and} \quad \mathbb{E}_{\tau \sim p_E}\left[\frac{1}{|\tau|}\sum_{(s,s') \in \tau}\left(\log \frac{p_F(s'|s)F(s)}{p_B(s|s')F(s')}\right)^2\right], \quad (1)$$

in which $|\tau|$ represents $\tau$'s length, $x$ is $\tau$'s (unique) terminal state, and $p_E$ is an *exploratory policy*. Intuitively, $p_E$ has the role of the data-generating distribution in a standard supervised learning context and is often defined as $p_E = (\epsilon)p_U + (1 - \epsilon)p_F$, an $\epsilon$-*greedy* version of $p_F$, with $p_U$ denoting an uniform policy; although more sophisticated techniques have been developed (Kim et al., 2024b; Rector-Brooks et al., 2023; Vemgal et al., 2023). In Appendix A we provide a more

thorough overview of GFlowNet learning, including the subtrajectory balance loss (SubTB, Madan et al., 2022) and divergence-based objectives (Malkin et al., 2023; Lahlou at al., 2023).

**Generalization bounds for neural networks.** The field of statistical learning theory (Vapnik, 1998; 2000) seeks to develop statistical certificates for the generalization of a learned model by providing high-probability upper bounds on the population error of an estimator as a function of the observed empirical risk. In the context of GFlowNets, we ask whether an empirically measured imbalance based on the observed trajectories, such as the losses in Equation 1 or other locally computed metrics (see Sections 4 and 5), are appropriate surrogates for the GFlowNet's overall distributional accuracy. In particular, we are interested in *inductive* statistical guarantees, namely, those based on the training set (as opposed to the *transductive* setting, in which a test set is used). To this end, the PAC-Bayes framework of McAllester (1998; 1999; 2013) often provides the tightest bounds (Pérez-Ortiz et al., 2021; Lotfi et al., 2024b;a; Dziugaite & Roy, 2017). In a nutshell, consider data $\mathbf{X} = \{X_i\}_{i=1}^m$ drawn from some data distribution, a significance level $\delta$, an empirical loss $\hat{\mathcal{L}}(\theta, \mathbf{X})$ and a population loss $\mathcal{L}(\theta) = \mathbb{E}_{\mathbf{X}}[\hat{\mathcal{L}}(\theta, \mathbf{X})]$ associated to the model's parameters $\theta$. Given a 'prior' (independent of $\mathbf{X}$) distribution $Q$ over $\theta$, a PAC-Bayes bound typically assumes the form

$$\mathbb{E}_{\theta \sim P}[\mathcal{L}(\theta)] \leq \mathbb{E}_{\theta \sim P}[\hat{\mathcal{L}}(\theta, \mathbf{X})] + \phi(\delta, P, Q, m). \tag{2}$$

The inequality holds with probability $1 - \delta$ over draws of $\mathbf{X}$, simultaneously for all 'posterior' distributions $P$ over $\theta$; and $\phi$ is a term penalizing the model's complexity (McAllester, 1999). We direct the reader to Alquier (2024) for a comprehensive introduction to PAC-Bayesian analysis. For bounded $\mathcal{L}$, the right-hand side of Equation 2 is termed *vacuous* if it is larger than an upper bound of $\mathcal{L}$. Although McAllester's original works posited that the data were independent and identically distributed (i.i.d.) and that the risk function was uniformly bounded (McAllester, 1998; 1999), recent advances relaxed these assumptions by deriving generalization bounds for non-i.i.d. data (Seldin et al., 2012b; Barnes et al., 2022), with applications to multi-armed bandits and RL (Fard & Pineau, 2010; Beygelzimer et al., 2011; Tasdighi et al., 2024), and for unbounded losses limited by high-probability bounds (Alquier & Guedj, 2017; Haddouche & Guedj, 2023; Casado et al., 2024; Mbacke et al., 2023). To the best of our knowledge, however, this is the first work promoting the development of PAC-Bayesian bounds for understanding the generalization of GFlowNets.

## 3 OVERVIEW OF OUR RESULTS

Before delving into the details of our work in Sections 4, 5, 6 (and further details in the appendices in the supplement), we provide below a brief discussion around our technical results under the light of the formalism presented in Section 2, alongside the main ideas they were built upon.

**Non-vacuous generalization bounds for GFlowNets.** The learning objectives in Equation 1, due to the unboundedness of the logarithm, cannot be directly incorporated into standard PAC-Bayesian theorems (McAllester, 2013), which assume that the risk function has at least bounded exponential moments (Casado et al., 2024; Rodríguez-Gálvez et al., 2024a;b). To circumvent this issue, our empirical analysis in Section 5.1 adopts the recently proposed FCS metric (Silva et al., 2024) as the risk functional measuring the accuracy of a trained GFlowNet, which may be written as

$$L_{\text{FCS}}(p_F) = \mathbb{E}_{(\tau_1, x_1), \dots, (\tau_B, x_B)} \left[ \text{TV} \left( p_T^{x_{1:B}}, R^{x_{1:B}} \right) \right] \in [0, 1], \tag{3}$$

where $p_T^{x_{1:B}}$ and $R^{x_{1:B}}$ are the respective restrictions of $p_T$ and $R$ to the $B$-sized multiset $\{\{x_1, \dots, x_B\}\} \subseteq \mathcal{X}$ of terminal states, and TV is the total variation distance. However, in spite of easily computable, $L_{\text{FCS}}$ is not an appropriate learning objective for GFlowNets due to the potential numerical instability of the non-log-domain. Instead, we minimize $\mathcal{L}_{\text{TB}}$ as a surrogate objective for $L_{\text{FCS}}$ during training and evaluate the generalization bound on $L_{\text{FCS}}$ in the inductive fashion mentioned in Section 2. Importantly, Figure 2 shows that the resulting bounds are remarkably tight.

**Oracle generalization bounds for GFlowNets.** As a complement, we also establish non-empirical high-probability upper bounds on the population risk of GFlowNets by assuming that a potentially intractable quantity bounds the corresponding loss function. In Section 5.2, we follow this rationale and demonstrate that there always is an $\alpha > 0$ for which the set of policy networks of the form $\alpha p_U + (1 - \alpha)p_F$ contains the solution to the flow assignment problem. Armed with such a family of models, which guarantee log probabilities uniformly bounded away from zero, we consider

the reverse KL divergence risk (Malkin et al., 2023) to avoid explicitly bounding the flow function $F$. Although informative, the resulting Theorem 5.2 only considers the trajectories—and not transitions—as data points. As the number of observed transitions is significantly larger than that of trajectories, we enrich our results by constructing a martingale difference sequence based on the DB loss and adapting Azuma's inequality (Azuma, 1967; Seldin et al., 2012b) to the context of independent martingales to derive a transition-level generalization bound for GFlowNets. Both approaches, which are respectively encapsulated in Theorems 5.2 and 5.4, show that the population risk can be bounded with high-probability as, apart from technical nuances,

$$\mathbb{E}_{\theta \sim P}[\mathcal{L}(\theta)] \lesssim \mathbb{E}_{\theta \sim P}[\hat{\mathcal{L}}(\theta)] + \mathcal{O}\left(\frac{\log t_m}{n^\alpha}\right) \quad (4)$$

in which $\hat{\mathcal{L}}$ is an empirical measure of risk, $t_m$ is the maximum trajectory length of the state graph, and $n$ is the number of observed data points: either trajectories (Theorem 5.2, $\alpha = 0.5$) or transitions (Theorem 5.4, $\alpha = 1$). From an analytical perspective, these results suggest that learning provable generalizable flow assignments is increasingly harder for state spaces having longer trajectories.

## 4 WHEN DO GFLOWNETS NOT GENERALIZE?

To start our discussion on the generalization of GFlowNets, we introduce simple, but non-trivial, examples in which a GFlowNet does not learn a generalizable policy network even after minimizing the loss on an arbitrarily large portion of the state space, raising the questions of *when* do GFlowNets generalize and *how* to measure such generalization, which we investigate in Sections 5.1 and 5.2.

**A non-generalizable data distribution.** To concretize our arguments, we recall the task of set generation for GFlowNets (Pan et al., 2023a;b; Bengio et al., 2023; Jang et al., 2024). Each state corresponds to a subset of a set $\mathcal{W} = \{1, \ldots, W\}$ for a given $W$; the generative process starts at an empty set $s_o = \emptyset$ and iterativey adds elements from $\mathcal{W}$ to $s_o$ until a prescribed size $T$ is achieved. For our purposes, we fix a function $u \colon \mathcal{W} \to [0, 1]$, representing the *log-utility* of each $w \in A(s_o) := \mathcal{W}$ and define the reward $R$ associated to $S$ as $R(S) = \mathbf{1}_{\{\#S=T\}} \exp\{\sum_{w \in s} u(w)\}$. Also, let $p_E$ be a forward policy s.t. $p_E(\cdot|s)$ is supported on $A(s) \setminus \{1\} := \mathcal{W} \setminus (\{1\} \cup s)$ for every $s$, i.e., the support of the marginal $p_{E,T}$ of $p_E$ on $\mathcal{X}$ is the set $\mathcal{X}'$ of subsets of $\{2, \ldots, W\}$. We next show that $\mathcal{X}'$ covers an arbitrarily large portion of $\mathcal{X}$ for specific choices of $T$ and $W$.

**Lemma 4.1.** *For each $\xi \in (0, 1)$, there exist $T$ and $W$ such that $|\mathcal{X}'| \geq \xi|\mathcal{X}|$.*

The (straightforward) proof of Lemma 4.1 can be found in Appendix D. Obviously, we cannot hope that a GFlowNet trained by minimizing an empirical risk defined on trajectories sampled from $p_E$ would generalize to unseen states, as no information regarding $u(1)$ would be available during training. To empirically validate our reasoning, we show in Figure 1 that a GFlowNet trained on samples from $p_E$ fails to learn the right distribution, whereas a standard $\epsilon$-greedy strategy succeeds. It is remarkable, however, that a GFlowNet is unable to successfuly sample from the target distribution even after minimizing the empirical risk on samples covering over 90% of the state space. From a statistical viewpoint, this

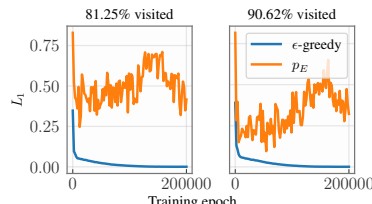

Figure 1: Convergence speed when actions are masked (blue) or not (orange) for different state space sizes.

behavior can be explained via a change of measure inequality: preference over states is not properly captured by the sampling distribution ($p_E$). We formalize this intuition in the proposition below.

**Proposition 4.2** (Generalization depends on the sampling distribution). *Let $(p_F, p_B, R)$ be a GFlowNet and $p_{E,T}$ be (any) distribution over $\mathcal{X}$. Also, recall $\pi(x)$ represents the normalized target and $p_T$ the learned marginal. Define $q_{E,T}$ as an uniform PMF on $\mathcal{X}$, i.e., $q_{E,T}(x) = 1/|\mathcal{X}|$. Then,*

$$\mathrm{TV}\left(p_T, \pi\right) \lesssim \sqrt{\left(1 + \chi^2(q_{E,T}||p_{E,T})\right) \mathbb{E}_{x \sim p_{E,T}}\left[\mathbb{E}_{\tau \sim p_B(\tau|x)}\left[\left(\log \frac{p_F(\tau)}{\pi(x)p_B(\tau|x)}\right)^2\right]\right]}, \quad (5)$$

*in which $\chi^2(P||Q)$ represents the $\chi^2$ divergence between $P$ and $Q$.*

We interpret Equation 5 in the following way: If the sampling policy ($p_{E,T}$) greatly deviates from the uniform ($q_{E,T}$), then a small empirical risk does not necessarily ensure an accurate distributional approximation. In contrast, Equation 5 does *not* entail that the uniform distribution is the optimal choice for sampling trajectories, as it does not address the algorithmic difficulty of minimizing the empirical risk via SGD. Illustratively, we show in Table 1 in Appendix B the values of $\chi^2(q_{E,T}||p_{E,T})$ when $p_{E,T}$ is far away from $q_{E,T}$ and of $\chi^2(q_{E,T}||p_{\epsilon,T})$ for the $\epsilon$-greedy policy considered in Figure 1. On a fundamental level, these examples underline the importance of taking into account the data distribution for understanding generalization performance. Section 5 elaborates on this problem through the lens of PAC-Bayes bounds, albeit with data-dependent priors (Dziugaite & Roy, 2017; 2018; Dziugaite et al., 2021; Pérez-Ortiz et al., 2021).

## 5 PAC-BAYESIAN GENERALIZATION BOUNDS FOR GFLOWNETS

Towards the objective of understanding GFlowNet generalization, we construct high-probability upper bounds on different risk functions. In Section 5.1, we build upon McAllester's empirical bound and Dziugaite's data-dependent priors to derive the first non-vacuous generalization bounds for GFlowNets. Then, to gain a clearer understanding of the factors hindering the generalizability of these models, we provide both trajectory- and transition-level oracle bounds in Section 5.2 by drawing upon the martingale-based PAC-Bayesian theory for non-i.i.d. data (Beygelzimer et al., 2011).

### 5.1 NON-VACUOUS EMPIRICAL GENERALIZATION BOUNDS

**GFlowNet learning as supervised learning.** To rigorously address the generalization of GFlowNets, we firstly frame the training of these models as a supervised learning problem (Shalev-Shwartz & Ben-David, 2014; Atanackovic & Bengio, 2024). For this, we assume that a set of independently sampled complete trajectories, $\mathcal{T}_n = \{\tau_1, \ldots, \tau_n\}$, is drawn from a fixed distribution and that each trajectory $\tau_i$ is annotated with a noise-free target, $y_i = p_B(\tau_i|x_i)R(x_i)$, with $x_i$ representing $\tau_i$'s unique terminal state. Importantly, the only supervision during training comes from the reward function; we do not make assumptions on the distribution over $\mathcal{T}_n$. In this context, minimizing $\mathcal{L}_{\mathrm{TB}}$ corresponds to finding the least-squares solution to the equation $\log Z + \log p_F(\tau) = \log p_B(\tau|x)R(x)$ in $p_F$. Importantly, this setting differs from conventional GFlowNet training algorithms, for which the sampling policy depends on the trajectories observed so far, that is, the trajectories are not independently sampled. Nonetheless, the question of whether GFlowNets generalize remains relevant even under our relatively simplified conditions, which may be seen as a single-iteration of an $\epsilon$-greedy strategy (Krichel et al., 2024).

**A bounded risk functional for GFlowNets.** As mentioned, PAC-Bayesian theory originally relied on the assumption of bounded risk functions (McAllester, 1998). Despite recent advances in extending the theory to unbounded losses (Casado et al., 2024; Rodríguez-Gálvez et al., 2024a;b; Haddouche & Guedj, 2023), most generalization bounds still depend on technical and hard-to-verify conditions, e.g., on exponential moments. For this reason, we use the FCS metric as a measure of risk (Silva et al., 2024); see Appendix B for an unbiased estimator $\hat{L}_{\mathrm{FCS}}(p_F, \mathcal{T}_n)$ of $L_{\mathrm{FCS}}(p_F)$.

**Data-dependent priors for PAC-Bayes.** For this, we first recall the techniques originally developed by Dziugaite & Roy (2017; 2018); Dziugaite et al. (2021) in a striking series of papers for probing the generalization of overparameterized neural networks in the supervised learning context. To start with, we state below Dziugaite et al. (2021)'s empirical PAC-Bayes bound, which combines results from McAllester (2013), Rivasplata et al. (2019), and Boucheron et al. (2013). For completeness, we also provide a self-contained proof of Proposition 5.1 in Appendix D in the supplement.

**Proposition 5.1** (Empirical PAC-Bayesian bounds)**.** *For any distribution $\zeta$ on parameters $\theta$ of $p_F$, let $L_{\mathrm{FCS}}(\zeta) = \mathbb{E}_{\theta \sim \zeta}[L_{\mathrm{FCS}}(R, p_T)]$ and define $\hat{L}_{\mathrm{FCS}}(\zeta, \mathcal{T}_n)$ similarly. Also, let $\alpha \in (0, 1)$ and let $P$ be a distribution on $\theta$ learned on an uniformly random $\lfloor (1-\alpha)n \rfloor$-sized subset $\mathcal{T}_{1-\alpha}$ of $\mathcal{T}_n$. Then,*

$$L_{\mathrm{FCS}}(P) \le \hat{L}_{\mathrm{FCS}}(P, \mathcal{T}_{1-\alpha}) + \min \left\{ \begin{array}{l} \eta + \sqrt{\eta(\eta + 2\hat{L}_{\mathrm{FCS}}(P, \mathcal{T}_{1-\alpha}))}, \\ \sqrt{\frac{\eta}{2}}, \end{array} \right. \tag{6}$$

*with probability at least $1 - \delta$ over $\mathcal{T}_{1-\alpha}$, in which $\eta := \frac{\mathrm{KL}(P||Q) + \log 2\sqrt{\lfloor (1-\alpha)n \rfloor}/\delta}{\lfloor (1-\alpha)n \rfloor}$ and $Q$ is a distribution that does not depend on $\mathcal{T}_{1-\alpha}$ but may depend on $\mathcal{T}_\alpha := \mathcal{T}_n \setminus \mathcal{T}_\alpha$.*

When the prior distribution $Q$ is naively chosen (e.g., as a standard Gaussian distribution), the KL divergence in Equation 6 often dominates the right-hand side of the equation and results in vacuous bounds, i.e., $L_{\text{FCS}}(P) \leq a$ for some $a > 1$. To address this issue, the influential work of Dziugaite & Roy (2017) proposed the use of a *data-dependent $Q$* learned by minimizing the empirical risk functional on a fraction $\alpha$ of the data and, after learning $P$ by minimizing Equation 6, evaluating the generalization bound on the remaining $(1 - \alpha)$ portion of the data, as presented in Proposition 5.1.

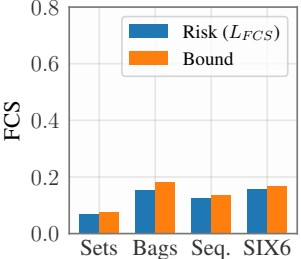

Figure 2: Non-vacuous generalization bounds for the FCS risk functional in Eq. 10.

**Empirical results.** We follow a similar approach to derive the first non-vacuous generalization bounds for GFlowNets in the literature. For this, we disjointly partition the dataset $\mathcal{T}_n$ with $n = 3 \cdot 10^4$ into sets $\mathcal{T}_\alpha$ and $\mathcal{T}_{1-\alpha}$ with $\alpha = 0.6$. We learn an isotropic Gaussian prior $Q$ on $\mathcal{T}_\alpha$ and then a diagonal Gaussian posterior $P$ on $\mathcal{T}_\alpha \cup \mathcal{T}_{1-\alpha}$ by minimizing the bound in Equation 6. Finally, the bound is evaluated on $\mathcal{T}_{1-\alpha}$ to obtain the statistical certificate (Pérez-Ortiz et al., 2021). Results in Figure 2 for the tasks of set generation (Pan et al., 2023a; Bengio et al., 2023), bag generation (Shen et al., 2023; Jang et al., 2024), sequence design (Malkin et al., 2022; 2023; Madan et al., 2022) with additive rewards, and SIX6 (Jain et al., 2022; Malkin et al., 2022; Shen et al., 2023) highlight the non-vacuousness of Equation 6 and the generalizability of the trained models. Please refer to Section 6 and to Appendix B for a detailed description of the experimental setup. Appendix A describes the design of the GFlowNet for each of these generative tasks.

## 5.2 ORACLE GENERALIZATION BOUNDS

Although the previous section's empirical results certify the generalization of the learned policy network to novel trajectories, they do not necessarily shed light on which characteristics of the generative task are hindering the model's generalization capability. In the remaining of this section, we thus derive generalization bounds that, despite not being directly computable, provide a finer understanding of which factors play a role when the goal is to learn a generalizable policy. In particular, we observe that larger trajectories and peakier target distributions tend to make generalization harder when a fixed sampling budget is available. In Section 6, we will see how a distributed algorithm may alleviate these issues (Yagli et al., 2020; Barnes et al., 2022; Sefidgaran et al., 2022).

**Trajectory-level bounds.** We start by deriving generalization bounds for GFlowNets when the trajectories are independently sampled (see Section 5.1) and the risk functional is the KL divergence between the forward and backward policies, i.e., $\text{KL}(p_B || p_F)$, in which $p_B(\tau) \propto p_B(\tau|x)R(x)$. This choice is motivated by Malkin et al. (2023)'s interpretation of GFlowNets as a hierarchical variational inference algorithms and by the ability of $\text{KL}(p_B || p_F)$ to focus the model on high-probability regions of the target, which is a desirable trait of GFlowNets. Remarkably, we show in Lemma B.1 that $\text{KL}(p_B || p_F)$ can be bounded by sensibly reparameterizing $p_F$ as a mixture policy. Then, as shown in Theorem 5.2 below, this reparametrization enables developing oracle generalization bounds in the fashion of the tight results we derived in Section 5.1.

**Theorem 5.2.** *Let $\mathcal{G} = (p_F, p_B, F)$ be a GFlowNet with policy network $p_F$ parameterized as in Lemma B.1. Also, let $Q$ be a probability distribution over the parameters $\theta$ of $p_F$. Denote $H[p_B] = -\mathbb{E}_{\tau \sim p_B}[\log p_B(\tau)]$ for $p_B$'s entropy and $M_T = \max_\tau(|\tau| \log(\alpha^{-1} \max_{s \in \tau} |\text{Ch}(s)|))$. Then,*

$$\mathbb{E}_{\theta \sim P}[\text{KL}(\pi || p_T)] \leq \mathbb{E}_{\theta \sim P}\left[\frac{1}{m} \sum_{1 \leq i \leq m} \log \frac{p_B(\tau_i)}{p_F(\tau_i)}\right] + (-H[p_B] + M_T)\,\eta(P, Q, n), \quad (7)$$

*in which we recall that $\eta(P, Q, n) = \sqrt{\frac{\text{KL}(P||Q) + \log 2\sqrt{n}/\delta}{n}}$ and $\pi(x) \propto R(x)$ is the target.*

A few remarks on the excess risk upper bound of Theorem 5.2. Firstly, the assumption that trajectories are sampled according to $p_B(\tau) \propto p_B(\tau|x)R(x)$ is consistent with popular strategies for learning GFlowNets that focus on sampling trajectories leading to high-reward states more often than those leading to low-reward states, e.g., using a replay buffer (Deleu et al., 2022). Secondly, in alignment with well-established practical knowledge, the result in Equation 7 shows it is harder to

achieve tighter generalization bounds when the target distribution is spiky with a small entropy term $H[p_B]$, and when the generative task is composed of longer trajectories or larger action spaces.

**Transition-level bounds.** For many applications, the number of observed complete trajectories when training GFlowNets can be orders of magnitude smaller than the number of collected state transitions. In this context, one may obtain significantly tighter generalization bounds by interpreting the transitions, and not the complete trajectories, as data samples (Lotfi et al., 2024a;b). Indeed, it is assumed that GFlowNets' outstanding potential emerges from its capacity to exploit the compositional structure of the space characterized by the state graph (Bengio et al., 2021; Nica et al., 2022; Shen et al., 2023; Atanackovic & Bengio, 2024). To incorporate this structure into our theoretical bounds, we shift our focus to the design of Azuma-Hoeffding-type concentration inequalities (Azuma, 1967; McDiarmid, 1998; Boucheron et al., 2013) applied to the stochastic process induced by the Markov Decision Process (MDP) governing the data-generating process. For this, we start defining a martingale difference sequence based on the DB loss (Bengio et al., 2023, Example 5).

**Definition 5.3** (A martingale difference sequence for the DB loss). Recall the detailed balance loss $\mathcal{L}_{\mathrm{DB}}(s, s') = (\log F(s)p_F(s'|s) - \log F(s')p_B(s|s'))^2$. For a fixed sampling policy $p_E$, we let

$$M(S_i, S_{<i}) = \mathcal{L}_{\mathrm{DB}}(S_i, S_{i-1}) - \mathbb{E}_{s_i \sim p_E(\cdot|S_{i-1})} \left[\mathcal{L}_{\mathrm{DB}}(s_i, S_{i-1})|S_{i-1}\right], \quad (8)$$

where $S_{<i} = \{S_1, \ldots, S_{i-1}\}$. Also, we define the natural filtration $\mathcal{F}_t = \sigma(S_1, \ldots, S_t)$ generated by the first $t$ states of the Markov process $\{S_i\}_{i \geq 1}$. Clearly, each $M(S_i, S_{<i})$ is $\mathcal{F}_i$-measurable and $\mathbb{E}_{S_i}[M(S_i, S_{<i})|\mathcal{F}_{<i}] = 0$, i.e., $\{M(S_i, S_{<i})\}_{i \geq 1}$ is a martingale difference sequence.

From this definition, it is immediate that $M_t := \sum_{1 \leq i \leq t} M(S_i, S_{<i})$ is a martingale w.r.t. the filtration $\{\mathcal{F}_t\}_{t \geq 1}$. We defer to Appendix B the discussion regarding its properties and the assumptions imposed on it for proving Theorem 5.4 below. Additionally, we define

$$\mathcal{L}(\theta) = \mathbb{E}_{\tau \sim p_E} \frac{1}{|\tau|} \sum_{1 \leq i \leq |\tau|} \mathbb{E}\left[\mathcal{L}_{\mathrm{DB}}(S_i, S_{i-1})|S_{i-1}\right] \text{ and } \hat{\mathcal{L}}(\theta) = \frac{1}{n} \sum_{1 \leq j \leq n} \frac{1}{t_j} \sum_{1 \leq i \leq t_j} \mathcal{L}_{\mathrm{DB}}(S_i^{(j)}, S_{i-1}^{(j)})$$

as the population and empirical DB-based risk functionals for GFlowNets. Under these conditions, Theorem 5.4 complements Theorem 5.2 with a generalization bound based on the DB loss.

**Theorem 5.4** (Transition-level generalization bounds for GFlowNets). *Let $M_t(\theta)$ be the martingale arising from Definition 5.3, with $\theta$ representing the parameters of the forward policy. Also, let $Q$ be a distribution on $\theta$. Assume that $\mathcal{L}_{\mathrm{DB}}(S_i, S_{i-1}) \leq U$ uniformly on $(S_i, S_{i-1})$ and that $M_{t_m}(\theta)^2 \leq K$, in which $t_m$ is the maximum trajectory length. Similarly, define $\lambda \leq 1/2U$ and $\beta \in (0, 1)$, and let $P$ be a data-dependent posterior distribution on $\theta$. Then, with probability at least $1 - \delta$ over the set of independent martingales $\{S_o, S_1^{(j)}, \ldots, S_{t_j}^{(j)}\}_{1 \leq j \leq n}$ such that $S_o = s_o$ almost surely,*

$$\mathbb{E}_{\theta \sim P}\left[\mathcal{L}(\theta)\right] \leq \frac{1}{\beta} \mathbb{E}_{\theta \sim P}\left[\hat{\mathcal{L}}(\theta)\right] + \alpha_{T,n}\left(\mathrm{KL}(P||Q) + \log\frac{2}{\delta}\right) + \frac{\log t_m}{\beta T \lambda} + \gamma \frac{\lambda K}{\beta T},$$

*in which $T$ is the number of observed transitions, $\alpha_{T,n} = \left(\frac{U}{2\beta(1-\beta)n} + \frac{1}{\beta T \lambda}\right)$, and $\gamma = e - 2$.*

Similarly to Theorem 5.2, Theorem 5.4 implies that obtaining tighter generalization guarantees is harder for larger state spaces with longer trajectories when the sampling process is limited by a maximum number of observable transitions (or states) $T$, which is an often imposed constraint for comparing the sample-efficiency of different learning objectives for GFlowNets in the literature (Pan et al., 2023b;a; Madan et al., 2022; Malkin et al., 2022; 2023). In the next section, we show how these issues can be addressed via a distributed learning scheme with network-level parallelization.

# 6 DIVIDE AND CONQUER: DISTRIBUTED LEARNING OF GFLOWNETS

In light of the above analysis, the diverse exploration of state graphs (Section 4) with smaller trajectory sizes (Section 5) is beneficial for the successful training of GFlowNets. In what follows, we show how these features can be efficiently implemented by recasting the GFlowNet training as an embarrassingly parallel divide-and-conquer algorithm, which we call *subgraph asynchronous learning* (SAL). This is, to the best of our knowledge, the first method enabling the distributed learning of GFlowNets with network-level parallelization. We remark that previous work on the topic (da Silva et al., 2024) promoted only the partitioning of the reward function for parallel Bayesian inference and that, in stark contrast to SAL, each client learned from the same state graph.

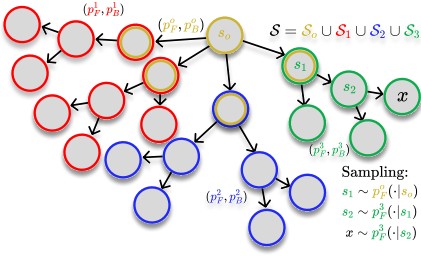

Figure 3: A fixed-horizon DAG partition with three leaves ($\mathcal{S}_1$, $\mathcal{S}_2$, $\mathcal{S}_3$) and one root ($\mathcal{S}_o$). For inference, the sampling policy is chosen based on the current state.

**Algorithm 1** Subgraph Asynchronous Learning

1: $\mathcal{S} = \mathcal{S}_o \cup \bigcup_{j=1}^{m} \mathcal{S}_j$ ▷ Fixed-horizon partition
2: $\mathcal{I}_j = \mathcal{S}_j \cap \mathcal{S}_o$ for $j \in \{1, \ldots, m\}$
3: ▷ Local training
4: **parfor** $j \in \{1, \ldots, m\}$ **do**
5:     ▷ Minimize $\mathcal{L}^j_{\text{ATB}}$ in $\mathcal{S}_j$ with SGD
6:     $(p^j_F, F_j) = \arg\min_{p_F, F} \mathcal{L}^j_{\text{ATB}}(p_F, F)$
7: **end parfor**
8: $R^o \colon x \mapsto \mathbf{1}_{\{x \in \mathcal{X}\}} R(x) + \sum_{j=1}^{m} \mathbf{1}_{\{x \in \mathcal{I}_j\}} F_j(x)$
9: $(p^o_F, F_o) = \arg\min_{p_F, F} \mathcal{L}_{\text{TB}}(p_F, F, R^o)$
10: **return** $\{(p^o_F, F_o)\} \cup \bigcup_{1 \le j \le m} \{(p^j_F, F_j)\}$

## 6.1 SUBGRAPH ASYNCHRONOUS LEARNING

**Overview.** There are two ingredients making up SAL: a fixed-horizon partition (FHP) and an assignment function (AF). In short, a FHP defines the state graph split explicitly, while an AF indirectly encodes it by assigning states to partitions. Here, we formally define the former concept. We introduce the idea of an AF and provide a comprehensive theoretical analysis of SAL in Appendix B.4.

**Convergence guarantees.** We introduce the notion of a FHP of a pointed DAG below. In the flow network perspective, such a partition can be viewed as a collection of possibly overlapping multi-source subnetworks, termed *leaves*, which are grouped together by a single-source network, referred to as *root*. We use the term *fixed-horizon* due to the fixed distance of the subnetworks' sources to $s_o$. Also, note that a FHP is only a *partition* in the set-theoretical sense when the state graph is a tree.

**Definition 6.1** (Fixed-horizon DAG partition). We say that $\mathcal{S} = \mathcal{S}_o \cup \left( \bigcup_{1 \le j \le m} \mathcal{S}_j \right)$ is a FHP of the state space $\mathcal{S}$, with *leaves* $\{S_j\}_{j=1}^m$ and *root* $\mathcal{S}_o$, when it satisfies the conditions below:

1. (Disjointness of sources) $s_o \in \mathcal{S}_o$ and the sets $\{\mathcal{I}_j \coloneqq \mathcal{S}_o \cap \mathcal{S}_j\}_{j=1}^m$ are pairwise disjoint.

2. (Completeness) If $s \in \mathcal{S}_j$ for a $j \ge 1$, then all descendants of $s$ are in $\mathcal{S}_j$.

3. (Regularity) If $d$ denotes the shortest-path distance, $d(s_o, \mathcal{I}_j) = d(s_o, \mathcal{I}_i)$ for all $i, j$.

Under Definition 6.1, we let $\mathcal{X}_j = \mathcal{S}_j \cap \mathcal{X}$ be the set of terminal states reachable from $\mathcal{I}_j$. For conciseness, we denote $\{\mathcal{S}_j\}_{j=0}^m = \text{FHP}(\mathcal{S}, m)$ when $\{\mathcal{S}_j\}_{j=0}^m$ is a FHP of $\mathcal{S}$ with $m$ components. We then illustrate a FHP of a tree in which $m = 3$ and the $\mathcal{I}_j$, represented by the doubly-stroked circles, are singletons for the blue and green leaves in Figure 3. We are now ready to define SAL.

**Definition 6.2** (SAL). Let $\{\mathcal{S}_j\}_{j=0}^m = \text{FHP}(\mathcal{S}, m)$. For each $1 \le j \le m$, let $\mathcal{G}_j = (p^j_F, p^j_B, F_j)$ be a GFlowNet and $p^j_E$ be any forward policy over the $\mathcal{S}_j$-induced subgraph of the state graph. Finally, let $q_j$ be any distribution with full support on $\mathcal{I}_j$. Then, define

$$\mathcal{L}^j_{\text{ATB}}(p^j_F, F_j) = \mathbb{E}_{s \sim q_j} \mathbb{E}_{\tau \sim p^j_E(\cdot|s)} \left[ \left( \log \frac{F_j(s) p^j_F(\tau|s)}{R(x) p^j_B(\tau|x)} \right)^2 \right], \tag{9}$$

in which $x$ represents $\tau$'s terminal state, as the *amortized trajectory balance* (ATB) objective. For the root, let $p^o_E$ be a policy in $\mathcal{S}_o$ and $\mathcal{G}_o = (p^o_F, p^o_B, R^o)$. For each $x \in (\mathcal{X} \setminus \bigcup_{j=1}^m \mathcal{X}_j) \cup (\bigcup_{j=1}^m \mathcal{I}_j)$, let $R^o(x) = R_j(x)$ if $x \in \mathcal{I}_j$ for some $j$ and $R^o(x) = R(x)$ otherwise. In this context, SAL follows a two-step procedure: first, $m$ models are trained in parallel by minimizing Equation 9; then, a global model is estimated by optimizing the TB loss with reward $R^o$, which we denote by $\mathcal{L}_{TB}(p_F, F, R^o)$ for a GFlowNet $(p_F, p_B, F)$. We summarize this approach in Algorithm 1.

Clearly, any flow-based learning objectives (e.g., SubTB (Madan et al., 2022), Munchausen DQN (Tiapkin et al., 2024), GAFlowNets (Pan et al., 2023b)), parametrizations (e.g., forward-looking (Pan et al., 2023a), LED (Jang et al., 2024), temperature-scaled (Kim et al., 2024a)), and off-policy sampling strategies (e.g., replay buffer (Vemgal et al., 2023) and local search (Kim et al., 2024b)) could be employed for estimating both the root and leaf GFlowNets in Definition 6.2. In

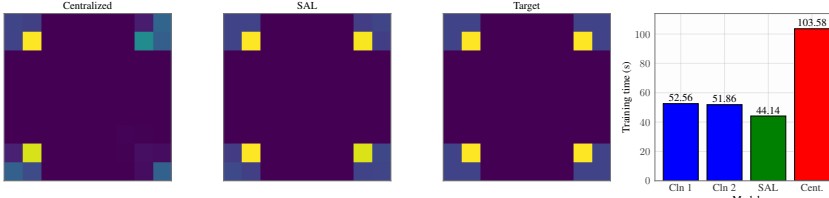

Figure 4: **SAL leads to faster mode discovery** and more accurate approximations given a fixed training time budget (right-most plot). Results for a centralized GFlowNet, for our algorithm (SAL), and the target reproduced from Malkin et al. (2022, Section 5.1) are shown from left to right. Running time for SAL equals the running time of the longest client plus that of the aggregation phase.

Appendix C, we demonstrate the soundness of SAL and conduct an extensive theoretical analysis on the character of the local distributions and error propagation within this framework. Similarly, Appendix E.1 discusses these issues from an empirical viewpoint.

**Recursive SAL.** As defined in Definition 6.2 and shown in Figure 3, SAL has a *single layer*: each leaf is directly connected to the root in the underlying FHP. Nonetheless, there is no obstacle preventing us from building SAL upon a multi-layered partition of the state graph, as illustrated in Figure 11 in the supplement. For this, we must first define a hierarchy of partitions. Then, we recursively learn a flow assignment for each partition by starting at the lowest levels of this hierarchy and moving upwards, in the fashion of backward-induction algorithms. Each learning step is based on minimizing the amortized trajectory balance loss in Equation 9 via SGD. We demonstrate the correctness of the resulting algorithm, termed Recursive SAL, in Proposition C.6, which follows from Theorem C.1 and an inductive argument. Although we do not provide an empirical assessment of Recursive SAL in this work, Appendix C.3 considers its potential implications.

## 6.2 EMPIRICAL ILLUSTRATION

**Experimental setup.** We evaluate the performance of SAL in six different generative tasks encompassing both synthetic and real-world problems (Appendix C.1). In Appendix C, we extensively discuss how to implicitly define a FHP via an assignment function, which allows for an efficient implementation of SAL. Also, please refer to Appendix B for a detailed account of the experimental setup.

**Results.** As expected, Figure 4 above, and Figures 7, 9, 10, and 13 in the supplement, show that SAL drastically speeds up the discovery of high-value states for all considered generative problems under varying computational constraints. Complementarily, Figure 6 and Table 2 in Appendix C.2 underline that our distributed algorithm achieves more accurate distributional approximations than its centralized counterpart. We discuss these promising results in more detail in Appendix C.2.

## 7 CONCLUSIONS

**Discussion.** We developed the first PAC-Bayesian bounds and non-vacuous statistical guarantees for the generalization of GFlowNets in the literature. Additionally, our theoretical results provided deeper insights into the negative effect of the trajectory length on the proven learnability of a generalizable policy. Inspired by these conclusions, our distributed algorithm SAL, which is also the first of its kind, exhibited promising performance in both synthetic and real-world problems.

**Future works and limitations.** We discuss the limitations of our work at large in Appendix E. In particular, we acknowledge that a deeper theoretical understanding of advanced sampling techniques is still required. From a practitioner's perspective, we believe that SAL can greatly improve the performance of GFlowNets in specialized domains, e.g., NLP (Hu et al., 2024) and drug discovery (Bengio et al., 2021), which are beyond the scope of our work. Finally, it has not escaped our attention that SAL is related to Mankowitz et al. (2016)'s Adaptive Skills, Adaptive Partitions (ASAP) framework for learning temporally extended actions in MDPs and may find fruitful applications in multi-task RL by interpreting each leaf (resp. root) GFlowNet as an intra- (resp. inter-) skill policy.

## ACKNOWLEDGEMENTS

This work was supported by the Fundação Carlos Chagas Filho de Amparo à Pesquisa do Estado do Rio de Janeiro FAPERJ (SEI-260003/000709/2023), the São Paulo Research Foundation FAPESP (2023/00815-6), and the Conselho Nacional de Desenvolvimento Científico e Tecnológico CNPq (404336/2023-0). We acknowledge the UKRI EPSRC grant EP/Y028783/1.

We acknowledge the Aalto Science-IT Project from Computer Science IT and the FGV TIC for providing computational resources.

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

# Supplementary Material for "Generalization and Distributed Learning of GFlowNets"

## A BACKGROUND AND RELATED WORKS

For probability measures $P$ and $Q$ on the same space $\mathcal{X}$, we recall for convenience that the Kullback-Leibler (KL) divergence is $\mathrm{KL}(P||Q) = \mathbb{E}_{x\sim P}\left[\log\left(\mathrm{d}P/\mathrm{d}Q\right)(x)\right]$, the chi-squared divergence is $\chi^2(Q||P) = \mathbb{E}_{x\sim P}\left[\left((\mathrm{d}Q/\mathrm{d}P)(x)\right)^2 - 1\right]$; and the total variation distance is given by $\mathrm{TV}(P,Q) = \sup_{A\subseteq\mathcal{X}}|P(A) - Q(A)|$. When $\mathcal{X}$ is finite, $\mathrm{TV}(P,Q) = 1/2\sum_{x\in\mathcal{X}}|P(x) - Q(x)|$. There are other notions of 'divergence' for probability measures; we have mentioned here the ones used in our paper. Readers are refereed to Boucheron et al. (2013) for further details on the topic.

### A.1 DIRECTED ACYCLIC GRAPHS

We briefly recall the definition of a *pointed directed acycliy graph*. For this, we firstly define the concept of a *finitely absorbing Markov transition kernel* (Lahlou at al., 2023) in a topological space. Henceforth, we let $(\{s_o\}\cup\mathcal{S}\cup\{s_f\},\mathcal{V})$ be a topological space endowed with a topology $\mathcal{V}$ and two special elements, $s_o$ and $s_f$. We denote $\bar{\mathcal{S}} = \{s_o\}\cup\mathcal{S}\cup\{s_f\}$ the *state space*; $s_o$ and $s_f$ are the *initial* and *final* states, respectively. We also assume that both $\{s_o\}$ and $\{s_f\}$ are open sets with respect to $\mathcal{V}$.

**Definition A.1** (Finitely absorbing Markov transition kernel (MTK)). Consider the measure space $(\bar{\mathcal{S}}, \Sigma, \mu)$ with measure $\mu$ and a Borel $\sigma$-algerba $\Sigma$. Let $\kappa\colon \bar{\mathcal{S}}\times\Sigma\to\mathbb{R}_+$ be a *reference kernel*, i.e., $\kappa(s,\cdot)\colon \Sigma\to\mathbb{R}_+$ is a measure absolutely continuous with respect to $\mu$ for all $s$, and we recursively define $\kappa^{\otimes t}(s, A) = \int \kappa^{\otimes t-1}(s, \mathrm{d}s')\kappa(s', A)$ for measurable $A\in\Sigma$. We say that $\rho_F\colon \bar{\mathcal{S}}\times\Sigma\to\mathbb{R}_+$ is a *finitely absorbing Markov transition kernel* if the following conditions are satisfied.

1. $\rho_F(s,\cdot)\colon \Sigma\to\mathbb{R}_+$ is an absolutely continuous probability measure with respect to $\kappa(s,\cdot)$;

2. there is a $t_m < \infty$ such that $\rho_F^{\otimes t_m}(s, \{s_f\}) = 1$ for every $s\in\{s_o\}\cup\mathcal{S}$ and $\rho_F(s_f, \{s_f\}) = 1$;

3. $s\mapsto\rho_F(\cdot, B)$ is continuous for every measurable $B\in\Sigma$;

4. if $\rho_F(s, \{s_f\}) > 0$, then $\rho_F(s, \{s_f\}) = 1$;

5. for every $A\in\Sigma$, there is a $t < t_m$ such that $\rho_F^{\otimes t}(s_o, A) > 0$.

In this work, $\bar{\mathcal{S}}$ is always finite, $\mathcal{V}$ is the discrete topology, and $\mu$ is the counting measure. The state graph $G$ is induced by $\rho_F$, i.e., $(u,v)$ is an edge in $G$ if and only if $\rho_F(u, \{v\}) > 0$. Acyclicity is ensured by the finitely absorbing property of $\rho_F$ (item 2 of Definition A.1). Notably, the finite $\bar{\mathcal{S}}$ assumption covers the vast majority of use-cases for GFlowNets. Under these conditions, we say $\kappa^\top$ is a *backward reference kernel* in $\bar{\mathcal{S}}$ with respect to $\kappa$ if $\kappa(u, \{v\}) = \kappa^\top(v, \{u\})$ for all $(u,v)\in\bar{\mathcal{S}}\times\bar{\mathcal{S}}$. We refer the reader to (Lahlou at al., 2023) for an overview of GFlowNets in infinite spaces.

### A.2 GENERATIVE FLOW NETWORKS

A GFlowNet can be seen as a tuple $(\{s_o\}\cup\mathcal{S}\cup\{s_f\}, P_F, P_B, \rho_F, \rho_B, \kappa, \kappa^\top, \mu, R)$ for which

1. $\kappa$ is a *forward reference kernel* on $\bar{\mathcal{S}}$;
2. $\kappa^\top$ is a *backward reference kernel* in $\bar{\mathcal{S}}$ with respect to $\kappa$;
3. $\rho_F$ (resp. $\rho_B$) is a finitely aborsbing MTK with respect to $\kappa$ (resp. $\kappa^\top$);
4. $R\colon \Sigma\to\mathbb{R}_+$ is a measure such that $R\ll\mu$;
5. $P_F\colon \bar{\mathcal{S}}\times\Sigma\to\mathbb{R}_+$ is a MTK, called the *forward policy*, such that $P_F(s,\cdot)\ll\rho_F(s,\cdot)$;
6. $P_B\colon \bar{\mathcal{S}}\times\Sigma\to\mathbb{R}_+$ is a MTK, called the *backward policy*, such that $P_B(s,\cdot)\ll\rho_B(s,\cdot)$.

We denote by $p_F$ and $p_B$ the densities of $P_F$ and $P_B$ with respect to their respective reference kernels. For simplicity, we interchangeably let $R(x)$ be the density of $R$ with respect to $\mu$. In this scenario, the set of terminal states $\mathcal{X}$ is defined by $\mathcal{X} = \{x\in\mathcal{S}\colon P_F(s, \{s_f\}) > 0\}$. In practice, $p_F$ is parameterized by a neural network and its parameters are estimated to ensure that the marginal of $P_F(s_o, \cdot)$ over $\mathcal{X}$ matches $R$ up to a normalizing constant. In the terminology of Section 2, the abstract actions space $\mathcal{A}$ would correspond to $\mathcal{A} = \bigcup_{s\in\bar{\mathcal{S}}}\{(s, u)\colon P_F(s, \{u\}) > 0\}$ and $A(s) = \{(s, u)\colon P_F(s, \{u\}) > 0\}$. For most problems, we identify the edge $(s, u)$ with an entity representing the difference between $u$ and $s$, e.g., a nucleotide base when $\mathcal{S}$ is the space of nucleotide strings. We complement the discussion in Section 2 on how to learn a GFlowNet in the next section.

### A.3 LEARNING GFLOWNETS

Below, we illustrate our definition of GFlowNets for three common generative tasks. These tasks encompass a large number of applications, e.g., Jain et al. (2022); Shen et al. (2023); Hu et al. (2024; 2023); Liu et al. (2023); Malkin et al. (2022); Pan et al. (2023b); Madan et al. (2022).

1. **Autoregressive generation**. Each object in $\mathcal{S}$ is a string of length up to a $L$, and $G$ is a tree rooted at $s_o$. Also, action sets $A(s)$ represent an alphabet and a transition $\mathcal{T}(s, a)$ appends the character $a$ to the string $s$. Here, $p_B(s|(s, a)) = 1$ for every $s \in \mathcal{S}$ and $a \in A(s)$.

2. **Set generation**. Each $s \in \mathcal{S}$ is a subset of $\mathcal{W} = \{1, \dots, W\}$, with $\mathcal{X}$ containing those $s$ of size $T$. Action sets are $A(s) = \mathcal{W} \setminus s$ and transitions $\mathcal{T}(s, a)$ add the element $a$ to $s$; see Figure 5.

3. **Hypergrid environment**. Each $s \in \mathcal{S}$ is a point within $\{0, \dots, H-1\}^d \times \{0, 1\}$ for given $H$ (size) and $d$ (dimension); $s_o = \mathbf{0}_{d+1}$ and the last coordinate indicates whether $s \in \mathcal{X}$. Also, $A(s) = \{e_i : s^i < H - 1\} \cup \{\top\}$, with $e_i$ denoting the $i$-th canonical vector in $\mathbb{R}^d$ and $\top$ a stop action. Transitions $\mathcal{T}(s, a)$ either add $a$ to $s$, if $a = e_i$ for some $i$; or set $s^{d+1} = 1$, if $a = \top$.

Notably, $\mathcal{T}(x, \cdot) \in \{s_f\}$ for every terminal state $x \in \mathcal{X}$. We provided examples of $R$, $F$, and $p_F$ throughout the main text; in particular, see Sections 4 and 6.2 and Appendix B. Figure 5 illustrates the state graph for the set generation task (omitting $s_f$). To learn a forward policy $p_F$, we minimize a stochastic objective based on the observed trajectories. Besides the ones shown in Equation 1, many loss functions has been recently proposed. The SubTB loss (Madan et al., 2022), for instance, is defined by

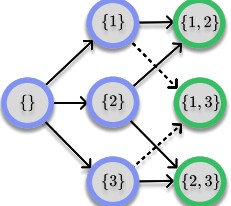

Figure 5: State graph for the set generation task ($W = 3$, $T = 2$).

$$\mathbb{E}_{\tau \sim p_E} \left[ \sum_{1 \leq n < m \leq |\tau|} \frac{\lambda^{m-n}}{\sum_{1 \leq n < m \leq |\tau|} \lambda^{m-n}} \left( \log \frac{F(\tau_n) p_F(\tau_{n:m})}{p_B(\tau_{n:m}) F(\tau_m)} \right)^2 \right]$$

with the constraint that $F(x) = R(x)$ for $x \in \mathcal{X}$ and $\tau_n$ representing the $n$th element within the trajectory $\tau$. Correspondingly, the VarGrad (Zhang et al., 2023a) and contrastive balance (da Silva et al., 2024) objectives avoid the estimation of $F$ by minimizing

$$\mathbb{E}_{\tau, \tau' \sim p_E} \left[ (\log p_F(\tau) - \log p_F(\tau') - \log p_B(\tau|x) + \log p_B(\tau'|x') - \log R(x) + \log R(x'))^2 \right],$$

which led to faster training convergence in some cases. On the same page, Malkin et al. (2023) considered a series of divergence-based loss functions for training GFlowNets, showing that the on-policy version of the TB loss (Equation 1) corresponds to the reverse KL between the forward and backward policies in terms of the gradients. In particular, we note that these learning objectives can only be used for estimating the parameters of the root network in SAL. For the leaf networks, which must provide an estimate of the flow function $F$ for the aggregation step, these loss functions cannot be used. Nonetheless, flow-based learning objectives such as TB and SubTB often exhibit a convergence speed comparable to that of variational alternatives and are frequently implemented for large-scale applications (Nica et al., 2022; Jain et al., 2022; Hu et al., 2024; Zhou et al., 2024). Learning objectives aside, there is a growing interest in the literature in the development of more effective parametrizations for GFlowNets, with remarkable results for the forward-looking GFlowNets (Pan et al., 2023a) and LED-GFlowNets (Jang et al., 2024), which residually reparameterize $F$ as $\log F(s) = \log \phi(s) + \log \tilde{F}(s)$ for a (given or learnable) $\phi$, temperature-scaled-GFlowNets (Kim et al., 2024a), in which $p_F(s'|s) \propto \exp\{\phi(\beta) \cdot \psi(s'|s)\}$ for neural networks $\phi$ and $\psi$ and an inverse-temperature parameter $\beta > 0$, and QGFN (Lau et al., 2023), which learns a Q-function concomitantly to $F$ and $p_F$ and prune the values of $p_F$ based on $Q$ during inference time for controlable greediness.

### A.4 RELATED WORKS

GFlowNets (Bengio et al., 2021; 2023; Lahlou at al., 2023) were canonically proposed as a reinforcement learning algorithm for sampling compositional objects (e.g., graphs) proportionally to a prespecified reward function. From a theoretical perspective, the relationship between GFlowNets and variational inference (Malkin et al., 2023), entropy-regularized Q-learning (Tiapkin et al., 2024; Deleu et al., 2024), and diffusion models (Lahlou at al., 2023; Sendera et al., 2024; Venkatraman et al., 2024) has been thoroughly established. From a practitioner's viewpoint, GFlowNets have

been successfuly applied to many problems including, but not restricted to, causal discovery (Deleu et al., 2022; 2023; da Silva et al., 2023), Bayesian phylogenetic inference (Zhou et al., 2024; da Silva et al., 2024), language and image modelling (Hu et al., 2024; Liu et al., 2023; Hu et al., 2023; Venkatraman et al., 2024), combinatorial optimization (Zhang et al., 2023a;b), and drug discovery (Bengio et al., 2021; Nica et al., 2022; Vemgal et al., 2023; Pan et al., 2023a). Indeed, we are confident that problems such as language modelling and drug discovery could greatly benefit from SAL if appropriate policy networks and fixed-horizon partitionings are designed. Nonetheless, given the open-endedness and specialized nature of these applications, we believe that they would be more suited for future, dedicated works and are, hence, not addressed in this text. Correspondingly, recent work by Jiralerspong et al. (2024) highlighted the competitive performance of stochastic GFlowNets in two-player zero-sum games, specifically, Tic-Tac-Toe and Connect-4, and we are optimistic that an extension of SAL to stochastic environments would exhibit promising results for games having larger trajectories, e.g., Chess and Go. Orthogonal to these advances, the issue of generalization in GFlowNets has also received significant attention in the literature (Atanackovic & Bengio, 2024; Krichel et al., 2024). In sharp contrast to previous works, ours is the first one that derives PAC-Bayesian bounds and provides non-vacuous statistical guarantees for GFlowNets, along with a theoretical analysis that highlights which factors are potentially harmful to the model's generalization performance. Notably, a recent discussion by Bengio & Malkin (2024) provides an interesting perspective on generalization, active learning, and GFlowNets in the context of abstract reasoning for machine-learning-based theorem proving and conjecture formation. Concomitantly, we note that there is a well-established interest in the community towards the development of more sample-efficient learning objectives for speeding up training convergence (Malkin et al., 2022; Deleu et al., 2022; Madan et al., 2022; Zhang et al., 2023a; da Silva et al., 2024; Tiapkin et al., 2024).

## A.5    ADDITIONAL REVIEW OF PAC-BAYES BOUNDS

Historically, McAllester (1998; 1999)'s PAC-Bayesian theorems, which were inspired by the work of Shawe-Taylor & Williamson (1997), were developed towards the objective of providing Probably Approximately Correct (PAC) guarantees to Bayesian algorithms with potentially misspecified prior distributions. Recently, the relationship between PAC-Bayesian theory and (approximate) Bayesian algorithms has been made explicit by Germain et al. (2016). From this perspective, Alquier (2024) provides an informative and comprehensive account of the literature on PAC-Bayes bounds, both theory and applications. In what are now well-established references, Catoni (2007) gives a rigorous foundation of PAC-Bayes bounds in supervised classification, including results on the form of the distributions that optimise the bounds; and Guedj (2019) provides a nice concise exposition of the essential form of PAC-Bayesian inequalities. In the context of contemporary machine learning, PAC-Bayesian theory has found enormous success in the development of numerical generalization bounds for overparameterized neural network classifiers, achieving non-vacuous results (Dziugaite & Roy, 2017; 2018) and tight certificates (Pérez-Ortiz et al., 2021), which subsequent works have applied even for large language models (Lotfi et al., 2024a;b) with billions of parameters through appropriate compression techniques (Dettmers et al., 2023). In a recent work, Malach (2024) introduced the notion of *length complexity* for next-token autoregressive learning on Chain-of-Thought data, referring to the minimum number of iterations required by an AR learner to compute a target function, which is (vaguely) connected to our results regarding the harmful effects of the maximum trajectory size on GFlowNet learning. Importantly, the advantageousness of distributed approaches for the generalization performance of learning algorithms was already pointed out by Yagli et al. (2020); Barnes et al. (2022); Sefidgaran et al. (2022); similarly to SAL, these authors consider the problem of training a set of models in parallel and subsequently aggregating them with a (possibly randomized) estimator in a central server. In spite of these advances, the development of tighter PAC-Bayes bounds with weaker assumptions on the risk functional, e.g., heavy tailedness instead of boundedness, is still an active research field (Holland, 2019; Wu et al., 2021; Balsubramani, 2015; London et al., 2014; Biggs & Guedj, 2023; Rivasplata et al., 2020; Rodríguez-Gálvez et al., 2024a;b). Also, the development of PAC-Bayesian theory in the setting of non-i.i.d. data is still relatively underdeveloped when compared against other branches of machine learning, albeit there are interesting results in online learning (Haddouche & Guedj, 2022), reinforcement learning (Fard & Pineau, 2010; Beygelzimer et al., 2011; Sakhi et al., 2023), and time series (Alquier et al., 2012). Finally, PAC-Bayesian theorems provide statistical guarantees for stochastic predictors, which are arguably not frequently used in practice, and the problem of derandomizing the resulting bounds

is still mostly open. Notably, the derandomization of PAC-Bayes bounds has a non-negligible cost, and we refer the reader to Miyaguchi (2019); Biggs & Guedj (2022) for further details on this topic.

# B    EXPERIMENTAL DETAILS AND ADDITIONAL DISCUSSIONS

All experiments were conducted on a single Linux machine with 128 GB of RAM and featuring a NVIDIA RTX 3090 GPU and 12th Gen Intel(R) Core(TM) i9-12900K CPU. Unless specified otherwise, the code for reproducing the experiments below was executed on this GPU.

## B.1    A NON-GENERALIZABLE DISTRIBUTION

For the experiments in Figure 1, we considered the set generation task (see Appendix A) with $W \in \{32, 64\}$ elements to choose from and set size $S = 6$, and the forward policy was parameterized by an MLP with 2 64-dimensional layers. The elements' log-utilities $u$ were sampled from $[-1, 1]$

| $(W, T)$ | $\chi^2(q_{E,T} \| p_{T,E})$ | $\chi^2(q_{\epsilon,T} \| p_{T,E})$ |
|---|---|---|
| $(32, 6)$ | $1.20 \cdot 10^3$ | $1.32$ |
| $(64, 6)$ | $4.56 \cdot 10^2$ | $1.24$ |

Table 1: $\chi^2$ divergence between the exploratory (pruned, $p_{E,T}$, and $\epsilon$-greedy, $p_{\epsilon,T}$) and uniform ($q_{E,T}$) distributions.

prior to training and the resulting values were normalized so that the largest reward of a set was 5. For both settings in Figure 1, the models were trained for 1500 epochs with a batch size of 128. To compute the quantities in Table 1, we compared the uniform policy of an untrained GFlowNet against a policy $p_E$ such that the (unnormalized) logit corresponding to the addition the element 1 is set to $\log p_E(1|s) = -11.5 \approx \log 10^{-5}$. Table 1 shows the large discrepancy between the resulting $p_E$ and an uniform policy, providing a taste for the upper bound in Proposition 4.2.

## B.2    NON-VACUOUS GENERALIZATION BOUNDS

**A bounded risk functional for GFlowNets.** We start recalling the definition of the flow-consistency in subgraphs (FCS) metric (Silva et al., 2024). Given a policy $p_E$, the FCS is defined as

$$L_{\text{FCS}}(R, p_T) = \mathbb{E}_{\tau_1, \dots, \tau_B \sim p_E} \left[ \frac{1}{2} \sum_{1 \leq i \leq B} \left| \frac{p_T(x_i)}{\sum_{1 \leq j \leq B} p_T(x_j)} - \frac{R(x_i)}{\sum_{1 \leq j \leq B} R(x_j)} \right| \right], \qquad (10)$$

in which $B \geq 2$ is a (typically small) given integer. Equivalently, FCS may be seen the expected total variation distance between the learned $p_T$ and target $R$ distributions over random subsets of $\mathcal{X}$. It was shown by Silva et al. (2024) that $L_{\text{FCS}}(R, p_T) = 0$ if and only if $p_T(x) \propto R(x)$, i.e., the model samples correctly from the distribution proportional to $R$ in $\mathcal{X}$. Then, equipped with the dataset $\mathcal{T}_n$ described in Section 4, an unbiased estimate of $L_{\text{FCS}}$ is

$$\hat{L}_{\text{FCS}}(\mathcal{T}_n, R, p_T) = \frac{1}{2N} \sum_{k_1, \dots, k_B \sim U\{1, \dots, n\}} \sum_{1 \leq i \leq B} \left| \frac{p_T(x_{k_i})}{\sum_{j=1}^{B} p_T(x_{k_j})} - \frac{R(x_{k_i})}{\sum_{j=1}^{B} R(x_{k_j})} \right|, \qquad (11)$$

in which the outer summation covers $N$ uniformly random $B$-sized subsets of $\{1, \dots, n\}$ and $x_{k_i}$ represents the $k_i$th observed terminal state in $\mathcal{T}_n$. Importantly, $L_{\text{FCS}} \in [0, 1]$ for any $R$ and $p_T$, which enables the implementation of well-known algorithms for tightening PAC-Bayesian generalization bounds through the adoption of data-dependent priors (Dziugaite et al., 2021; Maurer, 2004).

**Experimental details for computing non-vacuous bounds.** To achieve the results illustrated in Figure 2, we use $\mathcal{T}_\alpha$ to learn an isotropic Gaussian prior $Q$ with variance $10^{-6}$ over the parameters $\theta$ of an MLP with $3 \times 128$-dimensional layers defining the forward policy by minimizing the expected TB loss on $\mathcal{T}_\alpha$ under $Q$. For each problem, we used the same architecture of the neural network, changing only the input and output dimensions, and the resulting models were trained for 64 epochs on their respective datasets. Then, we freeze $\theta$ and learn both the mean and the diagonal covariance of a Gaussian posterior $P$ over the parameters of a policy network by minimizing the upper bound in Equation 6 with $\hat{L}_{\text{FCS}}$ substituted by an unbiased estimate of the TB loss on $\mathcal{T}_\alpha \cup \mathcal{T}_{1-\alpha}$. Finally, we evaluate the upper bound in Equation 6 on $\mathcal{T}_{1-\alpha}$ to certify its tightness. We closely followed the experimental setup of Dziugaite et al. (2021); Pérez-Ortiz et al. (2021) for conducting these experiments. In particular, the data-splitting protocol for learning the prior, learning the posterior, and evaluating the bound is analogous to the one used by Pérez-Ortiz et al. (2021). Similarly, in contrast to the other experiments, which rely on the Adam optimizer (Kingma & Ba, 2015), we use SGD with a fixed learning rate of $10^{-3}$ that presumably achieves a flat minimum (Keskar et al., 2017) with potentially better generalization properties (Hochreiter & Schmidhuber, 1997; Zhou et al., 2020; Haddouche et al., 2024). Finally, we acknowledge Dziugaite et al. (2021) for making their code publicly available and adhering to the best current practices of scientific reproducibility.

### B.3 ORACLE GENERALIZATION BOUNDS: LEMMATA

**Trajectory-level bounds.** The technical lemma below ensures that $\mathrm{KL}(p_B||p_F)$ can be directly bounded by adopting a mixture transition policy, sometimes called an *$\alpha$-uniform policy* (Hu et al., 2023), that keeps the trajectory-level probabilities away from zero and ensures the boundedness of the log-probabilities (Dziugaite et al., 2021; Lotfi et al., 2024a) without limiting the GFlowNet's ability to learn the correct solution that samples from $\mathcal{X}$ proportionally to the reward.

**Lemma B.1** (Realizability of mixture policies). *Let $p_U(\cdot|s)$ denote the uniform policy on the state space $\mathcal{S}$ with reward $R$, i.e., $p_U(s'|s) = \frac{1}{|\mathrm{Ch}(s)|}$. Then, there is a $\alpha \in (0,1]$ s.t. the family $\{\tilde{p}_F : \tilde{p}_F(\cdot|s) = \alpha p_U(\cdot|s) + (1-\alpha)p_F(\cdot|s)\}$ contains a policy sampling from $\mathcal{X}$ in proportion to $R$.*

In the classical statistical learning terminology, the result above states that the family of $\alpha$-uniform policy networks is *realizable*, meaning that a member of this family satisfies the desired balance conditions. However, as we note in the proof of Lemma B.1, finding such $\alpha$ depends on the knowledge of the minimum value of $R(x)$ on $\mathcal{X}$, which may be an NP-hard problem for some generative instances (Zhang et al., 2023b; Ma et al., 2013) that cannot be swiftly solved. Since the resulting generalization bound depends on hardly computable quantities, we call it a *oracle* bound, similarly to the distribution-dependent PAC-Bayesian inequalities in, e.g., (Alquier et al., 2012; Alquier, 2024).

**Transition-level bounds.** From Definition 5.3, we can readily conclude that the stochastic process

$$M_t := \sum_{1 \leq i \leq t} M(S_i, S_{<i}) = \sum_{1 \leq i \leq t} \mathcal{L}_{\mathrm{DB}}(S_i, S_{i-1}) - \mathbb{E}_{s_i \sim p_E(\cdot|S_{i-1})}[\mathcal{L}_{\mathrm{DB}}(s_i, S_{i-1})] \qquad (12)$$

is a martingale with respect to the filtration $\{\mathcal{F}_t\}_{t \geq 1}$. In Theorem 5.4, we developed concentration inequalities for $M_t$ to derive transition-level generalization bounds for GFlowNets. Complementarily, the lemma bellow shows how the martingale $M_t$ is connected to the traditionally implemented trajectory-wide DB loss (see Equation 1). There, we assume that trajectories have fixed length, an assumption that was also considered by Malkin et al. (2023) when showing that a GFlowNet can be seen as an instantiation of a hierarchical variational inference model.

**Lemma B.2.** *Let $p_E$ be the sampling distribution and $p_{E,T}$ be the corresponding marginal over terminal states. Then, by denoting $\tau = (S_1, \ldots, S_l)$ with fixed $l$,*

$$\mathbb{E}_{\tau \sim p_E}\left[\sum_{1 \leq i \leq l} \mathcal{L}_{\mathrm{DB}}(S_i, S_{<i})\right] = \sum_{1 \leq i \leq l} \mathbb{E}_{S_{i-1} \sim p_{E,T}}\left[\mathbb{E}_{S_i \sim p_E(\cdot|S_{i-1})}[\mathcal{L}_{\mathrm{DB}}(S_i, S_{i-1})|S_{i-1}]\right].$$
$$(13)$$

In other words, the trajectory-wise objective in the left-hand side of Equation 13, which is often used as a learning objective for GFlowNets (Pan et al., 2023a;b; Madan et al., 2022; Bengio et al., 2023; Jang et al., 2024; Silva et al., 2024), corresponds to the transition-wise objective in Equation 8 when the trajectories are sampled in a Markovian fashion. Under these circumstances, we defined the risk functional associated to a specific parameterization $\theta$ of the policy network as

$$\mathcal{L}(\theta) = \mathbb{E}_{\tau \sim p_E} \frac{1}{|\tau|} \sum_{1 \leq i \leq |\tau|} \mathbb{E}_{S_i \sim p_E(\cdot|S_{i-1})}[\mathcal{L}_{\mathrm{DB}}(S_i, S_{i-1})|S_{i-1}], \qquad (14)$$

in which $\mathcal{L}_{\mathrm{DB}}$ implicitly depends on $\theta$ via the forward policy $p_F$. Importantly, we take the trajectory's length $|\tau|$ into account when defining $\mathcal{L}(\theta)$, which is often done in practice (Zhang et al., 2023b). Then, given a set $\{s_o, S_1^{(j)}, \ldots, S_{t_j}^{(j)}\}_{j=1}^n$ of independently sampled trajectories, we define

$$\hat{\mathcal{L}}(\theta) = \frac{1}{n} \sum_{1 \leq j \leq n} \frac{1}{t_j} \sum_{1 \leq i \leq t_j} \mathcal{L}_{\mathrm{DB}}(S_i^{(j)}, S_{i-1}^{(j)}) \qquad (15)$$

as the empirical estimate of $\mathcal{L}(\theta)$. Under these conditions, Theorem 5.4 established a high-probability upper bound of $\mathcal{L}(\theta)$ as a function of $\hat{\mathcal{L}}(\theta)$ and of some characteristics of the generative process. We recall, however, that two assumptions were required to achieve this: that the DB loss and thus the martingale difference sequence $M$ are almost surely bounded and that the training is constrained by a pre-specified transition budget. In practice, the boundedness can be achieved by

either clipping the loss function (McAllester, 1999; 2013) or, when more detailed information about $R$ is available, constraining the output of the neural networks in the fashion of Lemma B.1 with the knowledge that the optimal flow $F^\star(s)$ satisfying the detailed balance is bounded by $F^\star(s) \in \left[\min_{x \in \mathcal{X}} R(x), \sum_{x \in \mathcal{X}} R(x)\right]$ for each state $s$ (Bengio et al., 2023), or a more refined version of this constraint (with upper and lower limits possibly depending on $s$). On the other hand, due to the CPU-bounded nature of GFlowNet transition sampling (which cannot be easily parallellized in a GPU), we assume that training is computationally limited by a fixed number of observed transitions. Hence, to promote an equitable assessment of different generative tasks, we assume in Theorem 5.4 that the number $n$ of sampled trajectories for training depends on a fixed budget of sampleable transitions $T$.

## B.4 Subgraph Asynchronous Learning

Please refer to Section C.2 for a detailed experimental evaluation of SAL. We would like to emphasize that all experiments below are based on standard practices for GFloNet training, with trajectories sampled from an $\epsilon$-greedy sampling policy, and *not* on the simplified setting of Section B.2.

## C SAL: IMPLEMENTATION AND THEORETICAL ANALYSIS

**Sampling correctness.** We first recall how to sample a $x \in \mathcal{X}$ in the context of SAL. For a given collection $\{p_F^o\} \cup \{p_F^j\}_{j=1}^m$ of forward policies trained in the style of Definition 6.2, we do so by starting at $s_o$ and following the root policy until we reach either a terminal state $x \in \mathcal{X}$ or a leaf partition $\mathcal{S}_j$. In the former case, we interrupt the generation and return $x$ as a sample. In the latter, we proceed to $\mathcal{X}$ by following the leaf's policy $p_F^j$, as shown in the highlighted trajectory in Figure 3. In Theorem C.1, we demonstrate that this approach samples $x \in \mathcal{X}$ proportionally to $R(x)$ when both the leaf and root policies globally minimize their respective learning objectives.

**Theorem C.1** (Sampling correctness of SAL). *Let* $\{\mathcal{S}_j\}_{j=0}^m = \mathrm{FHP}(\mathcal{S}, m)$ *and* $\{\mathcal{G}_j\}_{j=0}^m$ *be the corresponding GFlowNets. Let* $p_F^{\star,o}$ *and* $\{p_F^{\star,j}\}_{j=1}^m$ *be global minimizers of their respective learning objectives. Then, the marginal distribution over* $\mathcal{X}$ *induced by the learned policies* $\{p_F^{\star,j}\}_{j=0}^m$,

$$p_T^\star(x) = \sum_{1 \leq j \leq m} \sum_{s \in \mathcal{I}_j} \sum_{\tau:\, s_o \rightsquigarrow s} p_F^{\star,o}(\tau|s_o) \sum_{\tau':\, s \rightsquigarrow x} p_F^{\star,j}(\tau'|s), \tag{16}$$

*matches the target distribution* $\pi(x) := {R(x)}/{Z}$, *with* $Z = \sum_{x \in \mathcal{X}} R(x)$.

Remarkably, Theorem C.1 establishes SAL as the first asymptotically correct general-purpose distributed learning algorithm for GFlowNets. On the other hand, a successful implementation of SAL requires having an efficient mechanism concomitantly enabling to sample states from a given partition (to minimize $\mathcal{L}_{\mathrm{ATB}}$) and to recover the partition of a state (for inference). This is the reasoning behind what we name, and have long named, an *assignment function*. In Section C.2, we develop such mechanisms for some commonly considered generative tasks in the GFlowNet literature and provide an empirical analysis asserting the effectiveness of the resulting algorithm.

### C.1 SUMMARY OF THE EXPERIMENTAL CAMPAING

We provide a summary of the considered generative tasks for assessing the performance of SAL. For a comprehensive discussion on the implementation of SAL, please consult the sections below.

1. **Hypergrid** (Bengio et al., 2021; Malkin et al., 2022; 2023; Pan et al., 2023b; Krichel et al., 2024). We consider both a 8 x 8 and a 64 x 64 hypergrid environment (Section 2) with (Malkin et al., 2022, Section 5.1)'s reward function, which is illustrated in Figure 4 for $H = 8$.

2. **SIX6** (Jain et al., 2022; Malkin et al., 2022; Shen et al., 2023; Chen & Mauch, 2024; Kim et al., 2024a). We generate 8-sized nucleotide strings. The reward represents wet-lab DNA binding measurements to a human transcription factor (Barrera et al., 2016; Trabucco et al., 2022).

3. **PHO4** (Jain et al., 2022; Malkin et al., 2022; Shen et al., 2023; Chen et al., 2023). Similarly, we construct 10-sized nucleotide strings; the reward reflects wet-lab measurements of DNA binding activities to a yeast transcription factor (Barrera et al., 2016; Trabucco et al., 2022).

4. **Bit sequences** (Malkin et al., 2022; Madan et al., 2022; Rector-Brooks et al., 2023; Tiapkin et al., 2024). We produce 60-sized binary sequences. Given a subset $M$ of such sequences, we define $R(x) = \exp\{-\min_{m \in \mathcal{M}} d_L(x, m)\}$, in which $d_L$ is the edit distance.

5. **Sequence design** (Jain et al., 2022; da Silva et al., 2024). We build 8-sized sequences of $\{1, \ldots, 6\}$. Also, $R(x) = \sum_{i=1}^8 g(i) f(x_i)$, with $f$ and $g$ being $[-1, 1]$-valued functions.

6. **Set generation** (Bengio et al., 2023; Pan et al., 2023a). We assemble 16-sized subsets of a fixed 32-sized set. We employ the same additive reward function described in Section 4.

### C.2 AN EFFICIENT IMPLEMENTATION OF SAL

We start defining an *assignment function*.

**Definition C.2** (Assignment function). Let $f: \mathcal{S} \rightarrow \{0, 1, \ldots, m\} := [m]$, in which $m$ is the number of available computational units. Assume that $f$ satisfies the following conditions.

1. (Completeness). $f^{-1}(j) \neq \emptyset$ for each $j$, i.e., $f$ assigns at least one state to each available unit;

2. (Consistency). $\{\mathcal{S}_j = f^{-1}(j)\}_{j=0}^m$ is a fixed-horizon partition of $\mathcal{S}$.

Then, $f$ is called an *assignment function* and $\{S_j\}_{j=0}^m$ is the fixed-horizon partition associated to $f$.

Condition (1) above, which we call *completeness*, ensures that no computational unit is wasted, whereas condition (2) – *consistency* – guarantees that the partition of $\mathcal{S}$ induced by $f^{-1}$ is a FHP. In this context, we denote by $\mathrm{RV}(\mathcal{S})$ the space of $\mathcal{S}$-valued random variables (measurable functions). Then, we say that a function $g\colon [m] \to \mathrm{RV}(\mathcal{S})$ is a *stochastic inverse* of $f$ if $g(j) \in f^{-1}(j)$ with probability one for each $j \in [m]$; a similar concept exists in the literature of discrete normalizing flows (Hoogeboom et al., 2021; Tran et al., 2019). Notably, the distribution $q_j$ over the subnetworks' sources in the definition of SAL (see Eq. 9) corresponds to the PMF of the random variable $g(j)$.

Importantly, to efficiently implement SAL, one only needs to develop an $f$ that is both fast to compute and easily stochastically invertible. In this section, we show how to design such an assignment function for the problems of autoregressive design and set generation and for the hypergrid environment. The reader is reminded to recall Appendix A for an overview of each generative task.

**SAL for autoregressive models.** We first illustrate the concept of an assignment function for autoregressively generated objects, which are very common in applications (Jain et al., 2022; Malkin et al., 2022; Jiralerspong et al., 2024; Hu et al., 2024). For this problem, each state $s$ is represented as an element of the set $[[0, k-1]]^L$ for fixed $k$ (the vocabulary size, e.g., $k = 4$ for nucleotide strings) and $L$ (the sequence's length). Then, to construct a fixed-horizon partition, we choose a distance $D$ from the initial state and define

|             | Sets                  | Sequences             |
|-------------|-----------------------|-----------------------|
| Centralized | $0.092_{\pm 0.001}$   | $0.126_{\pm 0.012}$   |
| SAL         | $\mathbf{0.072}_{\pm 0.008}$ | $\mathbf{0.094}_{\pm 0.005}$ |

Table 2: Total variation distance between target and learned measures for the centralized model (top row) and SAL (bottom row).

$$h(s) = \sum_{1 \le i \le D} s_i \cdot k^{i-1} \tag{17}$$

as the $k$-ary representation of $s$. Then, $f(s) = \mathbf{1}_{\{\#s \ge D\}}(h(s) \,(\mathrm{mod}\, m) + 1)$ is our assignment function. Importantly, both $f(s)$ and $f^{-1}(j)$ add an negligible computational overhead to the training procedure. Indeed, to sample $s$ from $f^{-1}(j)$ for $j \ge 1$, we first define $h(s) = m \cdot \xi + (j-1)$ with $\xi$ randomly sampled from $[[0, \lceil k^D - 1/m \rceil]]$. Thus, to recover $s$ from $h(s)$, we only need to solve a (triangular) linear system; the details are provided next. Let $h_n(s) = m \cdot \xi + (j-1) \,(\mathrm{mod}\, k^n)$. Also,

$$h_n(s) = \sum_{1 \le i \le D} s_i k^{i-1} \,(\mathrm{mod}\, k^n) = \sum_{1 \le i \le n} s_i k^{i-1} \le k^n - 1 \tag{18}$$

when $n \le D$. Clearly, $h_1(s) = s_1$ and, recursively, $h_n(s) = k^{n-1}s_n + h_{n-1}(s)$. Therefore, $\mathbf{s} = (s_1, \ldots, s_D)$ jointly satisfy the triangular system $\mathbf{Ts} = \mathbf{h}$, in which $\mathbf{T}_{i,j} = \mathbf{1}_{\{i \ge j\}} \cdot k^{j-1}$ and $\mathbf{h}_i = h(s) \,(\mathrm{mod}\, k^i)$ for $i, j \in \{1, \ldots, D\}$. This system can be efficiently solved via backward substitution in parallel for a batch $\{h(s^1), \ldots, h(s^B)\}$ of $B$ sequences.

**SAL for set generation.** As in Section 4, we also consider the problem of generating $S$-sized sets with elements extracted from a source $\mathcal{W} = \{1, \ldots, W\}$ of fixed size $W$. In this setting, each $s \subseteq \mathcal{W}$ can be uniquely represented as a binary vector $s \in \{1, 0\}^W$ with $s_i = 1$ indicating that $i \in \mathcal{W}$ is a member of $s$. Notably, the elements $s$ at distance $D \le S$ to the initial state can also be completely described by $\sum_{i=1}^W s_i = D$. For these elements, the prefix $s_{1:D} \in \{1, 0\}^D$ has at least $\max\{0, 2D - W\}$ components equal to 1. Then, similarly to Equation 17, we define the assignment function $f$ for each $s \in \{1, 0\}^W$ with $\sum_{i=1}^W s_i = D$ as the binary representation of $s$ modulo the number of computational units,

$$f(s) = \sum_{1 \le i \le D} 2^{i-1} s_i \,(\mathrm{mod}\, m) + 1. \tag{19}$$

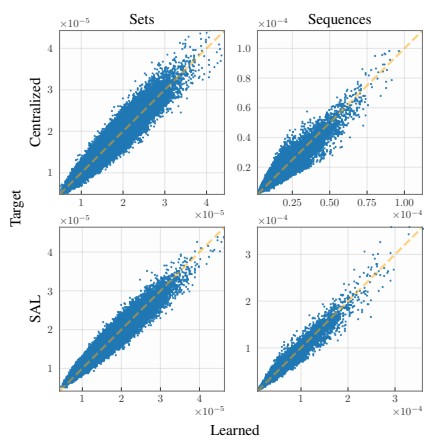

Figure 6: Target vs. learned distributions.

Figure 7: **SAL enacts faster mode discovery** under varying time budgets. The horizontal axis represents the total number of epochs used by the centralized model, which is set to the sum of the number of epochs for each leaf GFlowNet and the root GFlowNet to ensure the approaches are fairly compared. We provide additional evidence for the enhance performance of SAL in Figure 13.

On the other hand, let $\nu_{min} = \left\lceil 2^{\max\{0,2D-W\}-1}/m \right\rceil$ and $\nu_{max} = \left\lfloor 2^{D-1}/m \right\rfloor$. To stochastically reverse $f$, we define $h(j) = m \cdot \xi + (j-1)$ with $\xi \sim \mathcal{P}(\nu_{min}, \nu_{\max}, \lambda)$ sampled from a Poisson truncated at $[\nu_{min}, \nu_{max}]$ and parameterized by $\lambda$. Then, we let $[h(j)] \in \{1,0\}^D$ be the corresponding bit-wise representation of $h$ and $y_j \in \{1,0\}^{W-D}$ be a random binary vector with exactly $D - \sum_{i=1}^{D}[h(j)]_i$ components equal to 1, obtained via Fisher-Yates' shuffling algorithm (Fisher & Yates, 1963). Finally, we construct a sample $s = ([h(j)], y_j) \in \{1,0\}^W$ by concatenating $[h(j)]$ and $y_j$. Notably, an assignment function is closely related to the concept of *ranking* and *unraking* functions in computational combinatorics (Myrvold & Ruskey, 2001). To see this, we recall that a ranking $r$ (resp. unranking $u$) function of a set $\mathcal{S}$ (resp. $\{0, \ldots, |\mathcal{S}| - 1\}$) injectively maps each member of $\mathcal{S}$ to an element of $\{0, \ldots, |\mathcal{S}| - 1\}$ (resp. $\mathcal{S}$). A natural choice for $r$ is based on the lexicographic order on $\mathcal{S}$ (Liebehenschel, 1997); the corresponding $u$, however, may not be efficiently computable. Given a ranking function $r$ and a number $m$ of computing nodes for training a GFlowNet, we may defined an assignment function as $f(s) = r(s) \pmod{m} + 1$. We also provide an implementation of a lexicographic-based ranking and unranking functions for the set generation task to support future research on the development of more effective partitioning schemes for SAL.

**SAL for the hypergrid environment.** In conclusion, we also consider the difficult-to-explore hypergrid environment, which is defined by a distribution supported on $[[0, H-1]]^d$ for fixed $H$ (the grid's size) and $d$ (the grid's dimension) (Bengio et al., 2021). For this problem, we note the states $x$ at distance $D$ to the initial state can be fully described by the equation $\sum_{1 \le i \le d} x_i = D$. Equivalently, each $x$ satisfying the above equation can be injectively mapped to a point within the $(k-1)$-simplex. Hence, we define the assignment function $f$ over states at distance $D$ from $s_o$ as

$$f(x) = \min_{0 \le j < m} \left\{ \left(\frac{j}{m}\right)^{1/d-1} \le \frac{x_1}{D} < \left(\frac{j+1}{m}\right)^{1/d-1} + [j = m-1] \right\} + 1. \qquad (20)$$

The exponent $1/(d-1)$ is meant to ensure the workload is approximately homogeneously distributed among the computational units. To sample from $f^{-1}(j)$ for $j \ge 1$, we let

$$\nu_{min} = \left\lceil D \cdot (j-1/m)^{1/d-1} \right\rceil \quad \text{and} \quad \nu_{max} = \left\lfloor D \cdot (j/m)^{1/d-1} \right\rfloor + [j = m-1] \qquad (21)$$

and pick $x_1$ uniformly at random from $[[\nu_{min}, \nu_{max}]]$. Then, $(x_2, \ldots, x_d)$ is drawn from a Dirichlet-multinomial with number of trials $D - x_1$ and concentration parameter $\boldsymbol{\alpha}$ set (arbitrarily) to $\mathbf{1}$.

Remarkably, the hypergrid environment illustrates an approach differing from the strategy of encoding-as-integer and computing-the-remainder that was implemented for the other tasks. Figure 8 shows the partition for $d = 2$, $H = 8$, and $m = 2$, which was the setup for Figure 4. We represent in red and teal the sources of the subnetworks assigned to the leaves $j = 1$ and $j = 2$, respectively, and in blue the remaining states. Recall that, by definition, all descendants of a state $s$ are members of $s$'s partition. In this scenario, we hope that the development of sophisticated and expert-driven partitioning techniques will greatly benefit the use of GFlowNets in specialized domains, e.g., drug discovery.

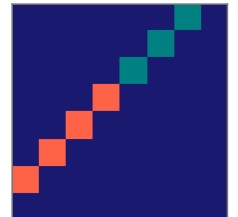

Figure 8: FHP for the hypergrid.

**SAL results in better distributional approximations.** Figure 4 shows that, when compared against a centralized approach, SAL achieves a better distributional approximation for the hypergrid environment under a fixed time-budget (for SAL, the training time is the longest client's training time

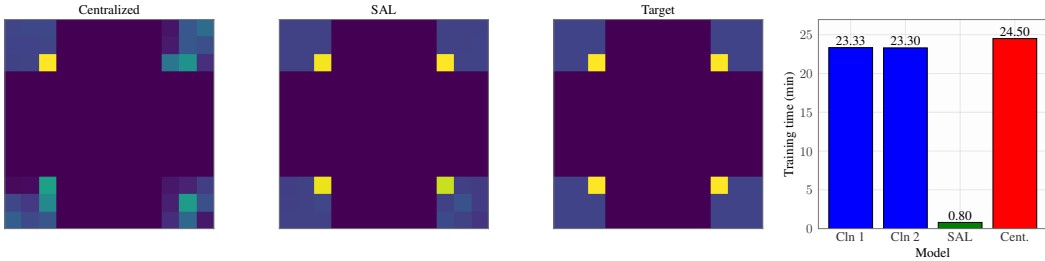

Figure 9: **SAL improves the discovery of high-value states** for all considered tasks. For a fair comparison, the centralized model is allowed to explore for twice the number of epochs permitted to each client, ensuring the training times are roughly the same for SAL and the standard GFlowNet.

Figure 10: **SAL results in a more accurate approximation** than a centralized approach for a similar time budget on the $12 \times 12$ grid. Complementarily to Fig. 4, all models are trained by minimizing the SubTB (instead of TB) objective with $\lambda = 0.9$. The running time for SAL is determined by the training time of the longest client (blue columns) plus the much faster aggregation step (green column).

plus the time for aggregation). Figure 6 and Table 2 corroborate this claim for the set generation and sequence design tasks, showcasing that SAL learns a distribution that matches the target more closely than a standardly trained GFlowNet. For the other tasks, learning an accurate distributional approximation is not as important as finding high-value objects, and that is the reason we do not consider them here. Notably, these results are consistent with Theorems 5.2 and 5.4: by reducing the size of the state graph that each model needs to focus on, SAL potentially facilitates the learning of a generalizable policy network and leads to a more accurate approximation to the target.

**SAL greatly improves mode-discovery.** In the GFlowNet literature, a *mode* is often defined as a state $x$ whose associated reward $R(x)$ is larger than a predefined threshold $t$; see, e.g., (Bengio et al., 2021; Pan et al., 2023a;b; Madan et al., 2022; Jang et al., 2024; Malkin et al., 2022). For our experiments, we fix $t = 0.1$ for the hypergrid environment and, for the other generative tasks, we sample an initial batch of $2 \cdot 10^4$ from the uniform policy and set $t$ as the $0.99$ quantile of the observed rewards. Importantly, the same threshold is used for both the centralized, leaf, and root GFlowNets. Under these conditions, Figures 4, 9, 7, and 13 show that SAL enacts a drastic improvement of the mode-discovery rate over a centralized approach for varying computational budgets in all considered generative problems, leading to the discovery of up to 8x more modes. There are two reasons for this. Firstly, the distributed nature of SAL ensures that a much larger portion of the state space is explored in a significantly shorter amount of time. Secondly, each client model focus on a subset of the state graph and may be regarded as a *specialist* in the corresponding subtask. By collecting the samples fostered by these local specialists, we end up with a significantly more diverse and valuable collection than the one that would be obtained by, e.g., independently training multiple GFlowNets in parallel. Remarkably, this interpretation highlights the relevance of appropriately defining a fixed-horizon partition of the state graph, an issue that defines a key future direction for our work, as we discussed in Section 7 of the main text.

**Alternative learning objectives for SAL.** For simplicity of exposition, we outlined SAL in Definition 6.2 as a collection of TB-minimizing GFlowNets. However, as previously discussed, one can straightforwardly adapt alternative learning objectives (e.g., SubTB (Madan et al., 2022)) and sampling techniques (e.g., replay buffer (Deleu et al., 2022)) to the context of SAL. We illustrate

these extensions here. Firstly, the SubTB objective for the $j$th partition would take the form

$$\mathcal{L}_{\text{SubTB}}^j(p_F, F) = \mathop{\mathbb{E}}_{s \sim q_j} \mathop{\mathbb{E}}_{\tau \sim p_E^j(\cdot|s)} \left[ \sum_{1 \le m < n \le |\tau|} \frac{\lambda^{n-m}}{\sum_{1 \le s < t \le |\tau|} \lambda^{t-s}} \left( \log \frac{F(\tau_m) p_F(\tau_{m:n}|\tau_m)}{F(\tau_n) p_B(\tau_{n:m}|\tau_n)} \right)^2 \right],$$

in which $q_j$ and $p_E^j$ are a distribution over initial states of and a sampling policy for the $j$th partition, respectively. Secondly, the replay buffer would store the trajectories $\tau$ leading to high-value states within the $j$th partition, as measured by either $R$ (for leaf partitions) or $R^o$ (for the root partition; see Algorithm 1). Figure 10 compares the accuracy of SAL against a centralized GFlowNet, both of which trained by SubTB minimization ($\lambda = 0.9$), for the $12 \times 12$ hypergrid environment. Similarly to Figure 4, SAL achieves a better distributional approximation in this case. Additionally, we found that SubTB leads to faster convergence with respect to (A)TB (not reported) in this particular problem, consistently with the evidence at Madan et al. (2022, Figure 1). On the other hand, our experiments did not provide evidence in favor of using the replay buffer. However, we acknowledge that a deeper empirical investigation, in the fashion of Vemgal et al. (2023)'s work, is required.

## C.3 THEORETICAL ANALYSIS AND EXTENSIONS

This section aims to answer two core questions regarding the nature of SAL. From a sampling perspective, we ask which distribution each client learns and suggest potential diagnostic techniques to evaluate their distributional accuracy. From a distributed learning standpoint, we assess the extent to which local errors are propagated to the global model. Additionally, we formally extend SAL to accommodate the learning over multi-layered fixed-horizon partitions of the state graph.

**Local sampling distributions.** Each leaf GFlowNet in SAL learns a distribution over a subset $\mathcal{X}_j$ of the set of terminal states $\mathcal{X}$. The character of such distribution, however, was not considered in the foregoing discussion, and one may wonder whether it just corresponds to the restriction of the target to $\mathcal{X}_j$. As we show below, this is not generally the case: the optimal *leaf distribution*, which we denote by $p_T^j$, depends on both $\mathcal{X}_j$ and on the specific structure of the state graph induced by the leaf $\mathcal{S}_j$.

**Proposition C.3** (Local sampling distributions). *Let $\{\mathcal{S}_j\}_{j=0}^m = \text{FHP}(\mathcal{S})$ and $\{\mathcal{G}_j\}_{j=0}^m$ be the corresponding GFlowNets. Also, denote by $\text{TDc}(s)$ the set of terminal descendants of $s$ on the original state graph, i.e., $x \in \text{TDc}(s)$ if $x \in \mathcal{X}$ and there is a directed path from $s$ to $x$. Then, for fixed backward policy $p_B$, the solution that globally minimizes Equation 9 satisfies*

$$F_j(s) = \sum_{x \in \text{TDc}(s)} R(x) \sum_{\tau: s \rightsquigarrow x} p_B(\tau|x) \quad and \quad p_T^j(x|s) = \frac{R(x)}{F_j(s)} \sum_{\tau: s \rightsquigarrow x} p_B(\tau|x) \tag{22}$$

*for each $j \in \{1, \ldots, m\}$, $s \in \mathcal{I}_j$, and $x \in \mathcal{X}_j := \bigcup_{s \in \mathcal{I}_j} \text{TDc}(s)$.*

When the sum over backward trajectories in Equation 22 does not depend on $x$, e.g., for autoregressively generated object (in which $p_B(\tau|x) = 1$ for the unique trajectory $\tau$ connecting $s$ to $x$) and sets (in which the sum depends only on the depth of $s$ when $p_B$ is fixed to an uniform policy), Proposition C.3 says that $p_T^j$ can be nicely interpreted as the restriction of the original target $R$ to the induced terminal set $\mathcal{X}_j$. We emphasize this fact in the corollary below.

**Corollary C.4** (Local sampling distributions for autoregressive models). *In the context of Proposition C.3, assume that the state graph is represented as a tree. Then, $p_B(\tau|x) = 1$ for every $\tau$ and*

$$F_j(s) = \sum_{x \in \text{TDc}(s)} R(x) \quad and \quad p_T^j(x|s) = \frac{R(x)}{\sum_{x \in \text{TDc}(s)} R(x)}. \tag{23}$$

*For the set generation task, it also holds that $p_T^j(x|s) \propto R(x)$ for each $x \in \text{TDc}(s)$ and $s \in \mathcal{I}_j$.*

Interestingly, the result above suggests a straightforward procedure for assessing the goodness-of-fit of the locally trained GFlowNets. When $\mathcal{X}_j$ is considerably smaller than $\mathcal{X}$, we can compute the normalized target in Equation 23 and directly compare it against the learned distribution in $\mathcal{X}$. Otherwise, any technique for diagnosing GFlowNets can be readily applied to probe the accuracy of $p_T^j$, e.g., measuring the Spearman correlation between $\log p_T^j(\cdot|s)$ and $\log R(x)$ for $x \in \mathcal{X}_j$ (Malkin et al., 2022; Madan et al., 2022; Shen et al., 2023; Tiapkin et al., 2024; Chen & Mauch, 2024).

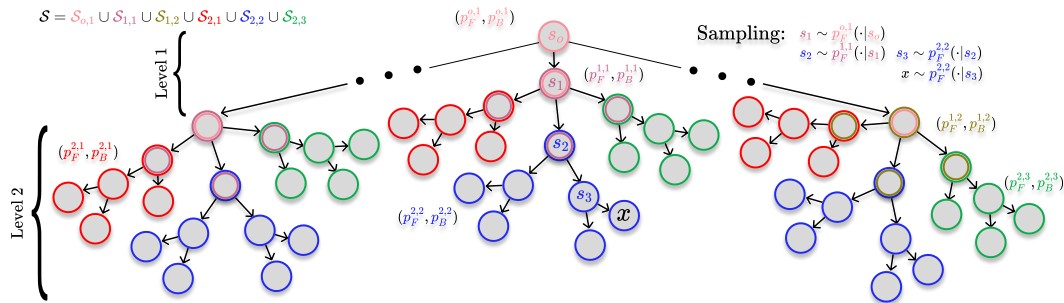

Figure 11: **Illustration of Recursive SAL.** We show a two-level partition with $m_1 = 2$ models within the first level and $m_2 = 3$ models within the second one. For training, we first train models at the bottommost layer (represented in blue, red, and green) and recursively proceed upwards towards the middle (magenta and yellow) and top (root partition, shown in pink) layers. For the non-root layers, learning is based on minimizing $\mathcal{L}_{\text{ATB}}$ with the reward defined as in Equation 25; for the root, we minimize $\mathcal{L}_{\text{TB}}$ instead. For inference, we start at $s_o$ and iteratively select the policy based on the current state, as illustrated in the highlighted trajectory and in the annotated text on the top-right corner.

**Sensibility to error propagation.** Proposition C.3 raises an important question: how do the errors of the leaf models $\{\mathcal{G}_j\}_{j=1}^m$ affect the global goodness-of-fit? To address this issue, the next proposition shows that the contribution of $\mathcal{G}_j$ to the overall distributional error is an increasing function of the probability mass associated to the $j$th leaf by the root model, $\mathcal{G}_o$, and of how inaccurate $\mathcal{G}_j$'s itself is.

**Proposition C.5** (Sensibility to error propagation). *Let $\{\mathcal{S}_j\}_{j=0}^m = \text{FHP}(\mathcal{S}, m)$ with GFlowNets $\{\mathcal{G}_j\}_{j=0}^m$. Assume that $\mathcal{G}_o$ satisfies its balance condition. Also, define*

$$Z_R = \sum_{x \in \mathcal{X} \cap \mathcal{S}_o} R(x), \quad Z_F = \sum_{x \in \bigcup_{1 \leq j \leq m} \mathcal{I}_j} F_j(x), \quad and \quad Z = \sum_{x \in \mathcal{X}} Z_R, \tag{24}$$

*and note that $Z_R + Z_F$ is the partition function associated to $\mathcal{G}_o$. Then, the TV distance between the learned distribution $p_T$ in Equation 16 and the target $\pi(x) \propto R(x)$ for $x \in \mathcal{X}$ satisfies*

$$\text{TV}\left(p_T, \pi\right) \leq \frac{Z_R}{2} \underbrace{\left| \frac{1}{Z} - \frac{1}{Z_R + Z_F} \right|}_{\text{Error in estimating } Z_F} + \frac{1}{2} \sum_{x \in \mathcal{X} \setminus \mathcal{S}_o} \underbrace{\left| \mathbb{E}_{s \sim p_{T, \setminus \mathcal{X}}^o} \left[ \pi(x) - \frac{Z_F}{Z_R + Z_F} p_T^{f(s)}(x|s) \right] \right|}_{\text{Error of the local approximations}},$$

*in which $p_{T, \setminus \mathcal{X}}^o$ is the restriction of $p_T^o$ to $\bigcup_{1 \leq j \leq m} \mathcal{I}_j$ and $f$ is the assignment function.*

Importantly, the bound above is tight in the sense that, when the root and leaf models satisfy their balance conditions, $Z = Z_R + Z_F$ and $\pi(x) \propto \mathbb{E}_s[p_T^{f(s)}(x|s)]$, as we show in the proof of Proposition C.5 in Appendix D. There, we also provide an alternative, trajectory-based upper bound on the TV distance that similarly highlights the relatively large impact of the distributional errors associated with large-probability leaves to the overall accuracy. Heuristically, this suggests that SAL may benefit from a FHP that approximately homogeneously distribute the probability mass among the leaf partitions, ensuring that no client has a disproportionate role on shaping the accuracy of the aggregated model. To achieve this, however, one needs prior knowledge of the reward function; the definition of a *good* fixed-horizon partition should be done in problem-by-problem basis. We also believe that a human expert would have a remarkable impact on the effectiveness of SAL for the highly-specialized, molecular-biology-based, tasks in which GFlowNet are often implemented.

**Recursive SAL.** In Section 6, we introduced an extension of SAL to recursively defined FHPs. Proposition C.6 formalizes this procedure and demonstrates through an inductive argument that the resulting model samples correctly from the target distribution.

**Proposition C.6** (Recursive SAL). *Let $\mathcal{S}$ be the vertices of a state graph with diameter $D$. Then, for sequences $0 = d_o < d_1 < d_2 < \cdots < d_k \leq D$ and $\{m_o = 1, m_1, \ldots, m_k\}$, we define $\bigcup_{1 \leq j \leq m_i} \mathcal{I}_{ij}$ as a disjoint $m_i$-partition of the states distanced $d_i$ from $s_o$. Also, let $\mathcal{X}_k = \{x \in \mathcal{X}: d(x, s_o) \geq d_k\}$ and, for $i < k$, let $\mathcal{X}_i = \{s: s \in \mathcal{I}_{i+1,j} \vee (d(s, s_o) \leq d_i \wedge s \in \mathcal{X})\}$. Finally,*

*we define $\mathcal{G}_i = \{(p_F^{i,j}, p_B^{i,j}, F_{ij}) : 1 \leq j \leq m_i\}$ as a set of GFlowNets trained on a state graph with initial states $\bigcup_j \mathcal{I}_{ij}$, terminal states $\mathcal{X}_i$, and reward function $R_i$ such that*

$$R_i(s) = \begin{cases} F_{i+1}(s), & \text{if } s \in \mathcal{I}_{i+1} \text{ and } i < k, \\ R(s), & \text{if } s \in \mathcal{X}. \end{cases} \tag{25}$$

*Then, when the GFlowNets $\cup_{i=0}^k \mathcal{G}_i$ satisfy their respective balance conditions, the generative process starting at $s_o$ and recursively following $p_F^{i,b}$ until either reaching $\mathcal{I}_{i+1,a}$, at which point the guiding forward policy is changed to $p_F^{i+1,a}$, for $0 \leq i \leq k$, or reaching $\mathcal{X}$, signaling to stop the generation and return the sampled object, samples each $x \in \mathcal{X}$ proportionally to $R(x)$.*

**Remarks on Recursive SAL.** In plain English, the above proposition says that we can use the learned flow function $F_{i+1}$ at the $(i+1)$th layer as the reward function of the GFlowNets within the $i$th layer to obtain a correct sampler when training the GFlowNets in a hierarchical fashion. In computational terms, the number of trained models grows linearly with the depth $k$ and width $\max_i m_i$ of the multi-layered partition. In the light of Theorems 5.2 and 5.4, however, each model would have to solve a considerably simpler problem and we may be able to use a significantly smaller neural network to parameterize the corresponding forward policies, with advantageous consequences for both generalization—via the KL term in Theorems 5.2 and 5.4, which increases with the number of estimable parameters—and storage. Albeit we do not provide an empirical evaluation of Recursive SAL in this work, we believe its implementation could be beneficial for problems with very large trajectory sizes and are optimistic about its potential applications in future endeavors.

## C.4 CONDITIONAL SAL

**SAL and state-conditional flows.** Bengio et al. (2023, Section 4.3) introduce *state-conditional flows* as a family $\{F_s\}_{s \in \mathcal{S}}$ of flow functions defined on the subgraphs $G_s$ induced by $\{s' \in \mathcal{S} : s \geq s'\}$, in which $s \geq s'$ means that there is a path from $s$ to $s'$ on the original state graph. Remarkably, a $\text{FHP}(\mathcal{S}, m)$ can be interpreted as a subset $\{F_s\}_{s \in \cup_{j=1}^m \mathcal{I}_j}$ of a state-conditional flow. In spite of these similarities, which serve only to strengthen the foundations of our work, we emphasize the novelty and demonstrated effectiveness of our distributed strategy for *learning* state-conditional flows.

**Learning reward-conditioned GFlowNets with SAL.** Recently, there has been growing interest in *reward-conditioned* flows, in which we learn a family $\{F_c\}_{c \in \mathcal{C}}$ of flow functions conditioned on some information $c \in \mathcal{C}$ given as an additional input to the neural network. In most applications, $c$ corresponds to either a temperature parameter (Zhang et al., 2023a; Kim et al., 2024a) defining the peakiness of the target distribution or pharmaceutical properties (Roy et al., 2023; Pandey et al., 2024) guiding the drug discovery process. In view of this, we extend SAL to accommodate the distributed learning of reward-conditioned GFlowNets on a *conditioned FHP*, i.e., a FHP that depends on the conditioning information. This may be formally expressed as follows.

*Remark* C.7 (Reward-conditioned SAL). Let $\mathcal{C}$ be a set of conditioning information and $\{R_c : \mathcal{X} \to \mathbb{R}_+ : c \in \mathcal{C}\}$ be the corresponding family of conditioned rewards. Also, let $F : \mathcal{C} \times \mathcal{S} \to \mathbb{R}_+$ and $p_F : \mathcal{C} \times \mathcal{S} \times \mathcal{S} \to [0, 1]$ be a conditional flow function and a conditional forward policy, that is, $p_F(c, \cdot, \cdot)$ is a forward policy for each $c$. Finally, let $\{S_j^c\}_{j=0}^m = \text{FHP}(\mathcal{S}, m, c)$ be a conditioned FHP and $\{\mathcal{G}_j\}_{j=0}^m$ be a family of reward-conditional GFlowNets. Following the arguments for demonstrating Theorem C.1, it is easy to see that SAL samples correctly in proportion to $R_c$ when each $\mathcal{G}_j$ satisfies its respective balance condition with respect to $R_c$ for every $c \in \mathcal{C}$ and $0 \leq j \leq m$.

Conditional GFlowNets are commonly implemented for *controllable generation* by setting the conditioning information $c$ at inference time (Roy et al., 2023); see also Lau et al. (2023)'s QGFN. In fact, Pandey et al. (2024) recently explored this principle for the effective exploration of chemical space at an atomic-level given some desirable pharmacological properties, e.g., synthetizability. In this regard, Remark C.7 ensures that most of these approaches can ba adapted to the distributed setting via SAL, and we believe that assessing the resulting methods is an important research direction.

## C.5 SAL AND EP-GFLOWNETS

The reward function of GFlowNets can often be decomposed as $R(x) = \prod_{i=1}^K R_i(x)$ (Jain et al., 2023; Deleu et al., 2023; Zhou et al., 2024; Pandey et al., 2024), e.g., in multi-objective problems in

which each $R_i$ is an objective and our goal is finding samples that are concomitantly high valued for every $R_i$. In these cases, da Silva et al. (2024) proposed a divide-and-conquer algorithm for learning $K$ GFlowNets in parallel, each targeting a reward $R_i$, and then aggregating them with an extra GFlowNet. The resulting model, termed EP-GFlowNet, trains $(K + 1)$ GFlowNets on the *same state graph*, in sharp contrast to SAL which trains a GFlowNet for each component of a FHP. To further highlight the distinction between SAL and EP-GFlowNets, we show below that these approaches can be implemented in a complementar manner.

**Proposition C.8** (EP-SAL). *Let $R(x) = \prod_{i=1}^{K} R_i(x)$ be a multiplicative decomposition of R. For each $1 \le i \le K$, let $\{S_j^i\}_{j=0}^{m_i} = \mathrm{FHP}_i(\mathcal{S}, m_i)$ be a fixed-horizon partition of the state space $\mathcal{S}$. Also, let $\mathcal{G}_i = \{(p_F^{i,j}, p_B^{i,j}, F_{i,j}): 0 \le j \le m_i\}$ be the root- and leaf-GFlowNets corresponding to the i-th FHP and denote by $p_F^i$ and $p_B^i$ the induced distributions over trajectories. Assume that each $p_F^i$ samples terminal objects in $\mathcal{X}$ proportionally to $R_i$. Finally, let $p_E$ be any positive probability measure over trajectories. Then, if a GFlowNet $(p_F, p_B, \mathcal{G})$ globally minimizes*

$$\mathrm{Var}_{\tau \sim p_E}\left(\log \frac{\prod_{i=1}^{K} p_F^i(\tau)}{\prod_{i=1}^{K} p_B(\tau|x)} + \log \frac{p_F(\tau)}{p_B(\tau)}\right), \tag{26}$$

*the marginal $p_T$ of $p_F$ over $\mathcal{X}$ matches $R := \prod_{i=1}^{K} R_i$ up to a normalizing constant.*

*Proof.* The result follows directly from Theorem C.1 and (da Silva et al., 2024, Theorem 3.1). □

For the sake of completeness, let $\tau_{1:d}$ denote the first $d$ transitions of $\tau$ and $\tau_{d:}$ be its complementar.

Then, let $d_i$ be the distance from the initial state of $s_o$ to the sets $\mathcal{I}_{i,j}$ in the underlying state graph for $j \in \{1, \dots, m_i\}$ (recall Definition 6.1, and note that $d_i$ does not depend on $j$). Under these conditions, the sampling distribution $p_F^i$ in Proposition C.8 can be formally written as

$$p_F^i(\tau) = \begin{cases} p_F^o(\tau) \text{ if } |\tau| \le d_i, \\ p_F^o(\tau_{1:d_i}) \sum_{j=1}^{m_i} \sum_{s \in \mathcal{I}_j} \mathbf{1}_{\{s \in \tau\}} p_F^{i,j}(\tau_{d_i:}|s), \text{ otherwise.} \end{cases} \tag{27}$$

Importantly, Equation 27 can be efficiently computed by keeping track of the partition associated to each sampled state. In practice, the computational overhead wrt directly evaluating $p_F$ is negligible.

**Empirical illustration.** From an empirical standpoint, Proposition C.8 says that SAL can be used to learn a set $\mathcal{G}_i$ of GFlowNets jointly sampling $x \in \mathcal{X}$ in proportion to $R_i(x)$. Then, given $\mathcal{G}_i$ for $1 \le i \le K$, a GFlowNet sampling in proportion to $\prod_{i=1}^{K} R_i$ can be obtained by minimizing EP-GFlowNet's learning objective. Figure 12 empirically validates this result for the task of set generation with the same hyperparameters considered in Figure 6 with $K = 2$ and $m_i = 2$ for each $i \in \{1, 2\}$. Each GFlowNet was trained for 512 epochs and the log-utilities defining $R_i$ were independently sampled for each client. Nonetheless, in spite of its soundness, the effectiveness of this mixed approach in realistic problems remains to be assessed. Looking at the bigger picture, these observations emphasize the *composability* of GFlowNets (Garipov et al., 2023), which might be relevant for the design of more data-efficient algorithms (Du & Kaelbling, 2024).

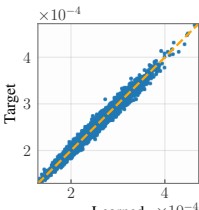

Figure 12: SAL of EP-GFlowNets.

# D  PROOFS

## D.1  PROOF OF LEMMA 4.1

We simply note that the space of $T$-sized subsets of $\{1, \ldots, W\}$ has size $\binom{W}{T}$ and the space of $T$-sized subsets of $\{2, \ldots, W\}$ has size $\binom{W-1}{T}$. Since

$$\frac{\binom{W-1}{T}}{\binom{W}{T}} = \frac{W-T}{W} \to 1 \tag{28}$$

when $W \to \infty$, we can always find for any $\xi \in (0,1)$ a $W$ and a $T$, both of which potentially depending on $\xi$, for which $|\mathcal{X}'| \geq \xi |\mathcal{X}|$. For the cases considered in Figure 1, in particular, we compute the following proportions: $(32-6)/32 = 81.25\%$ and $(64-6)/64 \approx 90.63\%$.

## D.2  PROOF OF PROPOSITION 4.2

Our proof has three steps. Firstly, we use Hölder's inequality to bound the expectation of $|\pi(x) - p_T(x)|$. Secondly, we rely on Jensen's inequality to bound $|\pi(x) - p_T(x)|$ with an expectation of $|p_F(\tau)/p_B(\tau|x) - \pi(x)|$ over $\tau$. Thirdly, we convert the probabilities to a log-scale with a simple technical argument based on the Taylor expansion of log. For this, let $\phi(x) = |\pi(x) - p_T(x)|$. Then,

$$
\begin{aligned}
\mathbb{E}_{x \sim q_{E,T}}[\phi(x)] &= \sum_{x \in \mathcal{X}} \phi(x) q_{E,T}(x) \\
&= \sum_{x \in \mathcal{X}} \phi(x) \cdot \frac{q_{E,T}(x)}{p_{E,T}(x)} p_{E,T}(x) \\
&\leq \left( \sum_{x \in \mathcal{X}} \phi(x)^q p_{E,T}(x) \right)^{\frac{1}{q}} \left( \sum_{x \in \mathcal{X}} \left( \frac{q_{E,T}(x)}{p_{E,T}(x)} \right)^p p_{E,T}(x) \right)^{\frac{1}{p}} \\
&= \left( \mathbb{E}_{x \sim p_{E,T}}[\phi(x)^q] \right)^{1/q} \left( \mathbb{E}_{x \sim p_{E,T}} \left[ \left( \frac{q_{E,T}(x)}{p_{E,T}(x)} \right)^p \right] \right)^{\frac{1}{p}}
\end{aligned}
\tag{29}
$$

for any $p, q > 1$ such that $1/p + 1/q = 1$. For $p = q = 2$, this bound becomes

$$\mathbb{E}_{x \sim q_{E,T}}[\phi(x)] \leq \left( \mathbb{E}_{x \sim p_{E,T}}[\phi(x)^2] \left( \chi^2(q_{E,T} \| p_{E,T}) + 1 \right) \right)^{\frac{1}{2}}. \tag{30}$$

For GFlowNets, we may write $p_T(x) = \mathbb{E}_{\tau \sim p_B}[p_F(\tau)/p_B(\tau|x)]$. Hence, by Jensen's inequality,

$$
\begin{aligned}
\mathbb{E}_{x \sim p_{E,T}} \left[ \phi(x)^2 \right] &= \mathbb{E}_{x \sim p_{E,T}} \left[ \left( \mathbb{E}_{\tau \sim p_B} \left[ \frac{p_F(\tau)}{p_B(\tau|x)} - \pi(x) \right] \right)^2 \right] \\
&\leq \mathbb{E}_{x \sim p_{E,T}} \left[ \mathbb{E}_{\tau \sim p_B} \left[ \left( \frac{p_F(\tau)}{p_B(\tau|x)} - \pi(x) \right)^2 \right] \right].
\end{aligned}
\tag{31}
$$

In conclusion, we show that

$$\left( \frac{p_F(\tau)}{p_B(\tau|x)} - \pi(x) \right)^2 \lesssim \left( \log \frac{p_F(\tau)}{p_B(\tau|x)} - \log \pi(x) \right)^2. \tag{32}$$

In fact, let $M = \max_{\tau,x} \frac{p_F(\tau)}{p_B(\tau|x)}$, which always exists due to the finiteness of the state space. For instance, $M \leq 1$ for autoregressive generative tasks (i.e., when $p_B(\tau|x) = 1$). Thus,

$$\left( \frac{p_F(\tau)}{p_B(\tau|x)} - \pi(x) \right)^2 \leq M^2 \left( \frac{p_F(\tau)}{M p_B(\tau|x)} - \frac{\pi(x)}{M} \right)^2.$$

The lemma below, which is a direct consequence of the mean value theorem, ensures that the quantity above is bounded above by the log-squared difference between $p_F(\tau)/p_B(\tau|x)$ and $\pi(x)$.

**Lemma D.1** (Lipschitzness of $x \mapsto e^x$). *For every $x, y \in (0, 1]$, $|\log x - \log y| \geq |x - y|$.*

*Proof.* Consider $f\colon (-\infty, 0] \to \mathbb{R}$, $f\colon t \mapsto e^t$, and notice that $|f'(t)| = |e^t| \leq 1$. Consequently, by the mean value theorem, $f$ is 1-Lipstchitz and $|e^t - e^s| \leq |t - s|$ for every $t, s \in (-\infty, 0]$. By letting $\log x = t$ and $\log y = s$, we conclude that $|x - y| \leq |\log x - \log y|$ for $x, y \in (0, 1]$. $\square$

In summary, we have shown that

$$\mathbb{E}_{x \sim q_{E,T}}[\phi(x)] \lesssim \left( \mathbb{E}_{x \sim p_{E,T}} \mathbb{E}_{\tau \sim p_B} \left( \log \frac{p_F(\tau)}{p_B(\tau|x)} - \log \pi(x) \right)^2 \left( \chi^2(q_{E,T}||p_{E,T}) + 1 \right) \right)^{1/2}. \quad (33)$$

The statement thereby follows by considering an uniform reference distribution, $q_{E,T}(x) = \frac{1}{|\mathcal{X}|}$,

$$\mathrm{TV}\,(p_T, \pi) = \frac{|\mathcal{X}|}{2} \mathbb{E}_{x \sim q_{E,T}}[\phi(x)]$$

$$\lesssim \left( \mathbb{E}_{x \sim p_{E,T}} \mathbb{E}_{\tau \sim p_B} \left( \log \frac{p_F(\tau)}{p_B(\tau|x)} - \log \pi(x) \right)^2 \left( \chi^2(q_{E,T}||p_{E,T}) + 1 \right) \right)^{1/2}.$$

## D.3 Proof of Proposition 5.1

For completeness, we provide a proof of Proposition 5.1. Clearly, it is enough to show that

$$L_{\mathrm{FCS}}(P) \leq \hat{L}_{\mathrm{FCS}}(P) + \sqrt{\frac{\eta}{2}} \quad \text{and} \quad L_{\mathrm{FCS}}(P) \leq \hat{L}_{\mathrm{FCS}}(P) + \eta + \sqrt{\eta(\eta + 2\hat{L}_{\mathrm{FCS}}(P))}, \quad (34)$$

in which we omit the dependence of $\hat{L}_{\mathrm{FCS}}$ on the dataset $\mathcal{T}_n$ for conciseness. We recall that $\eta = \frac{\mathrm{KL}(P||Q) + \log 2\sqrt{n_\alpha}/\delta}{n_\alpha}$, with $n_\alpha = \lfloor (\lfloor 1 - \alpha)n \rfloor$, is the complexity term that depends on the prior $Q$, posterior $P$, confidence $\delta$, and the number of data points $n_\alpha$. Notably, both inequalities directly follow from Maurer (2004, Theorem 5) bound: with probability $1 - \delta$ over $\mathcal{T}_n$,

$$\mathrm{kl}(\hat{L}_{\mathrm{FCS}}(P)||L_{\mathrm{FCS}}(P)) \leq \eta, \quad (35)$$

in which $\mathrm{kl}$ represents the binary KL divergence, i.e., $\mathrm{kl}(p||q) = p \log \frac{p}{q} + (1 - p) \log \frac{1-p}{1-q}$. Below, we show that $\mathrm{kl}(p||q)$ is greater than or equal to $(p-q)^2/2q$ when $p < q$.

**Lemma D.2.** *(Boucheron et al., 2013, Exercise 2.8). Let $h(t) = (1-t)\log(1-t) + t$ and $p\colon \{1, 0\} \to [0, 1]$ (resp. $q$) represent the PMF of a Bernoulli with parameter $p \in [0, 1]$. Then,*

$$\mathbb{E}_{x \sim \mathcal{B}e(q)} h\left( 1 - \frac{p(x)}{q(x)} \right) = \mathrm{kl}(p||q) \quad (36)$$

*and $h(t) \geq \frac{t^2}{2}$ for $t \in [0, 1]$. In particular, $\mathrm{kl}(p||q) \geq (p-q)^2/2q$ when $p \leq q$.*

*Proof.* Equation 36 follows from a direct algebraic manipulation of the left-hand side. On the other hand, define

$$g(t) = h(t) - \frac{t^2}{2} \quad (37)$$

for $t \in [0, 1]$. Then, $g$ is continuous, $g(0) = 0$, and $g(t) \to 1/2$ when $t \to 1$. Also, $g'(t) = -\log(1 - t) - t \geq 0$ for $t \in [0, 1]$ since $-\log(1 - t) = |\log(1 - t)| \geq t$. In conclusion,

$$\mathbb{E}_{x \sim \mathcal{B}e(q)} h\left( 1 - \frac{p(x)}{q(x)} \right) \geq \frac{1}{2} \mathbb{E}_{x \sim \mathcal{B}e(q)} \left( 1 - \frac{p(x)}{q(x)} \right)^2 \geq \frac{(q-p)^2}{2q} \quad (38)$$

when $p \leq q$. $\square$

By the symmetry of Equation 35 with respect to $L_{\mathrm{FCS}}$ and $\hat{L}_{\mathrm{FCS}}$, we conclude that

$$L_{\mathrm{FCS}}(P) - \hat{L}_{\mathrm{FCS}}(P) \leq \sqrt{2L_{\mathrm{FCS}}(P)\eta}. \quad (39)$$

Under these circumstances, the inequality $L_{\mathrm{FCS}}(P) \leq \hat{L}_{\mathrm{FCS}}(P) + \eta + \sqrt{\eta(\eta + 2\hat{L}_{\mathrm{FCS}}(P))}$ is obtained by solving the above quadratic inequality on $\sqrt{L_{\mathrm{FCS}}(P)}$. Through a similar reasoning, $\mathrm{kl}(p||q) \geq 2(p - q)^2$ by Pinsker's inequality and, consequently, $L_{\mathrm{FCS}}(P) \leq \hat{L}_{\mathrm{FCS}}(P) + \sqrt{\eta/2}$. These results jointly entail Proposition 6.

### D.4 Proof of Lemma B.1

Recall that $\tilde{p}_T(x) = \sum_{\tau \to x} \tilde{p}_F(\tau)$ for $\tilde{p}_F^{\alpha,p_F} = \alpha p_U + (1-\alpha)p_F$, in which we make the dependence of $\tilde{p}_F$ on $\alpha$ and on the (unconstrained) policy $p_F$ explicit. Let $\mathcal{F}(\alpha, p_F)$ be the family of such policies and $\mathcal{F}(\alpha)$ be the set of $\alpha$-greedy policies. It is straightforward to see that $\mathcal{F}(\alpha)$ is a convex set. Clearly, it is enough to ensure that $\min_{\alpha>0,p_F} \min_x \tilde{p}_T^{\alpha,p_F}(x) \leq \min_x \pi(x)$, namely, that the rarest object can be sampled correctly by properly adjusting $p_F$ and a (non-zero) $\alpha$. Indeed, Bengio [ref, Theorem 8] showed that, for each given backward policy $p_B$ and positive reward $R$, there is a unique forward policy $p_F$ for which the marginal $p_T(x) \propto R(x)$ for each $x \in \mathcal{X}$. Hence, since $\tilde{p}_T^{\alpha,p_F} = \alpha p_{U,T} + (1-\alpha)p_T$, with $p_{U,T}$ being the marginal of $p_U$ over $\mathcal{X}$, the realizability of $\mathcal{F}(\alpha)$ is ensured when $\alpha$ satisfies $\min_{x \in \mathcal{X}} \alpha p_{U,T}(x) < \min_{x \in \mathcal{X}} \pi(x)$, i.e., $\alpha < \min_x \pi(x)/\min_x p_{U,T}(x)$, in which case we may set a $p_F$ such that $p_T(x) = \frac{1}{1-\alpha}(\pi(x) - \alpha p_{U,T}(x))$. As an example, consider the set generation task, the details of which are provided in Section 4. There, $p_U$ induces an uniform distribution over $\mathcal{X}$ and we may set $\alpha = N/2 \min_x \pi(x) < \min_x \pi(x)/\min_x p_{T,U}(x)$. Importantly, our analysis is not considering the (limited) expressivity of the chosen parametric model for the policy network, which touches on a mostly open problem in the deep learning literature. Rather, we are concerned with the *feasibility* of finding a transition policy $\tilde{p}_F$ *consistent* and *compatible* with the given target distribution $R$, in the sense of Bengio et al. (2023, Definition 4, Definition 20).

### D.5 Proof of Theorem 5.2

We first show that the risk function is bounded. Then, Equation 7 follows directly from Maurer (2004, Theorem 5) and Jensen's inequality. Under these conditions, notice that

$$\text{KL}(p_B||p_F) = \mathbb{E}_{\tau \sim p_B}[\log p_B(\tau)] - \mathbb{E}_{\tau \sim p_B}[\log p_F(\tau)] = -H[p_B] - \mathbb{E}_{\tau \sim p_B}[\log p_F(\tau)]. \quad (40)$$

Also, by definition,

$$\begin{aligned}
p_F(\tau) &= \prod_{(s,s') \in \tau} p_F(s'|s) \\
&= \prod_{(s,s') \in \tau} (\alpha p_U(s'|s) + (1-\alpha)p_F(s'|s)) \\
&\geq \alpha^{|\tau|} \prod_{(s,s') \in \tau} \frac{1}{|\text{Ch}(s)|} \geq \left(\frac{\alpha}{\max_{s \in \tau} |\text{Ch}(s)|}\right)^{|\tau|},
\end{aligned} \quad (41)$$

and, consequently,

$$-\mathbb{E}_{\tau \sim p_B}[\log p_F(\tau)] \leq -\min_\tau |\tau| \log\left(\frac{\alpha}{\max_{s \in \tau} |\text{Ch}(s)|}\right) = \underbrace{\max_\tau |\tau| \log\left(\frac{\max_{s \in \tau} |\text{Ch}(s)|}{\alpha}\right)}_{=M_T},$$

i.e., $\text{KL}(p_B||p_F) \leq -H[p_B] + M_T$. In conclusion, the convexity of the KL divergence along with the the fact that $p_T$ and $\pi$ are respectively convex functions of $p_F$ and $p_B$ imply that $\text{KL}(\pi||p_T) \leq \text{KL}(p_B||p_F)$. The rest follows from Maurer (2004, Theorem 5) applied to $\text{KL}(p_B||p_F)$.

### D.6 Proof of Lemma B.2

The result follows directly from the Markov property of the MDP. Equivalently, we note that

$$\begin{aligned}
\mathbb{E}_{S_1,\ldots,S_t}\left[\sum_{1 \leq i \leq S} \mathcal{L}_{\text{DB}}(S_i, S_{<i})\right] &= \sum_{1 \leq i \leq t} \mathbb{E}_{S_1,\ldots,S_i}[\mathcal{L}_{\text{DB}}(S_i, S_{i-1})] \\
&= \sum_{1 \leq i \leq t} \mathbb{E}_{S_{i-1} \sim p_T^{(e)}}\left[\mathbb{E}_{S_i \sim p_e(\cdot|S_{i-1})}[\mathcal{L}_{\text{DB}}(S_i, S_{i-1})|S_{i-1}]\right].
\end{aligned} \quad (42)$$

### D.7 Proof of Theorem 5.4

Our proof has three main ingredients. Firstly, we build upon a Azuma-Hoeffding-type inequality to bound the expected transition-level error with the observed empirical error. Secondly, we derive

a trajectory-level bound of the transition-level results by relying on McAllester's linear PAC-Bayes inequality. Thirdly, we combine these results with a standard union bound argument. To start with, Beygelzimer et al. (2011, Theorem 1) shows that the martingale $M_t = \sum_{1 \leq i \leq t} M(S_i, S_{i-1})$ defined above, with $A_i \leq M(S_i, S_{i-1}) \leq B_i$ and $B_i - A_i \leq C$, satisfies

$$\mathbb{E}\left[\exp\left\{\lambda M_t - (e-2)\lambda^2 V_t\right\}\right] \leq 1, \tag{43}$$

in which $V_t = \sum_{1 \leq i \leq t} M(S_i, S_{i-1})^2$ and $\lambda \in [0, {}^1\!/C]$. In our context, $|M(S_i, S_{i-1})| \leq 2U$ by the triangle inequality, and we can take $C = 2U$. By assumption, $V_t \leq K$ for all $t \leq t_m$.

Then, for the martingale $M_t(\theta)$ and corresponding $V_t(\theta)$, with $\theta$ representing the parameters of the forward policy, by Donsker-Varadhan's variational formula, we notice that

$$\mathbb{E}_{\theta \sim P}\left[\lambda M_t(\theta) - (e-2)\lambda^2 V_t(\theta)\right] \leq \mathrm{KL}(P\|Q) + \log \mathbb{E}_{\theta \sim Q}\left[\exp\left\{\lambda M_t(\theta) - (e-2)\lambda^2 V_t(\theta)\right\}\right].$$

Similarly to (Seldin et al., 2012b, Theorem 1), let $\delta_t = {}^\delta\!/2t_m$, so that $\sum_{1 \leq t \leq t_m} \delta_t = {}^\delta\!/2$. Then, by Markov's inequality, with probability at least $1 - \delta_t$,

$$\mathbb{E}_{\theta \sim Q}\left[\exp\left\{\lambda M_t(\theta) - (e-2)\lambda^2 V_t(\theta)\right\}\right] \leq \frac{1}{\delta_t}\mathbb{E}\left[\mathbb{E}_{\theta \sim Q}\left[\exp\left\{\lambda M_t(\theta) - (e-2)\lambda^2 V_t(\theta)\right\}\right]\right];$$

where the outer expectation is with respect to the joint distribution of $\{S_1, \ldots, S_t\}$. By Tonelli's theorem and Equation 43, the right-hand side of the equation above satisfies

$$\frac{1}{\delta_t}\mathbb{E}\left[\mathbb{E}_{\theta \sim Q}\left[\exp\left\{\lambda M_t(\theta) - (e-2)\lambda^2 V_t(\theta)\right\}\right]\right] = \frac{1}{\delta_t}\mathbb{E}_{\theta \sim Q}\mathbb{E}\left[\exp\left\{\lambda M_t(\theta) - (e-2)\lambda^2 V_t(\theta)\right\}\right]$$

$$\leq \frac{1}{\delta_t} \leq \frac{2t_m}{\delta}.$$

Consequently, by Donsker-Varadhan's formula applied to $\lambda M_t(\theta) - (e-2)\lambda^2 V_t(\theta)$, bounding its exponential moment as above, and a union bound over $t$, yields with probability at least $1 - \frac{\delta}{2}$,

$$\mathbb{E}_{\theta \sim P}\left[M_t(\theta)\right] \leq (e-2)\lambda \mathbb{E}_{\theta \sim P}\left[V_t(\theta)\right] + \frac{\mathrm{KL}(P\|Q) + \log t_m + \log \frac{2}{\delta}}{\lambda}. \tag{44}$$

Hence, by the definition of $M_t(\theta)$ and the bounded-variance assumption,

$$\mathbb{E}_{\theta \sim P}\left[\frac{1}{t}\sum_{1 \leq i \leq t}\mathbb{E}[\mathcal{L}_{\mathrm{DB}}(S_i, S_{i-1})|S_{<i}]\right] \leq \frac{1}{t}\sum_{1 \leq i \leq t}\mathcal{L}_{\mathrm{DB}}(S_i, S_{i-1})$$
$$+ (e-2)\lambda \cdot \frac{K}{t} + \frac{\mathrm{KL}(P\|Q) + \log t_m + \log 2/\delta}{t\lambda}. \tag{45}$$

Nextly, let $S_1^{(j)}$ be independent samples from a forward policy $p_F(\cdot|s_o)$ for $1 \leq j \leq n$ and $\{S_1^{(j)}, \ldots, S_{t_j}^{(j)}\}$ be the correspondingly observed trajectories. Also, we recall that

$$\mathcal{L}(\theta) = \mathbb{E}_{S_1, S_2, \ldots, S_t}\left[\frac{1}{t}\sum_{1 \leq i \leq t}\mathbb{E}\left[\mathcal{L}_{\mathrm{DB}}(S_i, S_{i-1})|S_{<i}\right]\right] \tag{46}$$

and define

$$\hat{L}(\theta) = \frac{1}{n}\sum_{1 \leq j \leq n}\frac{1}{t_j}\sum_{1 \leq i \leq t_j}\mathbb{E}\left[\mathcal{L}_{\mathrm{DB}}(S_i^{(j)}, S_{i-1}^{(j)})|S_{<i}^{(j)}\right]; \tag{47}$$

the inner expectations are computed with respect to the Markovian data-generating process (recall that the conditional expectation $\mathbb{E}[\mathcal{L}_{\mathrm{DB}}(S_i, S_{i-1})|S_{<i}]$ is a random variable). By assumption, $\mathcal{L}(\theta) \leq U$. Hence, McAllester's linear PAC-Bayes inequality (McAllester, 2013, Theorem 2) entails, with probability at least $1 - \frac{\delta}{2}$ over draws of $\{S_1, \ldots, S_t\}$,

$$\mathbb{E}_{\theta \sim P}\left[\mathcal{L}(\theta)\right] \leq \frac{1}{\beta}\mathbb{E}_{\theta \sim P}\left[\hat{L}(\theta)\right] + \frac{U}{2\beta(1-\beta)} \cdot \frac{\mathrm{KL}(P\|Q) + \log 2/\delta}{n}. \tag{48}$$

Under these conditions, equations 45 and 48 jointly imply that, by a standard union-bound argument,

$$
\mathbb{E}_{\theta \sim P}\left[\mathcal{L}(\theta)\right] \leq \frac{1}{\beta}\mathbb{E}_{\theta \sim P}\left[\frac{1}{n}\sum_{1 \leq j \leq n}\left(\frac{1}{t_j}\sum_{1 \leq i \leq t}\mathcal{L}_{\mathrm{DB}}(S_i^{(j)}, S_{i-1}^{(j)}) + \frac{(e-2)\lambda K}{t_j} + \right.\right.
$$
$$
\left.\left.\frac{\mathrm{KL}(P||Q) + \log t_m + \log 2/\delta}{t_j \lambda}\right)\right] + \frac{U}{2\beta(1-\beta)} \cdot \frac{\mathrm{KL}(P||Q) + \log 2/\delta}{n}
$$

with probability $1 - \delta$ over draws of $(S_1, \ldots, S_t)$. Since $nt_j \geq T$ for all $t_j$, as we observe $T$ transitions ($n$ is between $\lfloor T/t_{min}\rfloor$ and $\lceil T/t_m \rceil$, with $t_{min}$ being the minimum length of a complete trajectory), and $t_j \leq t_m$, as $t_m$ is the trajectory's maximum lenght, the result above is equivalent to

$$
\mathbb{E}_{\theta \sim P}[\mathcal{L}(\theta)] \leq \frac{1}{\beta}\mathbb{E}_{\theta \sim P}\underbrace{\left[\frac{1}{n}\sum_{1 \leq j \leq n}\left(\frac{1}{t_j}\sum_{1 \leq i \leq t}\mathcal{L}_{\mathrm{DB}}(S_i^{(j)}, S_{i-1}^{(j)})\right)\right]}_{=\hat{\mathcal{L}}(\theta)} + \frac{(e-2)\lambda K}{T} +
$$
$$
\frac{\mathrm{KL}(P||Q) + \log t_m + \log 2/\delta}{T\lambda} + \frac{U}{2\beta(1-\beta)} \cdot \frac{\mathrm{KL}(P||Q) + \log 2/\delta}{n}. \tag{49}
$$

By aggregating the terms corresponding to $\mathrm{KL}(P||Q)$ and $\log 2/\delta$, we derive the desired upper bound on the expected risk of the DB loss.

### D.8    PROOF OF THEOREM C.1

Intuitively, when each balance condition is satisfied, each state $s$ on $\mathcal{I}_j$ is sampled in proportion to $F_j(s)$ and, conditioned on $s$, each terminal state will be sampled in proportion to $R(x)/F_j(s)$, implying that, marginally, each $x$ is sampled proportionally to $R(x)$. In the following, we make this argument rigorous. We first consider the case in which $x \in \mathcal{X} \setminus \mathcal{S}_o$. As we are assuming that $F_j(s)p_F(\tau|s) = p_B(\tau|x)R(x)$ for each trajectory $\tau$ starting at $s \in \mathcal{I}_j$ and finishing at $x$, we must conclude that

$$
p_T^j(x|s) = \sum_{\tau:\, s \rightsquigarrow x} p_F(\tau|s) = \frac{R(x)}{F_j(s)}\sum_{\tau:\, s \rightsquigarrow x} p_B(\tau|x). \tag{50}
$$

On the other hand, since $F_o(s_o)p_F^o(\tau|s) = p_B^o(\tau|s)F_j(s)$ for $s \in \mathcal{I}_j$,

$$
p_T^o(s|s_o) = \sum_{\tau:\, s_o \rightsquigarrow s} p_F(\tau|s_o) = \frac{F_j(s)}{F_o(s_o)}\sum_{\tau:\, s_o \rightsquigarrow s} p_B(\tau|s) = \frac{F_j(s)}{F_o(s_o)}, \tag{51}
$$

as the probability of reaching $s_o$ by starting from $s$ and following $p_B$ is equal to one since $s_o$ is the only sink state of the transposed state graph. In this context,

$$
\begin{aligned}
p_T(x|s_o) &= \sum_{1 \leq j \leq m}\sum_{s \in \mathcal{I}_j} p_T^j(x|s)p_T^o(s|s_o) \\
&= \sum_{1 \leq j \leq m}\sum_{s \in \mathcal{I}_j}\frac{F_j(s)}{F_o(s_o)} \cdot \frac{R(x)}{F_j(s)}\sum_{\tau:\, s \rightsquigarrow x} p_B^j(\tau|x) \\
&= \sum_{1 \leq j \leq m}\frac{R(x)}{F_o(s_o)}\sum_{1 \leq j \leq m}\sum_{s \in \mathcal{I}_j}\sum_{\tau:\, s \rightsquigarrow x} p_B^j(\tau|x) \\
&= \frac{R(x)}{F_o(s_o)}\sum_{s \in \bigcup_{1 \leq j \leq m}\mathcal{I}_j} p_B^j(\tau|s) = \frac{R(x)}{F_o(s_o)};
\end{aligned} \tag{52}
$$

i.e., $p_T(x|s_o)$ samples $x$ proportionally to $R(x)$. For the forth line above, we relied on the fact that the probability of eaching $\bigcup \mathcal{I}_j$ is equal to one when starting at $x \in \mathcal{X} \setminus \mathcal{S}_o$ and following $p_B$. Correspondingly, when $x \in \mathcal{X}$, it follows from the satisfiability of the trajectory balance condition that $p_T(x|s_o) \propto R(x)$. This ensures SAL is a sound distributed learning algorithm for GFlowNets.

### D.9 PROOF OF LEMMA C.3

The global minimizer of Equation 9 satisfies, for every $j$, $F_j(s)p_F^j(\tau) = R(x)p_B^j(\tau|x)$ for every trajectory $\tau \colon s \rightsquigarrow x$ starting at $s \in \mathcal{I}_j$ and finishing at $x \in \mathcal{X}_j$. Consequently,

$$p_T^j(x|s) = \sum_{\tau \colon s \rightsquigarrow x} p_F^j(\tau|s) = \sum_{\tau \colon s \rightsquigarrow x} \frac{p_B^j(\tau|x)R(x)}{F_j(s)} = \frac{R(x)}{F_j(s)} \sum_{\tau \colon s \rightsquigarrow x} p_B^j(\tau|s). \tag{53}$$

Similarly,

$$\sum_{\tau \colon s \rightsquigarrow x} F_j(s)p_F^j(\tau|s) = \sum_{\tau \colon s \rightsquigarrow x} p_B^j(\tau|x)R(x) \tag{54}$$

implies that

$$F_j(s) = \sum_{\tau \colon s \rightsquigarrow x} p_B^j(\tau|x)R(x) \tag{55}$$

since $\sum_{\tau \colon s \rightsquigarrow x} p_F^j(\tau|s)$ for every $s$. These equations jointly entail the proposition.

### D.10 PROOF OF PROPOSITION C.5

As in the demonstrations above, we consider two cases in separate. First, when $x \in \mathcal{X} \cap \mathcal{S}_o$, then $p_T(x) = R(x)/Z_F + Z_R$ due to the satisfiability of the balance condition by the model. Hence,

$$\begin{aligned} \sum_{x \in \mathcal{X} \cap \mathcal{S}_o} |p_T(x) - \pi(x)| &= \sum_{x \in \mathcal{X} \cap \mathcal{S}_o} \left| \frac{R(x)}{Z_F + Z_R} - \frac{R(x)}{Z} \right| \\ &= \left| \frac{1}{Z_F + Z_R} - \frac{1}{Z} \right| \sum_{x \in \mathcal{X} \cap \mathcal{S}_o} R(x) = \left| \frac{1}{Z_F + Z_R} - \frac{1}{Z} \right| Z_R. \end{aligned} \tag{56}$$

Second, when $x \in \mathcal{X} \setminus \mathcal{S}_o$, we note that

$$\begin{aligned} p_T(x) &= \sum_{1 \le j \le m} \sum_{s \in \mathcal{S}_j} \sum_{\tau \colon s_o \rightsquigarrow s \rightsquigarrow x} p_F(\tau|s_o) \\ &= \sum_{1 \le j \le m} \sum_{s \in \mathcal{S}_j} \left( \sum_{\tau \colon s_o \rightsquigarrow s} p_F^o(\tau|s_o) \right) \left( \sum_{\tau \colon s \rightsquigarrow x} p_F^j(\tau'|s) \right) \\ &= \sum_{1 \le j \le m} \sum_{s \in \mathcal{S}_j} p_T^o(s)p_T^j(x|s) = \sum_{1 \le j \le m} \sum_{s \in \mathcal{S}_j} \frac{F_j(s)}{Z_F + Z_R} \cdot p_T^j(x|s). \end{aligned} \tag{57}$$

Similarly, for any $\mathcal{X}$-valued function $f$,

$$f(x) = \sum_{1 \le j \le m} \sum_{s \in \mathcal{S}_j} \frac{F_j(s)}{Z_F} \cdot f(x); \tag{58}$$

hence,

$$\begin{aligned} \pi(x) - p_T(x) &= \sum_{1 \le j \le m} \sum_{s \in \mathcal{S}_j} \left( \frac{F_j(s)}{Z_F} \cdot \pi(x) - \frac{F_j(s)}{Z_F + Z_R} \cdot p_T^j(x|s) \right) \\ &= \sum_{1 \le j \le m} \sum_{s \in \mathcal{S}_j} \frac{F_j(s)}{Z_F} \left( \pi(x) - \frac{Z_F}{Z_F + Z_R} p_T^j(x|s) \right) \\ &= \mathbb{E}_{s \sim p_{T, \setminus \mathcal{X}}^o} \left[ \left( \pi(x) - \frac{Z_F}{Z_F + Z_R} p_T^{f(s)}(x|s) \right) \right]. \end{aligned} \tag{59}$$

By recalling that $\mathrm{TV}(\pi, p_T) = \frac{1}{2} \left( \sum_{x \in \mathcal{X}} |\pi(x) - p_T(x)| \right)$, this result, along with Equation 56 and Jensen's inequality applied to the function $x \mapsto |x|$, implies the proposition. To further strengthen our intuition, we also consider directly bounding the accuracy of $\mathcal{G}_o$ as a function of the trajectory-level inaccuracies of each $\mathcal{G}_j$. For this, we re-write $\pi(x)$ as

$$\pi(x) = \sum_{1 \le j \le m} \sum_{s \in \mathcal{S}_j} \pi(x) \sum_{\tau \colon s \rightsquigarrow x} p_B^j(\tau|x). \tag{60}$$

Correspondingly, by recalling the property $p_T(x) = \sum_{1 \leq j \leq m} \sum_{s \in \mathcal{S}_j} p_T^o(s) p_T^j(x|s)$, we conclude

$$
\begin{aligned}
|\pi(x) - p_T(x)| &= \left| \sum_{1 \leq j \leq m} \sum_{s \in \mathcal{S}_j} \sum_{\tau:\, s \rightsquigarrow x} \pi(x) p_B(\tau|x) - p_F(\tau|s) p_T^o(s) \right| \\
&\leq \underbrace{\sum_{1 \leq j \leq m} \sum_{s \in \mathcal{S}_j} \sum_{\tau:\, s \rightsquigarrow x} \left| \pi(x) p_B^j(\tau|x) - p_F^j(\tau|s) p_T^o(s) \right|}_{\text{Error associated to the } j\text{th client}}.
\end{aligned}
\tag{61}
$$

Hence, the total variation distance between $\pi$ and $p_T$ is bounded above by

$$
\text{TV}(\pi, p_T) = \frac{Z_R}{2} \left| \frac{1}{Z} - \frac{1}{Z_R + Z_F} \right| + \frac{1}{2} \sum_{x \in \mathcal{X} \backslash \mathcal{S}_o} \underbrace{\sum_{1 \leq j \leq m} \sum_{s \in \mathcal{S}_j} \sum_{\tau:\, s \rightsquigarrow x} \left| \pi(x) p_B^j(\tau|x) - p_F^j(\tau|s) p_T^o(s) \right|}_{\text{Error associated to the } j\text{th client}}.
$$

For tree-shaped state graphs, the second term of the equation above can be significantly simplified by noticing that (i) each $x$ is uniquely associated to a $j$, a relationship which we denote by $g(x) = j$, and (ii) that $p_B^j(\tau|x) = 1$ and $p_F^j(\tau|s) = p_T^j(x|s)$. Under these conditions,

$$
\text{TV}(\pi, p_T) \leq \frac{Z_R}{2} \left| \frac{1}{Z} - \frac{1}{Z_R + Z_F} \right| + \frac{1}{2} \sum_{x \in \mathcal{X} \backslash \mathcal{S}_o} \underbrace{\sum_{s \in \mathcal{S}_{g(x)}} \left| \pi(x) - p_T^{g(x)}(x|s) p_T^o(s) \right|}_{\text{Error associated to the } j=g(x)\text{th model}}.
\tag{62}
$$

## D.11 PROOF OF PROPOSITION C.6

We proceed by strong induction on the number $k$ of fixed-horizon partitions. For $k = 1$, the result above is equivalent to Equation C.1. Assume, then, that the statement holds for $j$ fixed-horizon partitions of the state graph for all $j < k$. Let $\mathcal{G}_i$, $0 \leq i \leq k$, be a sequence of GFlowNets satisfying the amortized trajectory balance condition. By induction, each $x \in \bigcup_{1 \leq j \leq k-1} \mathcal{X}_j$ is sampled proportionally to $\sum_{1 \leq j \leq k-1} \mathbf{1}[x \in \mathcal{X}_j] R_j(s)$. In particular, if $x \in \mathcal{X} \cup \bigcup_{1 \leq j \leq k-1} \mathcal{X}_j$, then $x$ is sampled proportionally to $R(x)$. For what remains, let $x \in \mathcal{X} \setminus \bigcup_{1 \leq j \leq k-1} \mathcal{X}_j$. Hence, for each state $s \in \bigcup_{1 \leq j \leq m_k} \mathcal{I}_{k,j} \subseteq \mathcal{X}_{k-1}$ and each trajectory $\tau \colon s \rightsquigarrow x$,

$$
F_k(s) p_F(\tau|s) = p_B(\tau|x) R(x),
\tag{63}
$$

i.e., $p_F(\tau|s) = {p_B(\tau|x) R(x)}/{F_k(s)}$. Thus, by marginalizing out the non-terminal components of $\tau$,

$$
p_T(x|s) = \frac{R(x)}{F_k(s)} \sum_{\tau:\, s \rightsquigarrow x} p_B(\tau|x)
\tag{64}
$$

and, since each $s$ is sampled proportionally to $R_{k-1}(s) := F_k(s)$,

$$
\begin{aligned}
p_T(x) &\propto \sum_{s \in \bigcup_{1 \leq j \leq m_k} \mathcal{I}_{k,j}} p_T(x|s) F_k(s) \\
&= \sum_{s \in \bigcup_{1 \leq j \leq m_k} \mathcal{I}_{k,j}} F_k(s) \cdot \frac{R(x)}{F_k(s)} \sum_{\tau:\, s \rightsquigarrow x} p_B(\tau|x) \\
&= R(x) \underbrace{\sum_{s \in \bigcup_{1 \leq j \leq m_k} \mathcal{I}_{k,j}} \sum_{\tau:\, s \rightsquigarrow x} p_B(\tau|x)}_{=1} = R(x).
\end{aligned}
\tag{65}
$$

This ensures that each $x \in \mathcal{X} \setminus \bigcup_{1 \leq j \leq k-1} \mathcal{X}_j$ is sampled proportionally to $R(x)$. By induction, each $x \in \mathcal{X}$ is sampled proportionally to $R(x)$. Hence, the recursive instance of SAL is a sound approach for sampling objects proportionally to a reward function.

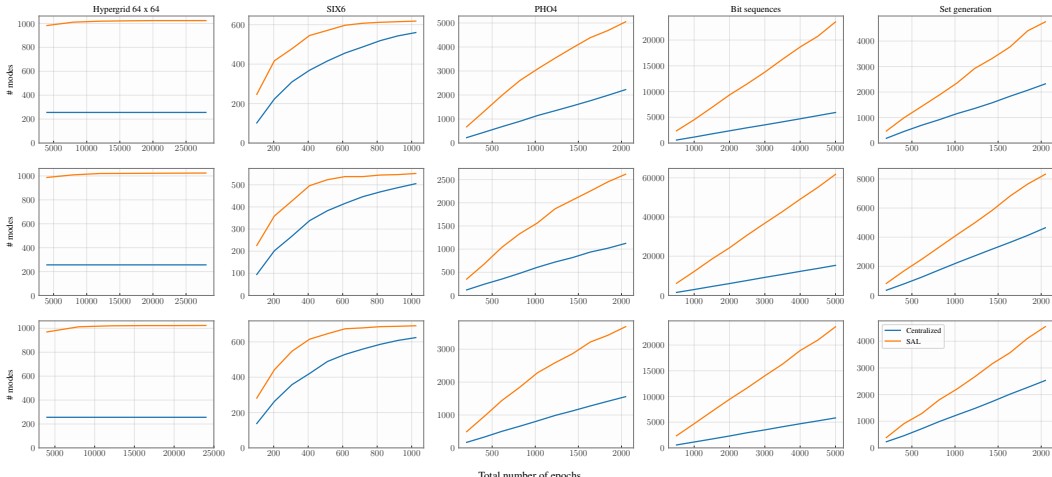

Figure 13: Complementary results for Figure 7 with different random seeds. Notice that, except for the hypergrid task, the threshold defining a mode is a random variable, which explains the variability (albeit consistency) of the number of modes found within the same column.

## E    LIMITATIONS AND FUTURE WORKS

We presently discuss the limitations of our work and the consequent opportunities for future research. Although informative, our theoretical analysis was limited to the case of i.i.d. sampled trajectories which, as highlighted in Section 5, does not necessarily reflect the nature of usual strategies for learning GFlowNets, e.g., $\epsilon$-greedy sampling. Indeed, the typical training of GFlowNets is closer in spirit to active learning (AL) (Cohn et al., 1994; Gal et al., 2017) in the sense that a batch of trajectories is sampled from a policy that is dynamically updated as more data points are observed, and it would be interesting and important to pursue an investigation in this direction (Jain et al., 2022; Malik et al., 2023). In contrast to AL, however, there is no explicit acquisition function guiding GFlowNet training. On the other hand, Deleu & Bengio (2023)'s interpretation of GFlowNets as Markov chains in the trajectory-space could be used together with Azuma's inequality (Azuma, 1967), in the fashion of Theorem 5.4, as a useful starting point for this. More specifically, one could consider that a sequence of trajectories $\{\tau_t\}_{t\geq 1}$ is observed during training and use the same techniques enabling the proof of Theorem 5.4, namely, constructing a martingale difference sequence $M_t = \mathcal{L}_{\mathrm{TB}}(\tau_t) - \mathbb{E}_{\tau_t}[\mathcal{L}_{\mathrm{TB}}(\tau_t)|\tau_{s<t}]$ and applying Theorem 1 of Seldin et al. (2012a) both within and between trajectories, the results of which would then be unified via an union bound argument.

Additionally, the promising results of SAL pave the road to a range of interesting investigations. Most promitently, we believe the development of principled partitioning methods can lead to substantial improvements in scaling GFlowNet training. Although our discussion is constrained to fixed-horizon partitions for easeness of exposition and implementation, the algorithm could in principle be extended to more general settings. Theorem 5.2 suggests that a good partition would ensure that the within- and between-partition distributions are close to uniform. Intuitively, we would like that the target distribution of both the leaf and root GFlowNets are relatively simple to approximate when compared against the original target and that the most important regions of the state space, as measured by the reward function, are appropriately covered. The best way to ensure these properties, however, remains an open problem and we think it is a promising venue for future endeavors.

### E.1    ADDITIONAL EXPERIMENTS

**Robustness of SAL with respect to the FHP's size.** It is intuitively clear that an increase in the number $m$ of partitions in a FHP would improve the coverage of the state graph and accelerate mode discovery. In doing so, however, we also enlarge the memory cost of the algorithm due to the necessity of aggregating a larger number of leaf GFlowNets in the server. To shed light on the effect of $m$ on SAL's performance, Figure 14 presents the number of modes found during training for the tasks of SIX6 and PHO4. As anticipated, SAL drastically improves upon a centralized GFlowNet,

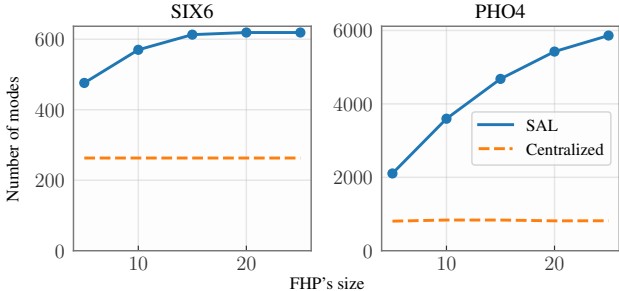

Figure 14: **SAL consistently outperforms a centralized GFlowNet** irrespective of the number of components defining the underlying FHP. As expected, the coverage of the state graph is an increasing function of the FHP's size. Results for the centralized model are included solely for comparison.



Figure 17: **Extremely inaccurate estimation of the flow function** might lead to mode collapse. (left) Standard SAL-trained GFlowNet for a $8 \times 8$ grid with 2-sized FHP. (middle) SAL-trained GFlowNet when the flow function $F_1$ is severely inaccurate. (right) Target distribution.

notably regardless of the size of the underlying FHP. Overall, our experiments throughout this work indicated that SAL is notably robust to the choice of hyperparameters defining its implementation.

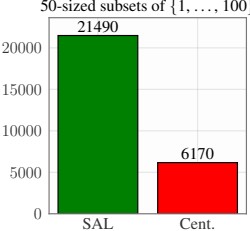

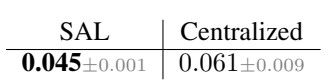

| SAL | Centralized |
|---|---|
| **0.045**±0.001 | 0.061±0.009 |

Table 3: SAL improves upon a centralized GFlowNet for large-scale set generation in terms of FCS (Silva et al., 2024).

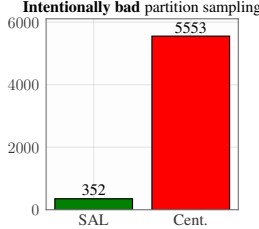

Figure 15: SAL leads to significantly faster mode discovery for large-scale set generation. Results averaged across 3 runs.

Figure 16: SAL underperforms when the within-subgraph sampling distribution is inadequately designed.

**Experiments on large-scale set generation tasks** ($|\mathcal{X}| \approx 10^{30}$)**.** To underscore the scalability of SAL with respect to the underlying domain's size, Figure 15 and Table 3 respectively show that our distributed algorithm entails faster mode discovery and lead to a more accurate distributional approximation for the problem of generating 50-sized subsets of $\{1, \ldots, 100\}$; see Section A.3 for a definition of this task. For a fair comparison, the centralized GFlowNet is trained for 60 seconds — twice the time allocated to each leaf GFlowNet and to the aggregation step. Besides, we adopt the experimental setup described in Section B. Importantly, these results suggest that the applicability of SAL is not limited by the domain sizes presented in the main text.

In the following paragraphs, we examine two potential causes for catastrophic failures of the aggregated model, as mentioned in Proposition C.5 on Section C.3.

**Insufficient training of a leaf GFlowNet.** As the aggregation phase in SAL relies on the locally estimated flows as a surrogate reward for the root GFlowNet (recall Algorithm 1), a natural driver of catastrophic failures is an inaccurately approximated flow function by a leaf GFlowNet. In particular, if a local flow $F_j$ is significantly larger than its correct value, the resulting model might allocate a substantial probability mass to the subgraph associated to the $j$th leaf GFlowNet. In this case, high-probability regions of the remaining subgraphs might be completely missed by the global GFlowNet. We illustrate this effect in Figure 17, which shows the learned distribution over a $8 \times 8$ grid for (i) a SAL-trained GFlowNet and (ii) a SAL-trained GFlowNet with a substantially over-estimated $F_1$. To emulate (ii), we train each leaf GFlowNet in a standard fashion and, during the aggregation phase, multiply the learned $F_1$ by 100. In both cases, we considered 2-sized FHPs and followed the experimental setup of Section 6.2. Notably, Figure 17 confirms that a severely inaccurate flow function $F_j$ may lead to a misrepresentation of the target distribution's high-probability regions by the global GFlowNet. In spite of these results, we stress that we *did not observed this pathological behavior throughout our experiments* in the main text and in Section C.

**Insufficiently diverse within-subgraph sampling distributions.** Recall in Definition 6.2 that SAL depends on a distribution $q_j$ over the initial states $\mathcal{I}_j$ of the $j$th subgraph defining the FHP; see Figure 3. Clearly, when $q_j$'s probability mass is overly concentrated in a relatively small subset of $\mathcal{I}_j$, the corresponding leaf GFlowNet might fail to accurately learn from the target distribution due a restricted exploration of the state graph — this was observed in Section 4. Also, the efficiency of mode discovery during training would be significantly hindered. In this context, Figure 16 illustrates the mode discovery rate for the task of generating 16-sized subsets of $\{1, \ldots, 32\}$ with the reward function described in Section A when each $q_j$ is a truncated Poisson distribution with mean equal to $^1/_{5000}$-th of the size of the corresponding $\mathcal{I}_j$. Importantly, similarly to Figure 17, this is an extreme corner case that did not pose an issue in our experiments. In practice, we suggest setting $q_j$ as an uniform distribution to maximize the diversity of the explored partitions.

