# OpenReview forum: "Generalization and Distributed Learning of GFlowNets"
_ICLR.cc/2025/Conference — ICLR 2025 Poster_

### Official Review · Reviewer_76jg · 2024-10-30

**Soundness:** 3
**Presentation:** 3
**Contribution:** 3
**Rating:** 6
**Confidence:** 2

**Summary:**

This paper investigates the Generative Flow Networks (GFlowNets), and presents several novel contributions to the understanding and development of GFlowNets: (i) Data-Dependent Generalization Bounds, which is the first data-dependent generalization bounds for GFlowNets, providing statistical guarantees for their learning capabilities; (ii) Impact of State Space Size,  which explores the influence of state space size on generalization performance, utilizing PAC-Bayesian inequalities to formalize these effects. (iii) Subgraph Asynchronous Learning (SAL) Algorithm, which is proposed to improve training efficiency and generalization in GFlowNets by partitioning the network into smaller subnetworks.

**Strengths:**

- This paper offers a unique perspective on GFlowNet generalization through PAC-Bayesian bounds.
- This paper is theoretically solid and well-orginazied.
- The authors validate their findings with experiments on both synthetic and real-world tasks, strengthening the reliability of the conclusions.

**Weaknesses:**

- The main idea of SAL (i.e., divide and conquer) is novel for GFlowNet. However, I wonder whether the aggregation accuracy will be compromised for large networks. In other words, is there a trade-off in the theoretical results about $m$?
- The time cost of the Fixed-horizon partition does not appear to be addressed. Does this operation impact model efficiency? More precisely, how can a Fixed-horizon partition be found efficiently?

**Questions:**

See it in the Weakness above.

---

> ### Author Response · Authors · 2024-11-20
>
> Thank you for attentively reviewing and appreciating our work. If our answers below  appropriately address your concerns, please consider raising your score. Otherwise, we will be more than happy to discuss our contributions in further detail.
>
> > The main idea of SAL (i.e., divide and conquer) is novel for GFlowNet. However, I wonder whether the aggregation accuracy will be compromised for large networks. In other words, is there a trade-off in the theoretical results about $m$?
>
> Thank you for the interest. By Theorem C.1 and Proposition C.5, a GFlowNet learned via SAL is accurate when its corresponding leaf models are adequately learned. In the limit, when each learning objective is globally minimized, Theorem C.1 ensures that the resulting GFlowNet samples from the right distribution — irrespective of $m$.
>
> In this context, for large $m$, two practical problems might arise. Statistically, a larger $m$ could increase the likelihood of catastrophic error propagation from an insufficiently trained leaf GFlowNet to the global GFlowNet. However, we did not observe this issue throughout our experiments. Computationally, enlarging $m$ would lead to a potential memory overhead when carrying out the aggregation phase. Nonetheless, most applications of GFlowNets are built upon small-scale models such as MLPs [1, 2] and GNNs [3, 4]. Hence, the memory cost of SAL is typically not a critical problem.
>
> [1] Deleu et al. Joint Bayesian Inference of Graphical Structure and Parameters with a Single Generative Flow Network. NeurIPS 2023.
>
> [2] Pan et al. Generative Augmented Flow Networks. ICLR 2023.
>
> [3] Bengio et al. Flow Network based Generative Models for Non-Iterative Diverse Candidate Generation. NeurIPS 2021
>
> [4] Zhang et al. Let the Flows Tell: Solving Graph Combinatorial Optimization Problems with GFlowNets. NeurIPS 2023.
>
> > The time cost of the Fixed-horizon partition does not appear to be addressed. Does this operation impact model efficiency? More precisely, how can a Fixed-horizon partition be found efficiently?
>
> Thank you for the question. In practice, the FHP is only implicitly defined by an assignment function and need not be explicitly constructed during either training or inference. Also, the FHP is problem-dependent and  fixed prior to training. As a consequence, there is no additional computational overhead related to the construction of the FHP.
>
> We thank you again for the review. We hope that our answers elevate your appraisal of our work. Should any clarifications be needed, we would be glad to discuss our contributions in more detail.

---

### Official Review · Reviewer_NTon · 2024-11-02

**Soundness:** 3
**Presentation:** 3
**Contribution:** 3
**Rating:** 8
**Confidence:** 3

**Summary:**

The authors of this paper investigate the generalization properties of Generative Flow Networks (GFlowNets), focusing on providing theoretical guarantees for their performance. They introduce the first data-dependent generalization bounds for GFlowNets, leveraging PAC-Bayesian inequalities to analyze how the state space size affects generalization. Building on these theoretical insights, they propose a distributed learning algorithm, which employs a divide-and-conquer strategy to train multiple GFlowNets on subgraphs and then aggregate them to improve mode coverage and distribution matching.

**Strengths:**

1. The authors address the gap in the theoretical understanding of GFlowNets, namely, their generalization properties, by deriving the first data-dependent generalization bounds, which is a novel and impactful contribution. Traditional analyses of generalization have primarily focused on models like neural networks or reinforcement learning agents, so extending this to GFlowNets represents a fresh and innovative application of statistical learning theory.

2. By proposing a divide-and-conquer approach, where multiple GFlowNets are trained on different subgraphs of the flow network and then aggregated through a coordinating GFlowNet, the authors introduce a novel parallel training strategy that could address scalability challenges.

3. The quality of the research presented is high, with clear efforts to rigorously derive and justify the theoretical claims. By framing the generalization bounds in terms of the state space size, the authors have identified a key factor that impacts GFlowNets' generalization capabilities and provided a principled way to analyze it.

4. The structure of the paper is logical, and the methodology and experiments are presented in a well-organized manner.

5. The significance of this work is substantial. GFlowNets have shown empirical promise for a variety of applications, including distribution matching and probabilistic modeling, yet a rigorous understanding of their generalization capabilities has been lacking. By providing data-dependent generalization bounds, this paper addresses a core theoretical question that could impact how GFlowNets are used and optimized in the future.

6. The divide-and-conquer approach of SAL not only enhances scalability but also has potential benefits for robustness and mode coverage in the generated distributions. These improvements could make GFlowNets more viable for applications in domains like natural language processing, biology, and engineering, where large datasets and complex distributions are the norm.

**Weaknesses:**

1. While the paper presents experimental results for both synthetic and real-world tasks, the scope of these experiments might be limited in terms of diversity of datasets, complexity of environments, and breadth of tasks. Here are some suggestions:

(i) Combinatorial Optimization over Graphs
Dataset: TSP, MAX-CUT, or Graph Coloring datasets commonly used in combinatorial optimization.
Task: Solve classic graph-based optimization problems, where the state space represents possible solutions (e.g., routes, cuts, or colorings) and the reward function evaluates the quality of each solution. For instance, in a MAX-CUT problem, the reward would be based on the cut size.
Importance for Distribution Learning: For combinatorial optimization tasks, covering a variety of valid solutions (e.g., different feasible paths with minimal distance in a graph problem) is crucial. Testing SAL here would show whether it can capture a distribution over multiple near-optimal solutions.

(ii) Financial Portfolio Optimization
Dataset: Historical stock or cryptocurrency price data.
Task: Generate diverse portfolio allocations that balance risk and reward, aiming to maximize expected returns with minimal risk. Rewards could be based on risk-adjusted performance metrics, like the Sharpe ratio.
Importance for Distribution Learning: Portfolio generation requires balancing exploration of different investment strategies with optimization for high returns. Distribution learning helps to capture diverse investment strategies and avoid over-optimizing for a narrow range of allocations, making it a good test for SAL.

(iii) Neural Architecture Search (NAS)
Dataset: CIFAR-10 or ImageNet for evaluating architectures.
Task: Generate neural network architectures that maximize performance on a validation set. Rewards could be based on model accuracy, computational efficiency, or other architecture characteristics.
Importance for Distribution Learning: NAS requires exploration across a diverse set of neural architectures rather than converging on a single high-performing design. This task would test whether SAL can generalize across different architectural choices and find multiple good architectures, covering a broad distribution in the search space.

2. The SAL approach, while promising for distributed training, may have significant computational and memory overhead due to the need to train multiple GFlowNets in parallel and then aggregate their results. This could be a limiting factor in resource-constrained environments or for applications where computational efficiency is critical.

3.  The generalization bounds do not take into account potential dependencies in the data that could arise in scenarios where GFlowNets are used for tasks like sequential decision-making. These dependencies could weaken the validity of the i.i.d. assumption and potentially affect the reliability of the bounds. It will be useful to discuss how the authors might extend their analysis to account for dependencies in sequential decision-making scenarios.

4. Although SAL is designed to improve mode coverage, aggregating multiple GFlowNets could still risk mode collapse if subgraph-trained models fail to adequately capture all modes of the target distribution. If some subgraphs do not represent certain regions of the distribution well, the aggregation process could overlook these regions, resulting in poor overall coverage.

**Questions:**

1. While the paper addresses generalization and distributed learning, it would be useful if it discussed the potential limitations of SAL more explicitly. For instance, are there scenarios where SAL might fail to converge, or does it require a careful balance of parameters to function well?

2. Additionally, are there inherent limitations to the generalization bounds derived, particularly when applied to different classes of GFlowNets or domains with very large or complex state spaces?

3. The paper notes that traditional training of GFlowNets resembles active learning. Could you expand on how you envision incorporating active learning strategies into your framework? How might this impact the performance of GFlowNets compared to the current approach?

4. In the experiments, were there any signs of mode collapse when aggregating the outputs of subgraph-trained GFlowNets? What strategies would you recommend or envision to mitigate this issue during the aggregation phase?

5. Are there empirical limits which have been identified in the experiments regarding the effectiveness of SAL as the complexity of the underlying GFlowNet increases?

6. SAL relies on fixed partitions to divide the state space into subgraphs, which may limit the flexibility and adaptability of the approach. In real-world applications, the optimal partitioning of the state space may not be obvious and may vary over time as the policy evolves. A rigid partitioning could result in subgraphs that are either too sparse (leading to inefficient training) or too dense (leading to high computational cost). How do the authors envision adapting the partitioning scheme to be more flexible or dynamic as the policy evolves during training?

---

> ### Author Response · Authors · 2024-11-20
> **Official Comment by Authors (1/4)**
>
> Thank you for the thoughtful feedback and for appreciating our work. We hope our answers resolve your concerns satisfactorily. If not, we would be happy to continue the discussion.
>
> ## Weaknesses
>
> > 1. While the paper presents experimental results for both synthetic and real-world tasks, the scope of these experiments might be limited in terms of diversity of datasets, complexity of environments, and breadth of tasks.
>
> Thank you for the suggestions. While we understand that these experiments could enhance the impact of our work, it unfortunately will not be possible to execute them during the short discussion period. Indeed, as combinatorial optimization was separately addressed by [1, 2] in the context of GFlowNets, we believe that both applications (ii) and (iii) are worthy of a work of their own. We focused our empirical analysis  on common benchmark tasks in the literature.
>
> For NAS, in particular, each leaf GFlowNet in SAL would only have to learn the final layers of the trained neural network. Nonetheless, defining an iterative generative process in the space of back-propagable neural network architectures is a challenging and unresolved issue in  the literature. As a consequence, it is mostly unclear how to implement a GFlowNet for addressing this problem.
>
> To strengthen our empirical evidence, however, we have included a large-scale experiment in Section E.1 (page 42) for generating $50$-sized subsets of $\\{1, … 100\\}$. In consistency with our prior results, SAL improves upon a centralized approach in terms of both mode discovery rate and distributional accuracy of the trained model.
>
> [1] Zhang et al. Let the Flows Tell: Solving Graph Combinatorial Optimization Problems with GFlowNets. NeurIPS 2023.
>
> [2] Zhang et al. Robust Scheduling with GFlowNets. ICLR 2023.
>
>
> > 2. The SAL approach, while promising for distributed training, may have significant computational and memory overhead due to the need to train multiple GFlowNets in parallel and then aggregate their results. This could be a limiting factor in resource-constrained environments or for applications where computational efficiency is critical.
>
> Regarding training time, we would like to highlight that SAL is always faster than centralized learning, as shown in Section C.1. In this case, the main potential bottleneck in SAL is the memory required for the aggregation step — which entails loading all leaf GFlowNets into a centralized server.
>
> In spite of this, the leading applications of GFlowNets utilize small-scale models, such as MLPs [1, 2] and GNNs [3, 4], to parameterize the policy networks. As a consequence, the memory overhead will be negligible for most practitioners. However, for applications such as NLP with  LLM-based policy networks, we acknowledge that lack of resources might be an issue.
>
> [1] Deleu et al. Joint Bayesian Inference of Graphical Structure and Parameters with a Single Generative Flow Network. NeurIPS 2023.
>
> [2] Pan et al. Generative Augmented Flow Networks. ICLR 2023.
>
> [3] Bengio et al. Flow Network based Generative Models for Non-Iterative Diverse Candidate Generation. NeurIPS 2021.
>
> [4] Zhang et al. Let the Flows Tell: Solving Graph Combinatorial Optimization Problems with GFlowNets. NeurIPS 2023.

---

> > ### Author Response · Authors · 2024-11-20
> > **Official Comment by Authors (2/4)**
> >
> > > 3. The generalization bounds do not take into account potential dependencies in the data that could arise in scenarios where GFlowNets are used for tasks like sequential decision-making. These dependencies could weaken the validity of the i.i.d. assumption and potentially affect the reliability of the bounds. It will be useful to discuss how the authors might extend their analysis to account for dependencies in sequential decision-making scenarios.
> >
> > Thank you for the opportunity to discuss potential extensions of our work. To start with, we would like to note that PAC-Bayesian analysis for non-i.i.d. data remains largely an open problem in the statistical learning literature. As far as we know, most efforts in this direction resulted in theoretical bounds for conditionally independent [1] or distributionally shifted [2] data. In the realm of sequential decision-making, we believe that Seldin’s [3] Azuma-Hoeffding-style inequalities for martingales stand out as the most promising techniques for the derivation of tight PAC-Bayesian bounds, having recently achieved impressive results in, e.g., text modeling [4].
> >
> > In particular, a GFlowNet might be characterized as a nested stochastic process (NSP). At a higher level, each batch of trajectories $\{\tau_{1}, \dots, \tau_{N}\}$ at each training iteration is sampled from a distribution influenced by the data seen up to this point (e.g., $\epsilon$-greedy). At a lower level, each trajectory $\tau$ is drawn from a Markovian decision process starting at the initial state $s_{o}$. In this sense, we think that tight and general-purpose generalization bounds for GFlowNets could be obtained by successively applying Seldin’s technique to each level of this NSP, thereby extending Theorem 5.4. We are confident that this is a fruitful research line for future endeavors.
> >
> >
> > [1] Pentina and Lambert. A PAC-Bayesian bound for Lifelong Learning. ICML 2014.
> >
> > [2] Germain et al. A New PAC-Bayesian Perspective on Domain Adaptation. ICML 2016.
> >
> > [3] Seldin et al. PAC-Bayesian Inequalities for Martingales. https://arxiv.org/abs/1110.6886
> >
> > [4] Lotfi et al. Unlocking Tokens as Data Points for Generalization Bounds on Larger Language Models. NeurIPS 2024.
> >
> >
> > > 4. Although SAL is designed to improve mode coverage, aggregating multiple GFlowNets could still risk mode collapse if subgraph-trained models fail to adequately capture all modes of the target distribution. If some subgraphs do not represent certain regions of the distribution well, the aggregation process could overlook these regions, resulting in poor overall coverage.
> >
> > You’re correct. In fact, this is a major issue of parallel algorithms for amortized inference, e.g. [1], without a clear solution. To mitigate this, we need to diagnose the goodness-of-fit of the local sampling distributions to determine if the subgraph-trained models require further training before being shared with a central server. For this, Proposition C.3 and Corollary C.4 are an important starting point for assessing the distributional accuracy of the leaf GFlowNets. Illustratively, we included a discussion of this effect in Section E.1 (page 41), alongside an illustration of a globally collapsed distribution resulting from an extremely badly estimated flow function in a leaf GFlowNet. However, we emphasize that these pathological results from the fabricated experiments were not observed throughout our empirical campaign.
> >
> > [1] Silva et al. Embarrassingly Parallel GFlowNets. ICML 2024.

---

> > > ### Author Response · Authors · 2024-11-20
> > > **Official Comment by Authors (3/4)**
> > >
> > > ## Questions
> > >
> > > Thank you for the attentive questions.
> > >
> > > > 1. … are there scenarios where SAL might fail to converge, or does it require a careful balance of parameters to function well?
> > >
> > > In our experience, the algorithm works in a relatively straightforward way and does not require a careful hyperparameter search at all. To illustrate this, we executed additional experiments for the realistic tasks of SIX6 and PHO4 with different sizes of FHPs. As we can see (Figure 14 in page 42), SAL outperforms a centralized algorithm in terms of mode discovery — regardless of the FHP’s size.
> > >
> > > In general, we are confident that SAL will significantly outperform conventional GFlowNets for any reasonable choice for the algorithm’s parameters, such as those outlined in Section C.1. Beyond the statistical perspective presented in the main text, a key reason for this is that SAL can harness substantially more computational power in a shorter time frame.
> > >
> > > > 2. Additionally, are there inherent limitations to the generalization bounds derived, particularly when applied to different classes of GFlowNets or domains with very large or complex state spaces?
> > >
> > > Our bounds are, in principle, agnostic to the nature of the problem at hand. In particular, they do not depend on the structure of the state graph or on the properties of the reward function. Nonetheless, they assume that the learning algorithm abides by the conditions outlined in Section 5.1, namely, i.i.d. trajectory sampling, which might restrict their applicability in complex domains requiring sophisticated training methods. In this regard, we believe that the approach described in Weakness#3 could significantly loosen these restrictions through the derivation of more broadly applicable generalization bounds for GFlowNets.
> > >
> > > > 3. The paper notes that traditional training of GFlowNets resembles active learning. Could you expand on how you envision incorporating active learning strategies into your framework? How might this impact the performance of GFlowNets compared to the current approach?
> > >
> > > We might have been imprecise here; we meant that there is an intuitive (although vague) similarity between GFlowNet  training and active learning. In a nutshell, a conventional approach for training a GFlowNet relies on a $\epsilon$-greedy strategy that alternates between (i) trajectory sampling and (ii) model update. The first step leverages the current model to draw informative samples from the state graph and, in this sense, resembles active learning. Canonically, a trajectory is deemed informative if it finishes at a high-reward state [1, 2, 3]. In contrast to the traditional active learning paradigm, however, there is no explicit acquisition function defining which portions of the state space would be more beneficial for training the GFlowNet. To avoid potential confusion, we inserted this discussion into the updated manuscript.
> > >
> > > [1] Kim et al. Local Search GFlowNets. ICLR 2024.
> > >
> > > [2] Malkin et al. GFlowNets and Variational Inference. ICLR 2023.
> > >
> > > [3] Pan et al. Generative Augmented Flow Networks. ICLR 2023.
> > >
> > > > 4. In the experiments, were there any signs of mode collapse when aggregating the outputs of subgraph-trained GFlowNets? What strategies would you recommend or envision to mitigate this issue during the aggregation phase?
> > >
> > > This is an interesting issue. In general, our experiments were carried out without remarkable complications. We found, however, that the trained model’s ability to accurately sample from the target might sensibly depend on the chosen distribution over the partition’s initial states (denoted by $q_{j}$ in Equation (9)). If $q_{j}$ is overly concentrated on a specific region of the $j$th component of the FHP, the resulting model may misrepresent the amount of flow within the corresponding subgraph and, as a consequence, lead to an inappropriate  distributional fit of the aggregated GFlowNet to the target (possibly due to mode collapse).
> > >
> > > As a rule of thumb, we recommend setting $q_{j}$ as an uniform distribution to maximize the diversity of samples and the coverage of the state graph during training — in the fashion of maximum entropy GFlowNets [1]. For concreteness, we have included in Section E.1 (page 42) this discussion and a specific and extreme example of a $q_{j}$ that results in mode collapse of the global model. As before,  this fabricated experiment is intended only for illustration purposes and does not represent the actual performance of SAL.
> > >
> > > Indeed, despite its potential issues, we have observed that successfully running SAL is significantly faster than accurately learning a centralized GFlowNet (Sections 6.2 and C.1). Additionally, as we briefly discussed in Section 6.1, SAL is amenable to most sophisticated techniques for GFlowNet training (e.g., replay buffer, local search), which severely mitigate the risk of mode collapse.
> > >
> > > [1] Shen et al. Towards Understanding and Improving GFlowNet training. ICML 2023.

---

> ### Author Response · Authors · 2024-11-20
> **Official Comment by Authors (4/4)**
>
> > 5. Are there empirical limits which have been identified in the experiments regarding the effectiveness of SAL as the complexity of the underlying GFlowNet increases?
>
> As mentioned, our experiments in Sections 6 and C.1 ran smoothly. As long as a FHP can be efficiently implemented, there does not seem to be clear computational constraints for SAL — independently of the complexity of the underlying GFlowNet. Currently, indeed, the main empirical limitation of SAL is its restriction to the problems described in Section C.1, namely, autoregressive, set-structured, and hypergrid navigation tasks. For specialized domains (e..g, drug discovery), devising effective FHPs is still an open challenge — worthy of a work of its own.
>
> > 6. How do the authors envision adapting the partitioning scheme to be more flexible or dynamic as the policy evolves during training?
>
> This is a tricky and compelling issue. Fundamentally, SAL is built upon a static FHP; it is unclear whether the theoretical results outlined in Section C.1 would be preserved for dynamically partitioned state graphs. We think, however, that a meta-learning approach would be ideal for addressing this issue by amortizing the learning of the partition-wise flow functions. Instead of learning a flow assignment for each component of a given FHP, we would learn a meta-model that receives a FHP as input and returns a flow assignment for each of the subgraphs. In doing so, we could harmlessly update the FHP as the policy evolves during training. Nonetheless, we are uncertain about the best way to define and train such a meta-model. A natural question is: how can such meta-learning be performed in a distributed or embarrassingly parallel fashion? A clear answer is currently out of reach, being an interesting venue to be investigated by future endeavors.
>
> As discussed in Section E, a related and simpler problem consists of finding an _optimal_ FHP for a given problem and utility function defining the goodness of a specific partition. In essence, this problem may be reduced to NAS: in both cases, we are searching for a discrete and complex structure that will enhance the training efficiency of the resulting model. Although there are standard strategies for approximately solving these problems, e.g., Bayesian optimization [1], they remain a largely unresolved issue in the machine learning literature.
>
> [1] Kandasamy et al. Neural architecture search with bayesian optimisation and optimal transport. NeurIPS 2018.
>
> Again, many thanks for the thoughtful review and for the interesting questions. We would be delighted to engage in further discussion if needed.

---

### Official Review · Reviewer_ZwMs · 2024-11-04

**Soundness:** 3
**Presentation:** 2
**Contribution:** 3
**Rating:** 5
**Confidence:** 3

**Summary:**

The paper contributes to the theory of Generative Flow Networks (GFN) on finite directed acyclic graphs.
The key contribution of the work consists in the formulation and the proof of several new so-called generalization bounds controlling the accuracy of the sampling distribution compared to its target.  The author uses these bounds as a motivation to introduce a divide-n-conquer distributed GFN training algorithms: SAL and recursive SAL. These algorithms pack several GFNs together by assuming that the initial flow of the one is the reward of another.

**Strengths:**

The paper provides new generalization bounds for Generative Flow networks together with rigorous proofs leveraging sophisticated pre-existing bounds.
The paper also introduces an interesting recursive GFN training: recursive SAL algorithm
The appendix is rich with further theoretical analysis.

**Weaknesses:**

We feel the authors do not properly exploit the bounds they produce:
- A full section is devoted to describing when GFN do not generalize. Still, the conclusion that the number of states or the length of trajectories is detrimental to training is disappointing.
- motivating corner cases are not  discussed
- The SAL algorithm does not need the bounds to be motivated, as divide and conquer is a very common strategy in CS in general.

To elaborate on this, let's consider equation 5. The hypotheses seem to allow any $p_{E,T}$, but the context provided above Lemma 4.1 suggests it is intended as the forward policy of the GFN at hand. Now, in practical cases, the forward policy will be very different from the uniform policy to the point that $\chi^2(q_{E,T}||p_{E,T})$ may diverge. I tend to agree that target distribution is key, but it is not discussed how one may leverage this. For instance, the bound suggests one may use the product $\mathcal L_{TB} \times \chi^2(q_{E,T}||p_{E,T})$ as a regularization during training, or even that modifying the reward as $R_{\delta}(x):= R(x)+\delta \varphi(x) $ for some summable function $\varphi$ and let $\delta$ go to zero controlled by $ \chi^2(q_{E,T}||p_{E,T})$ would allow controlling the generalization better. Coming back to the declared intent (describing when GFN does not generalize well), I am still looking for specific examples (even in the appendix) of GFN together with couples (graph, reward) that would typically make the bound high yet have a low loss. This would be very useful to showcase when one should be careful when using GFNs.

We feel the first four pages are too wordy. They announce the results several times, leaving too little space for the interesting matter: an insight discussion of the generalization bounds.

Section 4 is unclear; the set generation problem is barely understandable.

Some mathematical results are strangely placed. For instance, the purpose of Lemma 4.1 is unclear in the section context, while Lemma D.1 is both trivial and stated after it is used.

The recursive SAL algorithm (which we find of interest) is only introduced in the appendix.

**Questions:**

1) Could you provide insight and/or examples illustrating the meaning of the bounds you introduce?
2) I am confused about the link between the FCS and TB losses. Could the author clarify this link and how they differ? Is FCS loss used for training? Is it computable as a metric during training?
3) Are you able to provide deeper insight into the structural properties of the graphs or into the reward leading to lower generalization, deeper than the dimensionality curse?
4) How could one make your bounds actionable or act as a guardrail ?

---

> ### Author Response · Authors · 2024-11-20
> **Official  Comment by Authors (1/2)**
>
> Thank you for the detailed review and for the suggestions to enhance the presentation of our work. If you believe our answers satisfactorily address your concerns and clarify our contributions, please consider raising your score. Otherwise, we would be glad to discuss our results in further details.
>
> ## Weaknesses
>
> > A full section is devoted to describing when GFN do not generalize.
>
> We would like to clarify that the purpose of Section 4 is to argue that, depending on the exploratory distribution, learning a generalizable policy network is not possible even after observing an arbitrarily large portion of the state graph. In particular, Proposition 4.2 demonstrates that a small observed loss is not enough to ensure that the learned distribution is similar to the target. From these results, we conclude that a statistical guarantee for GFlowNets should be fine-tuned to the specific training data distribution, which is the approach we pursue in Section 5.1.
>
> > motivating corner cases are not discussed
>
> As mentioned above, we regard Section 4 as a motivating corner case for the technical results presented in Section 5. However, if you have further suggestions, we would gladly consider additional edge cases and discuss them in our work.
>
> > The SAL algorithm does not need the bounds to be motivated, as divide and conquer is a very common strategy in CS in general.
>
> Thank you for the chance to underline our motivation for SAL.
>
> While we agree that divide-and-conquer algorithms are common in computer science, this fact alone is not enough to justify the significant superior performance of SAL relatively to a centralized GFlowNet. By taking a PAC-Bayesian stance, we directly address an issue at the core of machine learning algorithms, namely, generalization. From a scientific standpoint, Sections 5.2 and 6 respectively ask (i) _what makes generalization difficult?_ and (ii) _how to facilitate the learning of a generalizable policy network?_.
>
> In  (i), we demonstrated via Theorems 5.2 and 5.4 that the trajectory length (represented by $\log t_{m}$) and the sample size ($n$) play an important role in the generalization of GFlowNets. In this sense, we introduced SAL as an actionable answer to question (ii) in view of (i): both the reduced $t_{m}$ (through the FHP) and the increased $n$ (by employing a distributed scheme) aim at facilitating the learning of a generalizable policy network.
>
> > We feel the first four pages are too wordy.
>
> > The recursive SAL algorithm (which we find of interest) is only introduced in the appendix.
>
> We appreciate your feedback. We integrated the discussion of recursive SAL into the main text to better emphasize our work’s contributions (page 10). Also, we shortened Section 3 to improve the paper’s readability.
>
> > Section 4 is unclear; the set generation problem is barely understandable.
>
> > Some mathematical results are strangely placed. For instance, the purpose of Lemma 4.1 is unclear in the section context
>
> Thank you for providing the chance to elaborate on Section 4.
>
> In short, the set generation task consists of sampling fixed-size subsets of a given set in proportion to a specific reward function; it is a standard benchmark in the GFlowNet literature (see, e.g., Bengio et al. 2023, Section 5.1).
>
> To generate them, we start from an initially empty set ($s_{o}$) and iteratively add elements to it until it achieves a prescribed size. We have included both an illustration of the corresponding state graph in Figure 5 on page 20 of the document and an updated (informal) description of the problem in the main text (Section 4).
>
> In this sense, Lemma 4.1 and Proposition 4.2 show that minimizing the empirical loss on an arbitrarily large portion of the state graph might not be enough to ensure that the model learns a generalizable policy network for the set generation task. As mentioned earlier, this demonstrates that the construction of non-vacuous statistical guarantees for GFlowNets should take into account _the specific trajectories observed during training_ rather than merely their quantity, which motivates our empirical analysis in Section 5.1.
>
>
> > Lemma D.1 is both trivial and stated after it is used.
>
>
> We agree that Lemma D.1 represents a simple and well-known fact. We have included it only to ensure that our proof is both comprehensive and complete; that is the reason it was deferred to the appendix.

---

> ### Author Response · Authors · 2024-11-20
> **Official Comment by Authors (2/2)**
>
> ## Questions
>
> > 1. Could you provide insight and/or examples illustrating the meaning of the bounds you introduce?
>
> We are happy to elaborate further on the insights from our bounds.
>
>
> **Insight 1: Small empirical loss does not imply generalization**. Proposition 4.2 shows that, depending on the trajectory distribution, a small empirical loss does not ensure that the GFlowNet accurately approximates its target. Contrapositively, this entails that a meaningful statistical guarantee can only be obtained through data-dependent bounds, as demonstrated in Section 5.1.
>
>
> **Insight 2: Generalization poses a trade-off between trajectory length and model size**. Theorems 5.2 and 5.4 present a trade-off between the description length (measured by the state graph’s diameter, $t_{m}$) and the dimensionality of the policy network’s parameter space. For instance, when generating sequences of size $2$, we may either start from an empty object and add one element at a time ($t_{m} = 2$) or simultaneously add both elements ($t_{m} = 1$). For $t_{m}=1$, we would need bigger neural networks representing larger action spaces in their output, which would negatively affect our generalization bounds through the KL term. On the other hand, setting $t_{m} = 2$ allows using smaller models ($\downarrow \mathrm{KL}$) at the cost of an increased description length ($\uparrow \log t_{m}$).  However, we acknowledge that striking a perfect balance between these two aspects remains open, and we believe it to be a fruitful venue for future works.
>
>
> > 2. I am confused about the link between the FCS and TB losses. Could the author clarify this link and how they differ? Is FCS loss used for training? Is it computable as a metric during training?
>
> Thank you for the question. From a fundamental viewpoint, when the distributional approximation of a GFlowNet is perfect, both the FCS and TB vanish.
>
> However, we note that FCS is not an appropriate learning objective for GFlowNets. First, computing FCS requires evaluating the marginal probability $p_{\top}(x)$ of a terminal state $x$, which makes it costlier than TB. Second, in contrast to TB, FCS is not in the log-domain; its optimization could be numerically unstable.
>
> On the other hand, FCS is bounded and perfectly fits in as a risk functional in a PAC-Bayesian analysis. Also, it was recently shown to be an appropriate tool for diagnosing the distributional accuracy of GFlowNets [1]. We have clarified this in the updated PDF (Section 3).
>
> [1] Silva et al. Analyzing GFlowNets: Stability, Expressiveness, and Assessment. SPIGM workshop @ ICML 2024.
>
> > 3. Are you able to provide deeper insight into the structural properties of the graphs or into the reward leading to lower generalization, deeper than the dimensionality curse?
>
> The “curse of dimensionality” (CoD) typically refers to the difficulty of solving a problem with a prescribed error depending exponentially on the dimension of the problem’s state space [1, 2]. In this sense, we are unaware of a deeper mathematical connection between the CoD and our results. In a concurrent study, for instance, Boussif et al. [3] showed that a reduction on the trajectory length of the state graph through the introduction of abstract actions improved the learning convergence of GFlowNets, in agreement with our generalization bounds. In doing so, however, the state space’s “dimension” remains unaffected — implying no direct association between the trajectory’s length and the problem’s “dimension”.
>
> [1] Donoho et al. High-Dimensional Data Analysis: The Curses and Blessings of Dimensionality. AMS Math Challenges Lecture, 2000.
>
> [2] John Rust. Using Randomization to Break the Curse of Dimensionality. Econometrica, 1997.
>
> [3] Boussif et al. Action abstractions for amortized sampling. https://arxiv.org/abs/2410.15184
>
>
> > 4. How could one make your bounds actionable or act as a guardrail ?
>
> As demonstrated in Proposition 5.1, our empirical bounds could be used as a guardrail for provably controlling the distributional accuracy of a trained GFlowNet with high probability. As a practical safeguard, if a practitioner deems the bound to be above his tolerance threshold, he could consider further training the GFlowNet before deployment.
>
>
> We appreciate the reviewer’s thoughtful questions. We would be happy to engage further, should additional clarifications be needed.

---

> ### Comment · Reviewer_ZwMs · 2024-11-28
>
> 1." We would like to clarify that the purpose of Section 4[..]" Understood, however, I fail to see how this is specific to GFNs: the reward may be located on a arbitrarily small fraction of the states and any algorithm would have similar difficulties. I agree with the conclusion of the need to fall-back to data-dependent bounds.
> 2. "As mentioned above, we regard Section 4 as a motivating corner". Let me clarify: Section 4 motivates data-dependent bounds, ok. My point is however different: when I have a theoretical bound, I want to know what it means. Examples are important to this end as they show when the bound is sharp, when some term dominate the other, when some part are big or small. This is essential to explaining such bounds. These examples should be provided at the very least in appendix, and its not my role to give them.
>
> 3. "In (i), we demonstrated via Theorems 5.2 and 5.4 that the trajectory length (represented by ) and the sample size () play an important role in the generalization of GFlowNets". My point is that, such a statement falls back to your section 4 example: say that the graph is a binary tree of depth D with reward on the leaves. Then, obviously, the length of the paths (which is D) have a significant impact on the ability to generalize (or even learn anything) as the number of relevant states (the leaves) grows as 2^D.
> The bound then says nothing more than "long paths imply high combinatorics hence hard problem".
>
> *Such an information does not help to understand generalization of GFN and is not specific to GFN*
>
> 4. "it is a standard benchmark in the GFlowNet literature" I know this is a common benchmark, my point was on the understandability of the way it is written in the present work.

---

> > ### Author Response · Authors · 2024-11-29
> >
> > We appreciate your engagement in the discussion.
> >
> > > Understood, however, I fail to see how this is specific to GFNs: the reward may be located on a arbitrarily small fraction of the states and any algorithm would have similar difficulties. I agree with the conclusion of the need to fall-back to data-dependent bounds.
> >
> > > This is essential to explaining such bounds. These examples should be provided at the very least in appendix, and its not my role to give them. Such an information does not help to understand generalization of GFN and is not specific to GFN
> >
> > Thank you for the opportunity to discuss our work in further detail. To start with, we would like to note that GFlowNets are inherently connected to both variational inference [1] and reinforcement learning [2] algorithms. As such, our results could be easily translated to these broader settings and — in this sense — are not specific to GFlowNets, as you mentioned.
> >
> > Additionally, we emphasize that the bounds in Section 5.2 provided a motivation for an algorithm. On this note, it is worth emphasizing what we believe is a connection between the results therein and the remarkable performance of SAL in Section 6: **our bounds furnish a solid foundation for the understanding of why SAL improves upon a centralized approach**, a fact that otherwise would solely rely on heuristic and potentially fragile arguments. An empirical analysis of the generalization of GFlowNets, in contrast, is presented in Sections 4 and 5.1. In this regard, instead of aiming to fully explain the generalization in GFlowNets, Section 5.2 should be regarded as a complement to Section 5.1 and as an introduction to Section 6.
> >
> > [1] Malkin et al. GFlowNets and Variational Inference. ICLR 2023.
> >
> > [2] Tiapkin et al. Generative Flow Networks as Entropy-Regularized RL. AISTATS 2024.
> >
> >
> > > 3. "it is a standard benchmark in the GFlowNet literature" I know this is a common benchmark, my point was on the understandability of the way it is written in the present work.
> >
> > We are grateful for your feedback. We hope that the updated description of the set generation task in Section 4 is sufficiently clear. If not, we are open to suggestions.

---

> > > ### Comment · Reviewer_ZwMs · 2024-11-29
> > >
> > > None withstanding the link between shortening paths and SAL,  I fail how your argument adress my comment regarding corner cases and insights beyond task complexity.

---

> > > > ### Comment · Reviewer_ZwMs · 2024-12-02
> > > >
> > > > All minor points I raised were addressed, and I recognize the theoretical contribution of the provided bound.
> > > >
> > > > However, I stand by my initial point: I expect such bounds to be discussed to provide apriori insight or be used directly to build a loss. As is, I am neither satisfied by the discussion provided in the paper nor by the answers to this question in the discussion. I think I am merely asking for a few sentences and/or a more elaborated discussion in appendix, which the author could easily provide.
> > > >
> > > >
> > > > In a mathematical litterature, i would accept such a bound as is. Since many work in the AI litterature tend to mistake theorical bounds for theoretical insight, I am not willing to raise my score.

---

### Official Review · Reviewer_7dqe · 2024-11-04

**Soundness:** 2
**Presentation:** 3
**Contribution:** 2
**Rating:** 6
**Confidence:** 3

**Summary:**

The authors develop the generalization bounds for GFlowNets, providing empirical guarantees that the models can generalize beyond the training data, using the PAC-Bayes technique. The authors explore the way how the structure of GFlow's network and trajectory lengths affect its generalization performance. Then the authors suggest a novel distributed algorithm called Subgraph Asynchronous Learning (SAL), which is shown to improve over its centralized counterparts in some experimental settings, such as subset generation, sequence design, and Hypergrid environments (Section 6).

**Strengths:**

To the best of my knowledge, neither the PAC-Bayes generalization bounds, nor the distributed training of Gflownets were previously considered in the literature, so both main contributions of the paper appears to be new. The literature review performed by the authors, is exhaustive and well-written, as well as the main text of the paper.

**Weaknesses:**

From the theoretical point of view, theoretical contribution of the paper is rather incremental, as the main theoretical findings of the paper are mostly corollaries from known results in PAC-Bayes literature. Moreover, the supervised setting for training GFlowNets is questionable and is of limited interest, since its implementation requires a sampler from the target distribution.

From the experimental point of view, I would also say that the proposed experimental investigation of the Subgraph Asynchronous Learning algorithm is insufficient. All considered environments can be considered as rather toyish, with actual number of terminal states less than $10^{10}$. This is quite far from the most challenging environments, for example, see for example the molecular generation problems from the recent papers [Jang et al, 2024], [Mohammadpour et al, 2024].

Although minor, I would also highlight some stylistic nuances in the exposition. For instance, the authors describe their own results in Theorem 5.2 as 'insightful' (lines 204 and 2130). It is uncommon in the literature to use such language when referring to one's own findings. Similarly, the statistical guarantees presented in the current submission, are often referred to as "non-vacuous" bounds. At the same time, it is not clear to me what exactly "non-vacuous" means in the context of statistical guarantees, which seems to be an overstatement.

***Minor typos***
1. I guess, that in line 148 $p_E$ should be defined as $p_E = (1-\epsilon)p_F + \epsilon p_{U}$;
2. page 3, line 149 - polciy -> policy;
3. Proposition 5.1 - $\tau_{\alpha}$ is undefined, only $\tau_{1 - \alpha}$ previously appeared;
4. page 9, line 223: assigment function -> assignment function;
5. line 979-980 - [ref] undefined;
6. General remark - the authors should be more accurate when using the $\lesssim$ symbol. At least it should be indicated, which problem-specific constants are omitted when it is used.

***References***
[Jang et al, 2024] Hyosoon Jang, Yunhui Jang, Minsu Kim, Jinkyoo Park, and Sungsoo Ahn. Pessimistic backward
policy for Gflownets, 2024.

[Mohammadpour et al, 2024] Sobhan Mohammadpour, Emmanuel Bengio, Emma Frejinger, and Pierre-Luc Bacon. Maximum
entropy gflownets with soft Q-learning. In International Conference on Artificial Intelligence and Statistics, pp. 2593–2601. PMLR, 2024.

**Questions:**

1. Is it possible to provide larger-scale experiments? For example, consider the applications of Subgraph Asynchronous Learning to molecular generation problems, such as QM9 or sEH, see [Mohammadpour et al,2024].

2. Can the authors provide more details on the way the Fixed-horizon DAG partition (definition 6.1) should be constructed? May be there are some heuristics or theoretically justified approached to it? It seems that the choice of partition should be of crucial importance for algorithm's performance, otherwise this partition itself can yield mode collapse and other issues with diversity of the samples.

***References***

[Mohammadpour et al, 2024] Sobhan Mohammadpour, Emmanuel Bengio, Emma Frejinger, and Pierre-Luc Bacon. Maximum
entropy gflownets with soft Q-learning. In International Conference on Artificial Intelligence and Statistics, pp. 2593–2601. PMLR, 2024.

---

> ### Author Response · Authors · 2024-11-20
> **Official Comment by Authors (1/3)**
>
> Thank you for carefully reviewing our paper and for acknowledging its novelty. We hope our answers below elevate your appraisal of our work. If so, please consider raising your score. Otherwise, we will be happy to engage further.
>
> ## Weaknesses
>
> > From the theoretical point of view, theoretical contribution of the paper is rather incremental, as the main theoretical findings of the paper are mostly corollaries from known results in PAC-Bayes literature
>
> Thank you for the chance to highlight the novelty of our contributions. From a technical standpoint, Theorem 5.2 introduced a strategy for bounding the GFlowNet’s policy network away from zero and a corresponding upper-bound on the KL divergence learning objective. On the other hand, in Theorem 5.4, we developed an Azuma-Hoeffding-style inequality for independently distributed martingales that, to the best of our knowledge, was not previously established in the literature.
>
> Although these results may not represent a major advance in the PAC-Bayesian realm, we would like to emphasize that our focus was on shedding light on the generalization properties of GFlowNets through the lens of PAC-Bayesian analysis. For this, we provided both empirically tight guarantees and a clearer understanding of GFlowNet generalization through oracle bounds. This is our primary theoretical contribution, which touches on an issue that has been of large interest in the GFlowNet literature [1, 2].
>
> On a side note, we also underline that similar studies were carried out for, e.g., diffusion probabilistic models [3], GANs [4], and LLMs [5, 6], with the sole objective of better understanding the generalization behavior of the corresponding models. Although these works might be seen as incremental from the perspective of the PAC-Bayesian literature, they are relevant and important for their respective communities.
>
> [1] Bengio et al. Flow Network based Generative Models for Non-Iterative Diverse Candidate Generation. NeurIPS 2021.
>
> [2] Shen et al. Towards Understanding and Improving GFlowNet training.ICML 2023.
>
> [3] Li et al. On the Generalization Properties of Diffusion Models. NeurIPS 2023.
>
> [4] Mbacke et al. PAC-Bayesian Generalization Bounds for Adversarial Generative Models. ICML 2023.
>
> [5] Lotfi et al. Unlocking Tokens as Data Points for Generalization Bounds on Larger Language Models. NeurIPS 2024.
>
> [6] Lotfi et al. Non-Vacuous Generalization Bounds for Large Language Models. ICML 2024.
>
> >  Moreover, the supervised setting for training GFlowNets is questionable and is of limited interest, since its implementation requires a sampler from the target distribution.
>
> Thank you for the opportunity to clarify our assumptions. We start by noticing that **none of our experiments assume that samples from the target distribution are available**. Also, except for Theorem 5.2, our theoretical results are not restricted to any data distribution in particular.
>
> In Section 5.1, we (perhaps unfortunately) used the term “supervised learning” to illustrate the shift from the traditional GFlowNet training paradigm into a supervised learning problem, which was also considered by [1], for a better fit in the conventional PAC-Bayesian ontology. This was only a didactic artifact. In fact, the **only supervision comes from the reward function**. We integrated this discussion into the updated manuscript.
>
> [1] Atanackovic et al. Investigating Generalization Behaviours of Generative Flow Networks. SPIGM workshop at ICML 2024.

---

> ### Author Response · Authors · 2024-11-20
> **Official Comment by Authors (2/3)**
>
> > From the experimental point of view, I would also say that the proposed experimental
> investigation of the Subgraph Asynchronous Learning algorithm is insufficient. All considered environments can be considered as rather toyish, with actual number of terminal states less than $10^{10}$.
>
> To the best of our knowledge, our experimental campaign is on par with the wider GFlowNet literature in terms of diversity of the generative tasks [1, 2, 3, 4, 5], encompassing six different problems that are **often used to benchmark GFlowNets**. As we demonstrated in the main text and in the supplement, SAL significantly improved upon a centralized approach for all the considered experiments. We believe this provides strong evidence for the effectiveness of our distributed algorithm.
>
> As we acknowledged in Sections 7 and E, however, experiments on specialized domains such as drug discovery (molecule generation) comprise a promising venue for the evaluation of SAL. Nonetheless, designing a fixed-horizon partition for these environments would require the development of expert techniques that deserve a work of their own. In our paper, we developed strategies for devising a FHP in the context of set generation, sequence design, and hypergrid navigation (please refer to Section C.1).
>
> To ensure that SAL scales beyond the size of the considered generative tasks, we executed additional experiments on the generation of $50$-sized subsets of a $100$-sized fixed set (~$10^{30}$ elements) in Section E.1 (page 42). As we can see, a SAL-trained GFlowNet also achieves a better distributional approximation and coverage of the state graph than its centralized counterpart in this relatively large domain.
>
> [1] Malkin et al. GFlowNets and Variational Inference. ICLR 2023.
>
> [2] Malkin et al. Trajectory balance: Improved credit assignment in GFlowNets. NeurIPS 2022.
>
> [3] Pan et al. Better Training of GFlowNets with Local Credit and Incomplete Trajectories. ICML 2023.
>
> [4] Silva et al. On Divergence Measures for Training GFlowNets. NeurIPS 2024.
>
> [5] Shen et al. Toward Understanding and Improving GFlowNet training. ICML 2023.
>
> > Although minor, I would also highlight some stylistic nuances in the exposition. For instance, the authors describe their own results in Theorem 5.2 as 'insightful' (lines 204 and 2130). It is uncommon in the literature to use such language when referring to one's own findings.
>
> Thank you for your suggestion. By “insightful”, we intended to mean that the derived bound carried useful information regarding the generalization properties of GFlowNets. To avoid a potential overstatement, we have replaced “insightful” by “informative”. We are open to further feedback.
>
> > Similarly, the statistical guarantees presented in the current submission, are often referred to as "non-vacuous" bounds.
>
> We are thankful for your detailed feedback. However, we note that “non-vacuous” is a standard terminology in the PAC-Bayesian literature [1, 2]. In a nutshell, a statistical guarantee is _vacuous_ if it is trivially true. In the context of Proposition 5.1, for example, we would call a high-probability bound of the form $L_{FCS} \le 1.8$ _vacuous_, as $L_{FCS} \le 1$ by design. By stating the verifiable non-vacuousness of our results in Section 5.1, we mean that the chosen risk functional ($L_{FCS}$) is non-trivially upper bounded with high probability; this is shown in Figure 2. Having said that, we understand that the term may appear an overstatement. We added a clarification on its context and meaning in Section 2.
>
> [1] Pierre Alquier. User-friendly introduction to PAC-Bayes bounds. Foundations and Trends® in Machine Learning, 2024.
>
> [2] Lotfi et al. Non-Vacuous Generalization Bounds for Large Language Models. ICML 2024.
>
> > Minor typos.
>
> Thank you for catching these typos. We corrected them in the updated manuscript.
>
> > 1. Proposition 5.1 - $\tau_{\alpha}$ is undefined, only $\tau_{1−\alpha}$ previously appeared;
> > 5. line 979-980 - [ref] undefined;
>
> Briefly, $\mathcal{T}\_{\alpha}$ is the complement of $\mathcal{T}\_{1 - \alpha}$ and [ref] refers to Boucheron et al. (2013).
>
> ## Questions
>
> > Is it possible to provide larger-scale experiments?
>
> As we mentioned, it will not be possible to run large-scale drug discovery experiments during the discussion period. Nonetheless, our results on the larger set generation tasks (see above) indicate that SAL can be scaled to bigger state graphs. We also stress that the considered benchmark tasks are of potentially practical interest and standard in the evaluation of novel GFlowNet learning algorithms.

---

> ### Author Response · Authors · 2024-11-20
> **Official Comment by Authors (3/3)**
>
> > Can the authors provide more details on the way the Fixed-horizon DAG partition (definition 6.1) should be constructed?
>
> Thank you for your question. In practice, a FHP is only implicitly defined via an _assignment function_ (AF) that separates the states into (possibly overlapping) subgraphs. In doing so, we do not need to explicitly construct a  FHP. This AF is defined via either a hash- or ranking-based indexation of the state space. The former is standard in, e.g., PostgreSQL-based databases, and ensures that the state graph is approximately homogeneously divided between each component of the partition. For a lengthy discussion on how to construct a FHP, we refer the reviewer to Section C.1 in the appendix.
>
> > May be there are some heuristics or theoretically justified approached to it?
>
> You correctly noted that a poorly designed FHP would not be beneficial for the training of GFlowNets. In Section E.1, we have included additional experiments illustrating this point in extreme and fabricated cases. However, similar observations could be made regarding an inappropriately chosen exploratory policy (as examined in Section 4) or loss function in conventional GFlowNets. In this sense, the efficacy of a particular FHP is ultimately an empirical problem that should be experimentally investigated.
>
> In our work, our partitioning strategy described in Section C.1 was based on the heuristic of allocating similarly sized subgraphs to each of FHP’s components, which was demonstrably effective in practice (Section 6.2). In particular, we did not observe either mode collapse or insufficient sample diversity throughout our experimental campaign. Intuitively, the main reason for this is that SAL greatly increases the exploratory potential of a standard GFlowNet by distributing the task of traversing the state graph. As a consequence, issues such as mode collapse are far less likely under SAL than under a centralized GFlowNet.
>
> That said, although our heuristics for constructing a FHP significantly enhanced GFlowNet training, we agree that the design of theoretically optimal FHPs is an interesting direction for future works — as stated in Section E.
>
> We again thank the reviewer for their feedback. We would be happy to further discuss our contributions if needed.

---

> > ### Comment · Reviewer_7dqe · 2024-12-02
> >
> > This is true that the benchmarks chosen by the authors are standard, however, they are typically followed by more complicated experiments (for example, I referenced two particular molecular generation environments - sEH and QM9). Thus, not questioning the validity of the proposed examples, I would still suggest the authors to consider further experimental evaluation of their methods. Moreover, I bring the author's attention to the fact, that many of the conclusions about the validity and advantages of the method are conducted based on simple Minigrid environment (see Appendix C and E), with size of Minigrid being 12 x 12. Even inside the Minigrid framework one can consider more convincing evaluation based on larger Hypergrid dimension. At the same time, the simple examples provided by the authors are useful and provide intuition about their method.
> >
> > I still feel that the theoretical contribution of the paper is rather incremental, but, given the rebuttal provided by the authors, i will increase my score to 6.

---

### Author Response · Authors · 2024-12-04

Dear reviewers and chairs,

Thank you for your service.

As the discussion period draws to a close, we would like to thank the reviewers for their thoughtful feedback and for dedicating their valuable time to the peer reviewing process. We also appreciate their acknowledgement of our contributions. We made our best effort to address the raised concerns within the time that we had, which we believe significantly improved the paper.

Best regards,

Authors.

---

### Meta-Review · Area_Chair_Md5C · 2024-12-26

**Metareview:**

The paper presents two main contributions in the field of Generative Flow Networks (GFlowNets): data-dependent generalization bounds using PAC-Bayesian analysis and a new distributed training algorithm called Subgraph Asynchronous Learning (SAL). The theoretical work provides new insights into GFlowNet generalization, while SAL introduces a divide-and-conquer approach to improve training scalability.
The paper demonstrates strong technical merits through its clear writing, systematic derivations, and logical structure. Despite  several significant limitations that were identified by the reviewers such as incremental theoretical contributions and  experimental validation that  lacks sufficient complexity to demonstrate real-world applicability the reviewers had consensus that the paper has enough merits to be accepted.

**Additional Comments On Reviewer Discussion:**

Future iterations should include more diverse and complex experimental evaluations, focusing on more practical domains. Maybe beyond the scope of the present work, but computational challenges in SAL, particularly its overhead and risk of mode collapse during aggregation, that could improve applicability, may need to be clearly highlighted in the revised version.

---

### Decision · Program_Chairs · 2025-01-22

Accept (Poster)